# Depth in Motion: Robust Self-Supervised Learning via Representation-Optimization-Supervision Synergy

## Abstract

Self-supervised monocular depth estimation recovers scene geometry from unlabeled monocular videos, yet its reliance on photometric constancy tends to cause failures in dynamic scenes: motion and occlusion corrupt correspondences, bias optimization toward texture-sparse regions, and drive residuals into heavy-tailed distributions that undermine supervision. To address these challenges, we propose a Representation–Optimization–Supervision Synergy Network (ROSS-Net), which establishes a holistic defense by restructuring the entire estimation flow to mitigate interlinked failure modes. At the representation level, the Motion-Anomaly Filter (MAF) validates correspondences across appearance, feature, and temporal cues to filter motion-induced mismatches while preserving dynamic evidence. At the optimization level, the Spectral-Discrepancy Calibrator (SDC) calibrates depth-axis spectra to counter low-frequency optimization bias while adding no inference-time overhead. At the supervision level, the View-Consistent Integrator (VCI) trims and reweights outlier residuals, converting noisy reprojections into robust supervision. Experiments on KITTI and NYUv2 show that ROSS-Net significantly outperforms prior methods under motion and occlusion, and generalizes strongly to unseen domains such as Make3D and ScanNet.

## 1 Introduction

Depth estimation from monocular video has become a core task in computer vision, driving applications in autonomous driving (Wang et al., 2019), robotics (Dong et al., 2022), and human–robot interaction (Robinson et al., 2023). Recent progress in self-supervised learning has propelled this field by framing training as photometric reconstruction under epipolar geometry (Godard et al., 2017; Han et al., 2022; Lyu et al., 2021; Wu et al., 2025; Zhou et al., 2017), achieving strong results in static environments. However, in dynamic scenes the rigid-scene and illumination-constancy assumptions are often violated, injecting deviations that existing formulations struggle to mitigate.

To address such disruptions, prior work handles departures from the photometric assumption through minimum reprojection (Godard et al., 2019; Guizilini et al., 2020), semantic masking (Feng et al., 2022; Dong et al., 2024), motion compensation via optical flow warping (Miao et al., 2023), teacher–student distillation (Watson et al., 2021; Bangunharcana et al., 2023), and multi-frame cost volumes (Wu et al., 2023). These measures prove effective when departures from rigidity or illumination constancy are minor. Nonetheless, three critical issues remain, as illustrated in Fig. 1. First, **edge misalignment:** appearance-only edges are retained while weak-appearance depth discontinuities are neglected, leading to false matches and missed correspondences. Second, **gradient misalignment:** low cosine alignment between the image gradient and the mapped depth gradient widens the matching peak and destabilizes optimization. Third, **heavy-tailed photometric errors:** when dynamics, specularities, and illumination changes contaminate all views, minimum reprojection fails to isolate clean supervision and binary masks discard the probabilistic structure for reliability-aware weighting, resulting in supervision dominated by outliers. Taken together, the above strategies alleviate surface effects rather than address the underlying failure modes.

Building on these observations, we propose the Representation–Optimization–Supervision Synergy Network (ROSS-Net), a self-supervised framework for monocular and multi-frame depth estimation in dynamic scenes structured into three synergistic levels (Fig. 1). At the **representation** level, the

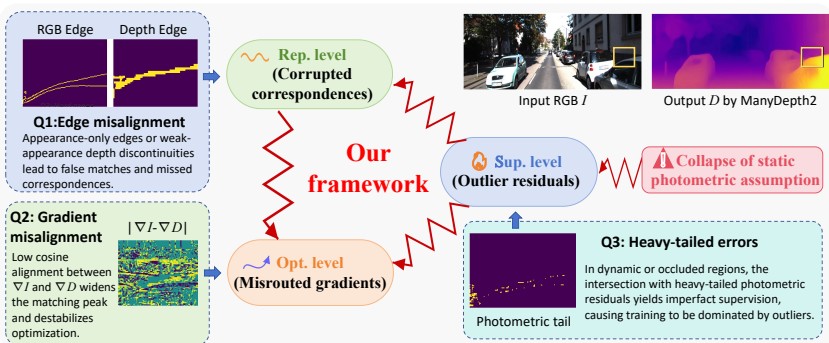

Figure 1: **Motivation and framework overview.** RGB–Depth mapping highlights three critical violations of the static photometric assumption: **Q1:** edge misalignment (appearance-only edges vs. weak-appearance depth discontinuities), **Q2:** gradient misalignment (poor $\nabla I$–$\nabla D$ alignment), and **Q3:** heavy-tailed photometric errors in dynamic or occluded regions. These observations motivate a three-level framework for holistic defense across representation, optimization, and supervision.

Motion-Anomaly Filter (MAF) applies geometric gating informed by RGB–depth edge association and temporal epipolar consistency, producing cleaner correspondences and sharper cost volumes to resolve edge misalignment. At the **optimization** level, the Spectral-Discrepancy Calibrator (SDC) derives per-pixel uncertainty from spectral entropy with frequency-aware weighting and leverages this estimate in training without inference-time overhead to mitigate low-frequency dominance and correct gradient misalignment. At the **supervision** level, the View-Consistent Integrator (VCI) transforms heavy-tailed reprojection residuals into bounded, calibrated supervision via robust consensus mechanisms, limiting outliers while preserving partial consistency. Trained end-to-end, ROSS-Net produces cleaner correspondences, mitigates frequency bias in optimization, and stabilizes supervision against outliers, thereby preserving boundaries and thin structures and demonstrating competitive accuracy with strong cross-dataset generalization.

Our contributions can be summarized as follows:

• We propose ROSS-Net, a self-supervised framework for monocular and multi-frame depth estimation that establishes a holistic defense across representation, optimization, and supervision, enabling robustness to dynamic motion and occlusions.

• ROSS-Net leverages three synergistic strategies: MAF enforces spatio-temporal epipolar validity for cleaner correspondences and sharper cost-volume peaks; SDC applies entropy-derived, frequency-aware weighting to suppress low-frequency bias without inference-time overhead; and VCI transforms heavy-tailed residuals into bounded, calibrated consensus through trimmed quantiles, logistic credibility, and noisy-OR aggregation.

• Comprehensive experiments on KITTI and NYUv2, together with cross-dataset transfer to Make3D and ScanNet, demonstrate that ROSS-Net achieves state-of-the-art accuracy and robustness in dynamic and occluded scenes.

## 2 RELATED WORK

**Self-Supervised Monocular Depth Estimation.** Self-supervised monocular depth estimation has emerged as a dominant paradigm by removing the need for dense ground-truth depth. Zhou *et al.* (Zhou et al., 2017) introduced a seminal framework that jointly learns depth and ego-motion via photometric reconstruction between adjacent frames, establishing that scene geometry can be learned directly from raw videos. Building on this foundation, Godard *et al.* (Godard et al., 2019) improved stability with minimum-reprojection loss and auto-masking, forming a widely adopted baseline. Subsequent work broadened the design space along complementary axes: geometry-aware formulations (Yang et al., 2020; Watson et al., 2021) incorporate stereo or multi-view constraints to enhance scale consistency; optimization strategies introduce priors such as edge-aware smoothness (Chen et al., 2023) and semantics (Lin & Li, 2024) to better preserve structure; and representation capacity is strengthened by attention/Transformer backbones (Gao et al., 2025; Shim & Kim,

2023) and adaptive discretization (Bhat et al., 2021) for multi-scale refinement. Despite these advances, most models remain dependent on the static photometric constancy assumption. In dynamic scenes, moving objects and occlusions corrupt correspondences, structured errors bias optimization toward low-frequency regions, and photometric residuals become heavy-tailed, ultimately undermining training stability. These limitations suggest that handling correspondence reliability, optimization bias, and outlier-contaminated supervision in dynamic settings remains an open challenge.

**Robustness under Dynamic Scenes.** To cope with dynamics and occlusions, existing methods adopt various heuristics and structured strategies. Early remedies rely on heuristic masking, most notably the auto mask in Monodepth2 (Godard et al., 2019), which suppresses dynamic pixels by discarding samples incompatible with a static warp but also removes partially consistent evidence. More structured approaches introduce explicit motion handling, either by disentangling object and camera motion via segmentation (Zhou et al., 2019) or by adding motion-specific branches that jointly predict depth and object motion (Zhou et al., 2025), capturing dynamics more directly at the cost of extra modules and hyperparameters. Occlusion-aware consistency constraints (Feng et al., 2022) further mitigate invisible or corrupted pixels, although they tend to be conservative near boundaries. Beyond representation, robust photometric penalties such as truncated or census-style objectives (Choi et al., 2025) improve outlier tolerance but often downweight ambiguous yet informative regions indiscriminately. Recent self-supervised depth methods introduce dynamic masks and uncertainty-aware fusion at the loss and cost-volume levels to suppress unreliable regions (Chen et al., 2024), but still mainly treat dynamics as noise. In contrast, our MAF provides an inference-time dual-domain parallax mask that gates temporal cost-volume construction, preserving weak yet consistent dynamic evidence rather than discarding it.

## 3 METHODOLOGY

### 3.1 MOTIVATION AND OVERVIEW

Self-supervised monocular depth estimation has witnessed substantial advances, yet its reliance on static photometric consistency undermines robustness in dynamic scenes. Under motion and occlusion, correspondences become unreliable, which simultaneously manifests as *edge misalignment* in cost-volume evidence, *gradient misalignment* that steers optimization toward texture-invariant low-frequency solutions (Sun et al., 2023; Watson et al., 2021; Rahaman et al., 2019; Miao et al., 2023), and *heavy-tailed residuals* that destabilize supervision. Together, these coupled failures corrupt geometric evidence and degrade depth accuracy.

To address these challenges, we propose a Representation–Optimization–Supervision Synergy Network (ROSS-Net), a framework that establishes a holistic defense by directly targeting each failure mode. **At the representation level**, the **Motion-Anomaly Filter (MAF)** reduces edge misalignment by purifying multi-frame correspondences and suppressing dynamic mismatches in the cost volume. **At the optimization level**, the **Spectral-Discrepancy Calibrator (SDC)** corrects gradient misalignment by recalibrating depth updates with frequency-aware uncertainty. **At the supervision level**, the **View-Consistent Integrator (VCI)** attenuates heavy-tailed residuals through order-statistic consensus, yielding calibrated supervision from noisy reprojections. In synergy, these components form a coherent reliability pipeline for stable depth learning under motion and occlusion, as illustrated in Fig. 2.

### 3.2 REPRESENTATION LEVEL: MOTION-ANOMALY FILTER

Multi-frame matching in dynamic scenes is fragile under motion, occlusion, and strong parallax, which introduce correspondence outliers and bias cost-volume aggregation. As shown in Fig. 3, MAF is a representation-level allocation module that selects displacement-salient regions through an appearance-feature change gate and stabilizes their contributions over time before warping-based matching. MAF does not use unwarped differences as a rigidity test or a dynamic static classifier, since physical consistency is evaluated by reprojection-based correlation and is further bounded by SDC and VCI.

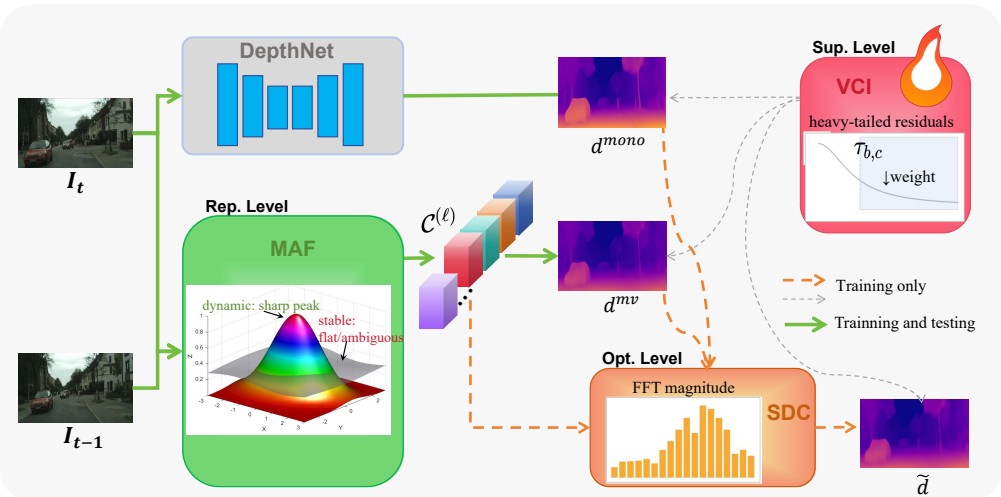

Figure 2: **Overview of ROSS-Net.** It integrates MAF, SDC, and VCI to jointly mitigate dynamic mismatches, optimization bias, and residual outliers, enabling stable and reliable depth estimation.

**Appearance–feature change agreement.** Given two consecutive frames, we first detect salient appearance change by (Yu et al., 2025a) as a high-recall cue:

$$\mathcal{K}^{\text{seed}}(u, v) = \mathbb{I}\Big[ \big\| I_t(u, v) - I_{t-1}(u, v) \big\|_1 > \varepsilon \Big], \tag{1}$$

where $\varepsilon$ controls the sensitivity. $\mathcal{K}^{\text{seed}}$ is a change flag, not a reliability weight.

At pyramid level $\ell$, a shared encoder $E(\cdot)$ produces features $\mathbf{F}_t^{(\ell)}$ and $\mathbf{F}_{t-1}^{(\ell)}$, and we measure feature-space change with threshold $\zeta_\ell$ by (Cui et al., 2021) as:

$$\mathcal{G}_\ell(u', v') = \mathbb{I}\Big[ \big\| \mathbf{F}_t^{(\ell)}(u', v') - \mathbf{F}_{t-1}^{(\ell)}(u', v') \big\|_2 > \zeta_\ell \Big]. \tag{2}$$

Combining the two cues yields the appearance–feature change gate as:

$$\mathcal{K}_\ell^{\text{cons}}(u', v') = \mathbb{I}\Big[ \mathcal{K}_\ell^{\text{seed}}(u', v') = 1 \ \wedge \ \mathcal{G}_\ell(u', v') = 1 \Big]. \tag{3}$$

where $\mathcal{K}_\ell^{\text{seed}}$ is the downsampled $\mathcal{K}^{\text{seed}}$ aligned to level $\ell$. This gate keeps regions that are simultaneously salient in pixel and feature space, allocating cost-volume capacity to displacement-rich structures while filtering out weak photometric fluctuations.

**Temporal-gated cost volume.** To suppress transient spikes and enforce temporal coherence, we first smooth the gate with an exponential moving average (EMA) (Cai et al., 2021):

$$\mathcal{K}_{\ell, t}(u', v') = \begin{cases} \mathcal{K}_\ell^{\text{cons}}(u', v'), & t = 1, \\ \eta \, \mathcal{K}_{\ell, t-1}(u', v') + (1 - \eta) \, \mathcal{K}_\ell^{\text{cons}}(u', v'), & t > 1 \end{cases} \tag{4}$$

where $\eta \in [0, 1]$ controls the temporal smoothing strength in the EMA.

The $\mathcal{K}_{\ell, t}(u', v')$ is then used to modulate features:

$$\mathbf{F}_{\{t, t-1\}}^{\prime(\ell)}(u', v') = \mathcal{K}_{\ell, t}(u', v') \odot \mathbf{F}_{\{t, t-1\}}^{(\ell)}(u', v'). \tag{5}$$

The level-$\ell$ cost volume is final constructed by warping-based correlation:

$$\mathcal{C}^{(\ell)}(u', v', d) = \Big\langle \mathbf{F}_t^{\prime(\ell)}(u', v'), \ \mathbf{F}_{t-1}^{\prime(\ell)}\big(\pi(u', v', d)\big) \Big\rangle, \tag{6}$$

where $\pi(u', v', d)$ reprojects $(u', v')$ under depth $d$ and becomes $(u' - d, v')$ in rectified stereo. After aligning all pyramid levels to a common resolution, we stack $\mathcal{C}^{(\ell)}$ to form the unified cost volume $\mathcal{C}$.

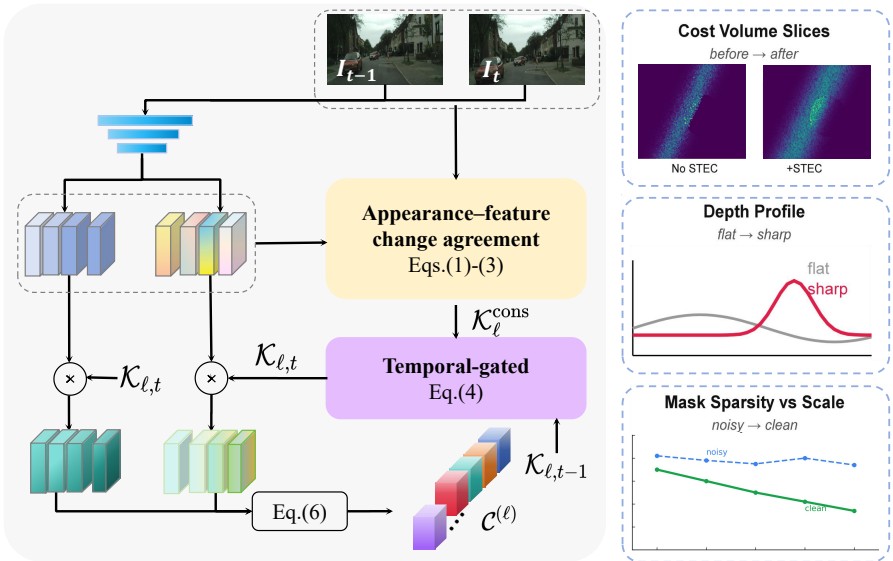

Figure 3: **Illustration of MAF.** It validates correspondences across appearance, feature, and temporal cues, producing cleaner cost volumes, sharper depth profiles, and sparser mask.

### 3.3 OPTIMIZATION LEVEL: SPECTRAL-DISCREPANCY CALIBRATOR

Even with MAF, the cost volume in dynamic and occluded regions can still mislead optimization, either by dispersing likelihood mass across depths or by producing a sharp but wrong peak when geometry is violated. SDC addresses both without hard thresholding raw entropy by learning frequency-aware spectral calibration and mapping calibrated evidence to pixelwise uncertainty for training-time monocular–multi-view residual fusion without extra inference overhead (see Fig. 4).

**Spectral-aware adaptive entropy calibration.** We obtain a normalized depth-likelihood tensor by applying a sigmoid $\sigma$ to the cost volume and then normalizing along the depth dimension, yielding $\hat{\mathcal{C}} = \mathcal{N}_d\big(\sigma(\mathcal{C})\big)$. With the depth-axis Fourier transform $\mathcal{F}_d(\cdot)$, we define the normalized spectral magnitude $\Psi_m = |\mathcal{F}_d(\hat{\mathcal{C}})|_m / (|\mathcal{F}_d(\hat{\mathcal{C}})|_1 + \delta_s)$, where $\delta_s$ is a stabilizer, and concentrated spectra indicate coherent geometric evidence whereas diffuse spectra suggest ambiguous matching.

To adaptively weigh frequency cues, we assign each spectral bin $m$ a confidence score $c_m$ that calibrates the raw entropy $\mathcal{E}_r$ into the adaptive evidence:

$$\mathcal{E}_{\text{SDC}} = - \underbrace{\sigma\big(A\,\phi(\Psi_m) + b\big)}_{\text{confidence score } c_m} \odot \underbrace{\Big(\Psi_m \odot \ln(\Psi_m + \delta_e)\Big)}_{\text{raw entropy } \mathcal{E}_r}. \tag{7}$$

where $\phi(\cdot)$ is a lightweight spectral encoder over frequency bands, $A$ and $b$ are learnable parameters, and $\delta_e$ is a stabilizer for $\ln(\Psi_m)$ near zero.

**Uncertainty-guided fusion.** We feed this entropy evidence into a regression head $\varphi(\cdot)$ to obtain the uncertainty map $\mathcal{U} = \varphi(\mathcal{E}_{\text{SDC}})$, which we use to blend co-registered predictions via residual correction (Chen et al., 2024; Wang et al., 2023):

$$\widetilde{d} = d^{\text{mv}} + \mathcal{U}\left(d^{\text{mono}} - d^{\text{mv}}\right), \tag{8}$$

where $\mathcal{U} \in [0, 1]$ serves as an uncertainty-aware residual weighting: it controls the injection of the monocular–multi-view discrepancy $(d^{\text{mono}} - d^{\text{mv}})$ into the multi-view estimate $d^{\text{mv}}$ (with $d^{\text{mono}}$ as the monocular counterpart; see Appendix A.5). $\mathcal{U} = 0$ leaves $d^{\text{mv}}$ unchanged, while higher values apply stronger residual correction, yielding $\widetilde{d}$ closer to $d^{\text{mono}}$.

### 3.4 SUPERVISION LEVEL: VIEW-CONSISTENT INTEGRATOR

Dynamic motion and occlusions push reprojection errors into heavy-tailed, heteroscedastic regimes, so standard photometric supervision becomes outlier-dominated and hard masking discards partially

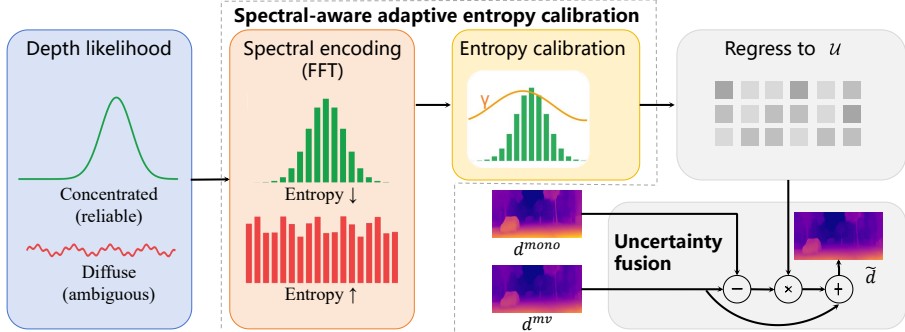

Figure 4: **Illustration of SDC.** It performs spectral entropy reweighting to accentuate high-entropy discontinuities, regresses a spatial uncertainty map $\mathcal{U}$, and leverages it for uncertainty-informed residual fusion during training—stabilizing gradients without incurring inference-time overhead.

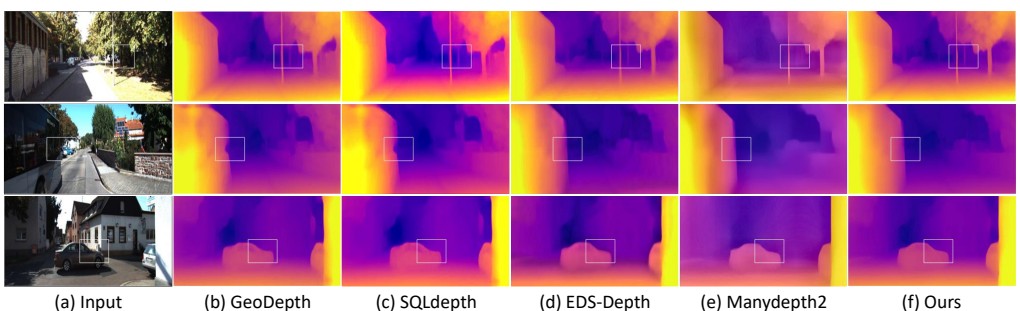

Figure 5: Qualitative results on the KITTI dataset. Our ROSS-Net recovers sharper object boundaries and more consistent depth in dynamic regions compared to prior methods.

consistent cues. VCI scores residual credibility via order statistics and softly aggregates multi-view supervision, suppressing extreme outliers while preserving usable dynamic evidence for calibrated training.

Given the per-source residual tensor $\mathcal{R} \in \mathbb{R}^{B \times C \times H \times W}$, where $c \in \{1, \dots, C\}$ indexes source frames, we collect spatial samples $\mathcal{S}_{b,c} = \{\mathcal{R}_{b,c}(u,v) \mid (u,v) \in \Omega\}$ and estimate a robust residual scale $\tau_{b,c}$ using a trimmed estimator that discards the largest $\kappa$ portion of values and returns the $\vartheta$ percentile of the remaining set, thereby bounding the influence of heavy tails (Chen et al., 2024). Residuals are then mapped to credibility scores by a logistic transform:

$$W_{b,c}(u,v) = \sigma\Big(\lambda\left(\tau_{b,c} - \mathcal{R}_{b,c}(u,v)\right)\Big), \tag{9}$$

where $\lambda > 0$ controls sharpness. Small residuals yield high credibility, while large ones are smoothly attenuated rather than hard-rejected.

To aggregate responses across multiple views, VCI employs a noisy-OR aggregation $r_b(u,v) = 1 - \prod_{c=1}^{C}\left(1 - W_{b,c}(u,v)\right)$, which preserves supervision whenever any source agrees, while strongly suppressing jointly inconsistent pixels. Thus the photometric objective is modulated by the consensus field:

$$\mathcal{L}_{\text{VCI}} = \sum_{b} \sum_{(u,v) \in \Omega} r_b(u,v)\,\mathcal{L}_{\text{ph}}(u,v; D_b) + \gamma\,\mathcal{L}_s(D_b), \tag{10}$$

where $\mathcal{L}_{\text{ph}}$ is the photometric loss, $\mathcal{L}_s$ is edge-aware smoothness (see Appendix A.1), and $\gamma$ is a weighting coefficient (Godard et al., 2019). By this consensus-modulated loss, the total training loss jointly supervises the monocular ($d^{\text{mono}}$), multi-view ($d^{\text{mv}}$), and uncertainty-calibrated ($\widetilde{d}$) predictions:

$$\mathcal{L}_{\text{total}} = \mathcal{L}_{\text{VCI}}(d^{\text{mono}}) + \mathcal{L}_{\text{VCI}}(d^{\text{mv}}) + \mathcal{L}_{\text{VCI}}(\widetilde{d}). \tag{11}$$

Table 1: Quantitative comparison on KITTI and Make3D datasets. "Mode" indicates supervision type: M = Monocular, S = Stereo, MS = Monocular + Stereo. Best results are in **bold**.

| Dataset | Method | Size | Mode | Train | Test | RMSE↓ | RMSE log↓ | Sq Rel↓ | Abs Rel↓ | $\delta < 1.25$ ↑ | $\delta < 1.25^2$ ↑ | $\delta < 1.25^3$ ↑ |
|---|---|---|---|---|---|---|---|---|---|---|---|---|
| KITTI | Monodepth (Godard et al., 2017) | 512×256 | S | ✓ | ✓ | 5.927 | 0.247 | 1.344 | 0.148 | 0.803 | 0.922 | 0.964 |
| | 3Net (Poggi et al., 2018) | 512×256 | S | ✓ | ✓ | 5.888 | 0.208 | 1.201 | 0.119 | 0.844 | 0.941 | 0.978 |
| | Monodepth2 (Godard et al., 2019) | 640×192 | S | ✓ | ✓ | 4.960 | 0.208 | 0.873 | 0.109 | 0.864 | 0.948 | 0.975 |
| | BRNet (Han et al., 2022) | 640×192 | S | ✓ | ✓ | 4.716 | 0.197 | 0.792 | 0.103 | 0.876 | 0.954 | 0.978 |
| | Monodepth2 (Godard et al., 2019) | 640×192 | MS | ✓ | ✓ | 4.750 | 0.196 | 0.818 | 0.106 | 0.874 | 0.957 | 0.979 |
| | DepthHints (Watson et al., 2019) | 640×192 | MS | ✓ | ✓ | 4.627 | 0.189 | 0.769 | 0.105 | 0.875 | 0.959 | 0.982 |
| | HR-Depth (Lyu et al., 2021) | 640×192 | MS | ✓ | ✓ | 4.612 | 0.185 | 0.785 | 0.107 | 0.887 | 0.962 | 0.982 |
| | R-MSFM6 (Zhou et al., 2021b) | 640×192 | MS | ✓ | ✓ | 4.625 | 0.189 | 0.787 | 0.111 | 0.882 | 0.961 | 0.981 |
| | SQLdepth (Wang et al., 2024b) | 640×192 | MS | ✓ | ✓ | 4.175 | 0.167 | 0.697 | 0.088 | 0.919 | 0.969 | 0.984 |
| | SPIdepth (Lavreniuk & Lavreniuk, 2025) | 640×192 | MS | ✓ | ✓ | 4.211 | 0.167 | 0.691 | 0.091 | 0.911 | 0.965 | 0.984 |
| | Monodepth2 (Godard et al., 2019) | 640×192 | M | ✓ | ✓ | 4.863 | 0.193 | 0.903 | 0.115 | 0.877 | 0.959 | 0.971 |
| | HR-Depth (Lyu et al., 2021) | 640×192 | M | ✓ | ✓ | 4.632 | 0.185 | 0.792 | 0.109 | 0.884 | 0.962 | 0.983 |
| | CADepth-Net (Yan et al., 2021) | 640×192 | M | ✓ | ✓ | 4.535 | 0.181 | 0.769 | 0.105 | 0.892 | 0.964 | 0.983 |
| | DIFFNet (Zhou et al., 2021a) | 640×192 | M | ✓ | ✓ | 4.483 | 0.180 | 0.768 | 0.104 | 0.896 | 0.965 | 0.983 |
| | MonoFormer (Bae et al., 2023) | 640×192 | M | ✓ | ✓ | 4.580 | 0.183 | 0.846 | 0.104 | 0.891 | 0.962 | 0.982 |
| | SC-DepthV3 (Sun et al., 2023) | 640×192 | M | ✓ | ✓ | 4.709 | 0.186 | 0.756 | 0.118 | 0.864 | 0.960 | 0.981 |
| | SRD-Depth (Liu et al., 2023) | 640×192 | M | ✓ | ✓ | 4.619 | 0.186 | 0.762 | 0.111 | 0.877 | 0.961 | 0.982 |
| | Swin-Depth (Shim & Kim, 2023) | 640×192 | M | ✓ | ✓ | 4.510 | 0.182 | 0.739 | 0.106 | 0.890 | 0.964 | 0.984 |
| | Lite-Mono (Zhang et al., 2023) | 640×192 | M | ✓ | ✓ | 4.561 | 0.182 | 0.750 | 0.107 | 0.892 | 0.964 | 0.983 |
| | ShuffleMono (Feng et al., 2024) | 640×192 | M | ✓ | ✓ | 4.821 | 0.193 | 0.850 | 0.114 | 0.872 | 0.957 | 0.980 |
| | Liu *et al.* (Liu et al., 2024) | 640×192 | M | ✓ | ✓ | 4.724 | 0.187 | 0.747 | 0.114 | 0.868 | 0.960 | 0.984 |
| | Dynamo-Depth (Sun & Hariharan, 2024) | 640×192 | M | ✓ | ✓ | 4.505 | 0.183 | 0.758 | 0.112 | 0.873 | 0.959 | 0.984 |
| | GeoDepth (Wu et al., 2025) | 640×192 | M | ✓ | ✓ | 4.381 | 0.176 | 0.694 | 0.100 | 0.897 | 0.966 | 0.984 |
| | EDS-Depth (Yu et al., 2025b) | 640×192 | M | ✓ | ✓ | 4.184 | 0.170 | 0.619 | 0.095 | 0.905 | 0.969 | **0.985** |
| | Manydepth2 (Zhou et al., 2025) | 640×192 | M | ✓ | ✓ | 4.232 | 0.170 | 0.649 | 0.091 | 0.909 | 0.968 | 0.984 |
| | **Ours** | 640×192 | M | ✓ | ✓ | **4.079** | **0.162** | **0.597** | **0.082** | **0.921** | **0.970** | **0.985** |
| Make3D | Monodepth2 (Godard et al., 2019) | 640×192 | M | × | ✓ | 7.418 | 0.163 | 3.589 | 0.322 | - | - | - |
| | HR-Depth (Lyu et al., 2021) | 640×192 | M | × | ✓ | 7.024 | 0.159 | 3.208 | 0.315 | - | - | - |
| | CADepth-Net (Yan et al., 2021) | 640×192 | M | × | ✓ | 7.066 | 0.159 | 3.086 | 0.312 | - | - | - |
| | DIFFNet (Zhou et al., 2021a) | 640×192 | M | × | ✓ | 7.008 | 0.155 | 3.313 | 0.309 | - | - | - |
| | Lite-Mono (Zhang et al., 2023) | 640×192 | M | × | ✓ | 6.981 | 0.158 | 3.060 | 0.305 | - | - | - |
| | Zhao *et al.* (Zhao et al., 2024) | 640×192 | M | × | ✓ | 7.095 | 0.158 | 3.200 | 0.316 | - | - | - |
| | Xiong *et al.* (Xiong et al., 2024) | 640×192 | M | × | ✓ | 7.005 | 0.161 | 3.102 | 0.319 | - | - | - |
| | GeoDepth (Wu et al., 2025) | 640×192 | M | × | ✓ | 6.735 | 0.153 | 2.750 | 0.296 | - | - | - |
| | Manydepth2 (Zhou et al., 2025) | 640×192 | M | × | ✓ | 6.643 | 0.151 | 2.604 | 0.291 | - | - | - |
| | EDS-Depth (Yu et al., 2025b) | 640×192 | M | × | ✓ | 6.714 | 0.152 | 2.419 | 0.299 | - | - | - |
| | SQLdepth (Wang et al., 2024b) | 640×192 | M | × | ✓ | 6.856 | 0.151 | 2.402 | 0.306 | - | - | - |
| | SPIdepth (Lavreniuk & Lavreniuk, 2025) | 640×192 | M | × | ✓ | 6.672 | 0.144 | 1.931 | 0.299 | - | - | - |
| | **Ours** | 640×192 | M | × | ✓ | **6.302** | **0.135** | **1.758** | **0.283** | - | - | - |

## 4 EXPERIMENTS

### 4.1 SETUPS

**Datasets.** We conduct experiments on both outdoor and indoor benchmarks. For outdoor scenes, we use KITTI (Geiger et al., 2013) (Eigen split (Eigen et al., 2014)) with 39,180 training, 4,424 validation, and 697 test triplets, and Make3D (Saxena et al., 2009) with 134 images for cross-dataset generalization. All outdoor images are resized to $640 \times 192$. For indoor scenes, NYUv2 (Gupta et al., 2010) has 335 training/validation sequences and 654 test images, while ScanNet (Dai et al., 2017) uses the StructDepth split (Li et al., 2021) with 533 test images. Indoor images are resized to $320 \times 256$. Additional evaluations on BONN RGB-D (Palazzolo et al., 2019), TUM-Dynamic (Sturm et al., 2012), Cityscapes (Cordts et al., 2016), and DrivingStereo (Yang et al., 2019) are provided in Appendix A.9 and A.10 to assess robustness in strongly dynamic scenes and generalization to additional driving domains. This suite covers diverse structures and dynamics for rigorous cross-domain evaluation.

**Evaluation Metrics.** We use seven standard metrics covering error and accuracy. Error metrics (Abs Rel, Sq Rel, RMSE, RMSE log) quantify the magnitude of prediction errors and are reported lower-is-better, while accuracy metrics ($\delta < 1.25$, $\delta < 1.25^2$, $\delta < 1.25^3$) measure the proportion of predictions within each $\delta$-based threshold and are reported higher-is-better.

**Implementation Details.** All models are implemented in PyTorch and trained on a single NVIDIA RTX A5000 GPU. We follow the standard Monodepth2 ResNet-based encoder–decoder backbone pretrained on ImageNet (Godard et al., 2019). We use Adam (Adam et al., 2014) with an initial learning rate of $1 \times 10^{-4}$, decayed by $10\times$ at epoch 15 on KITTI and epoch 5 on NYUv2. Training runs for 20 epochs on KITTI and 15 on NYUv2 with a batch size of 12. Hyperparameters are set to $\vartheta = 0.8$, $\gamma = 0.1$, $\varepsilon = 0.05$, $\zeta_\ell = 0.5$, $\eta = 0.5$, $\delta_s = 10^{-8}$, $\delta_e = 10^{-8}$, and $\lambda = 8.0$.

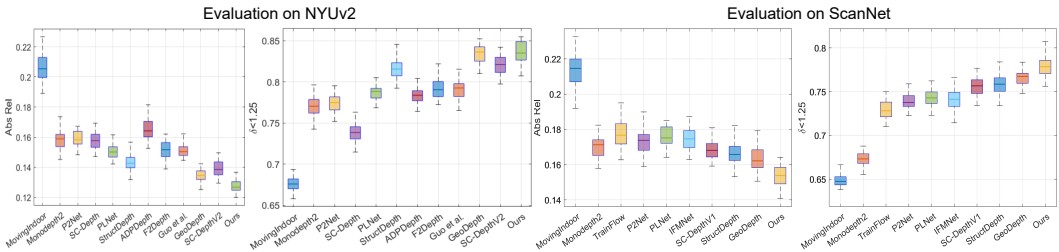

Figure 6: Box-plot comparison on NYUv2 and ScanNet datasets.

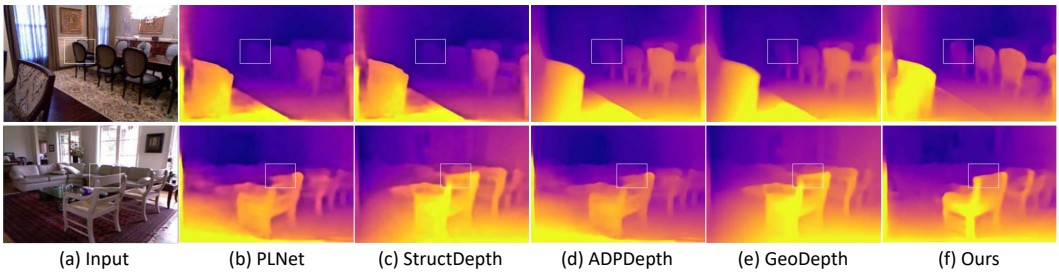

(a) Input     (b) PLNet     (c) StructDepth     (d) ADPDepth     (e) GeoDepth     (f) Ours

Figure 7: Qualitative results on the NYUv2 dataset. Our ROSS-Net outputs sharper structural details and more reliable depth in cluttered indoor regions compared to prior methods.

## 4.2 COMPARISON ON OUTDOOR SCENE

**Comparison on KITTI.** Table 1 reports results on KITTI with the standard evaluation crop and an 80 m cap. ROSS-Net outperforms prior methods across all metrics. Compared with the best prior results, it reduces RMSE, RMSE log, Sq Rel, and Abs Rel by $2.30\%$, $2.99\%$, $3.55\%$, and $6.82\%$, respectively, and improves $\delta < 1.25$ and $\delta < 1.25^2$ by $0.22\%$ and $0.10\%$, while matching the best on $\delta < 1.25^3$. Notably, even when trained solely on monocular video, ROSS-Net rivals or surpasses stereo- or hybrid-supervised methods. The strongest gains in Abs Rel and Sq Rel demonstrate that our framework effectively suppresses large errors from motion and occlusion. This validates our multi-level defense, where MAF, SDC, and VCI jointly target cost-volume noise, spectral bias, and outlier residuals. Qualitative results in Fig. 5 corroborate these trends, showing cleaner planar regions and sharper, better-aligned boundaries under motion and occlusion.

**Generalization to Make3D.** We evaluate zero-shot transfer by training on KITTI and testing on Make3D without any dataset-specific tuning. As reported in Table 1, ROSS-Net delivers state-of-the-art results on all metrics. These results suggest that the reliability-oriented mechanisms learned on KITTI transfer well to scenes with different camera intrinsics, larger sky regions, and shifted depth and illumination statistics. The consistent reduction of both error metrics and the improved $\delta$ thresholds show that the gains are not tied to a specific driving domain, but reflect a domain-robust improvement in geometric reliability. Qualitative comparisons in Fig. 13 (see Appendix A.10) demonstrate preserved scene geometry and boundary sharpness under large domain shifts.

## 4.3 COMPARISON ON INDOOR SCENE

**Comparison on NYUv2.** We follow the standard NYUv2 protocol with the official evaluation crop and a 10 m depth cap, using single-scale inference without post-processing. As shown in Fig. 6, the box plots indicate that ROSS-Net achieves the lowest median Abs Rel and the highest median $\delta < 1.25$ across test samples, indicating consistent gains over baselines. These improvements stem from our robustness-driven design: MAF removes unstable correspondences before cost-volume building, while SDC rebalances ambiguous spectra to curb over-smoothing, jointly strengthening geometric supervision. Qualitative results in Fig. 7 show more faithful geometric structures and finer high-frequency details, indicating improved consistency in challenging indoor layouts.

Table 2: Ablation study of our ROSS-Net on the KITTI dataset. Best results are in **bold**.

| Method | RMSE log ↓ | Abs Rel ↓ | $\delta < 1.25$ ↑ | $\delta < 1.25^2$ ↑ | $\delta < 1.25^3$ ↑ |
|---|---|---|---|---|---|
| Baseline (Xiang et al., 2023) | 0.176 | 0.102 | 0.898 | 0.965 | 0.984 |
| Baseline + MAF | 0.169 | 0.093 | 0.913 | 0.969 | 0.984 |
| Baseline + SDC | 0.170 | 0.096 | 0.910 | **0.970** | 0.984 |
| Baseline + VCI | 0.170 | 0.094 | 0.903 | 0.968 | 0.984 |
| Baseline + MAF + VCI | 0.167 | 0.086 | 0.914 | **0.970** | **0.985** |
| Baseline + SDC + VCI | 0.168 | 0.088 | 0.912 | **0.970** | **0.985** |
| **Ours** | **0.162** | **0.082** | **0.921** | **0.970** | **0.985** |

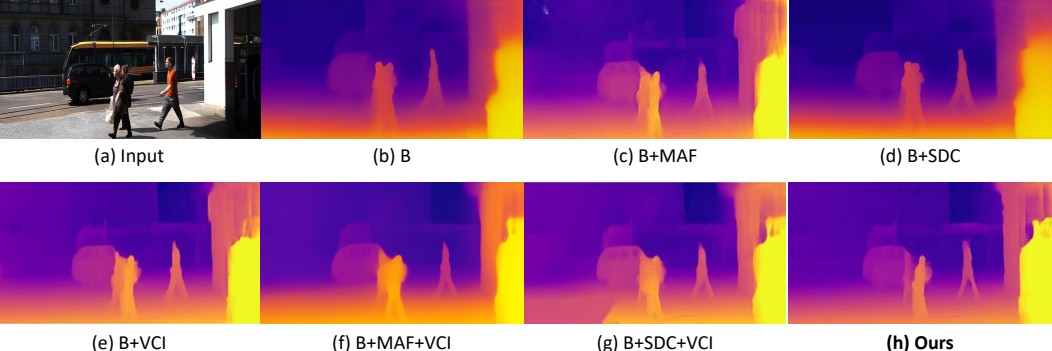

(a) Input  (b) B  (c) B+MAF  (d) B+SDC

(e) B+VCI  (f) B+MAF+VCI  (g) B+SDC+VCI  **(h) Ours**

Figure 8: Ablation of components on the KITTI dataset.

**Generalization to ScanNet.** To evaluate cross-dataset robustness, we test NYUv2-trained models on the unseen ScanNet dataset. Fig. 6 shows that ROSS-Net consistently outperforms the baselines across all metrics, positioning at favorable extremes of the box plots. This suggests that the NYUv2-learned representations generalize well to diverse indoor scenes, reflecting domain-agnostic handling of unreliable pixels rather than dataset-specific tuning. Qualitative comparisons in Fig. 14 (see Appendix A.10) highlight structural integrity and fewer boundary artifacts on unseen indoor scenes.

## 4.4 ABLATION STUDY

**Effectiveness of MAF.** As shown in Table 2, adding MAF to the baseline lowers RMSE log and Abs Rel by 3.98% and 8.82%, and improves $\delta < 1.25$ and $\delta < 1.25^2$ by 1.67% and 0.41%, while remaining comparable on $\delta < 1.25^3$. This indicates that enforcing spatio-temporal epipolar constraints suppresses dynamic mismatches and stabilizes cross-frame correspondences. When combined with VCI, MAF delivers further gains, suggesting that residual reweighting benefits from temporally stabilized matches. Fig. 8 shows that MAF enhances coherence on thin structures and preserves sharp depth discontinuities.

**Effectiveness of SDC.** As presented in Table 2, introducing SDC reduces RMSE log and Abs Rel by 3.41% and 5.88%, and increases $\delta < 1.25$ and $\delta < 1.25^2$ by 1.34% and 0.52%, while maintaining performance on $\delta < 1.25^3$. These results show that frequency-aware uncertainty helps exploit informative high-entropy regions that are otherwise underused. The benefit is amplified when SDC is paired with VCI, which suppresses heavy-tailed residuals, and together they redirect learning from overly stable regions to structurally ambiguous areas. Fig. 8 shows more stable reconstructions over textureless or repetitive surfaces such as roads and facades.

**Effectiveness of VCI.** Relative to the baseline (Xiang et al., 2023), VCI yields lower RMSE log (3.41%) and Abs Rel (7.84%), and higher $\delta < 1.25$ (0.56%) and $\delta < 1.25^2$ (0.31%), while showing stable performance on $\delta < 1.25^3$ according to Table 2, confirming its effectiveness against heavy-tailed residuals from spurious matches. In combination with MAF or SDC, VCI brings larger gains than either alone, suggesting synergy between residual reweighting, temporal stabilization, and un-

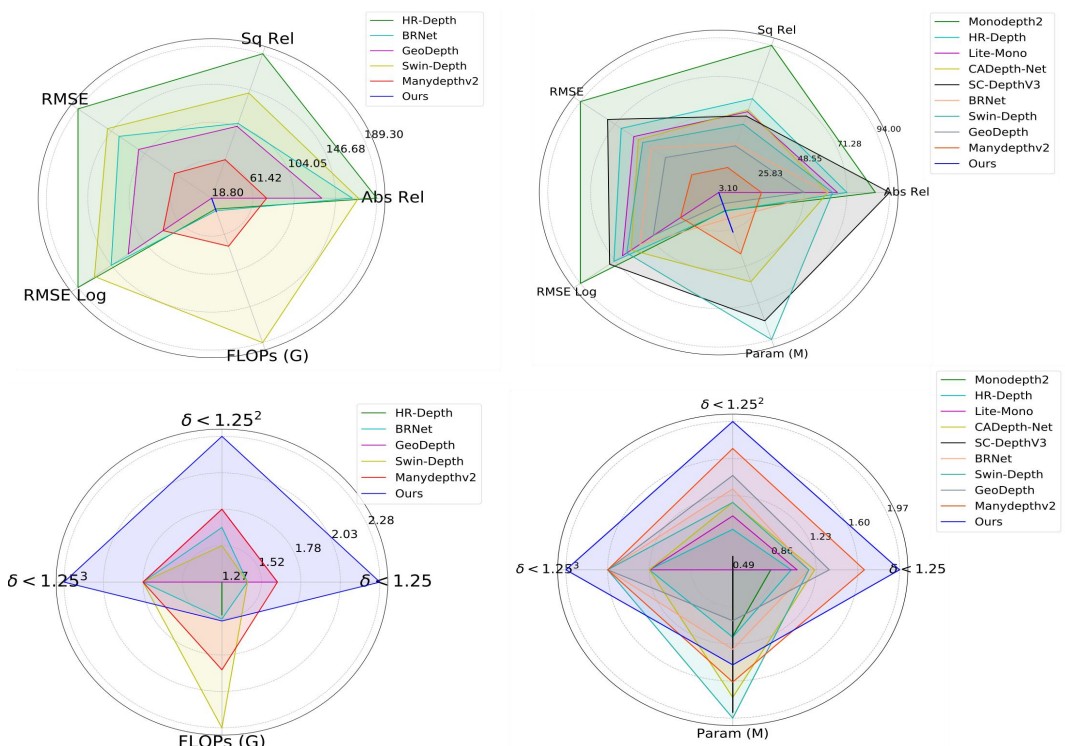

Figure 9: Complexity–performance analysis of different methods.

certainty calibration. Fig. 8 shows that VCI reduces artifacts in dynamic regions such as moving pedestrians and vehicles, improving robustness to outliers and occlusions.

## 4.5 COMPLEXITY ANALYSIS

To better understand the efficiency–accuracy trade-offs among different methods, we compare the model complexity of several state-of-the-art approaches and our ROSS-Net in terms of FLOPs and parameter counts, alongside standard depth estimation metrics. Fig. 9 provides a radar-plot visualization of this comparison. The two subplots on the left plot FLOPs against error and accuracy metrics, while the two on the right present the same metrics with respect to parameter counts. ROSS-Net achieves superior accuracy and lower error while maintaining moderate computational complexity. It consistently outperforms models of lower complexity (*e.g.*, Lite-Mono, GeoDepth) at a similar computational scale, and often surpasses more complex models (*e.g.*, Swin-Depth, ManyDepthv2).

## 5 CONCLUSION

In this paper, we propose ROSS-Net, a self-supervised framework for monocular depth in dynamic scenes that reframes robustness as a synergy across representation, optimization, and supervision. MAF establishes correspondence consistency across appearance, feature, and temporal attributes, mitigating motion-induced mismatches while preserving informative dynamics; SDC corrects low-frequency optimization bias via spectral calibration without inference-time overhead; and VCI bounds heavy-tailed residuals through trimmed reweighting, yielding reliable supervision from noisy reprojections. Extensive experiments demonstrate state-of-the-art performance on KITTI and NYUv2, with strong generalization to Make3D and ScanNet. Future work will target extreme photometric and temporal distortions and broaden deployment to multi-task and multi-sensor settings under tight runtime budgets.

# 6 Ethics Statements

This work focuses on self-supervised monocular depth estimation and does not involve human subjects, personal data, or sensitive attributes. All experiments are conducted on publicly available datasets under their respective licenses. We are not aware of any direct ethical risks, though potential misuse in safety-critical applications (e.g., autonomous driving) should be accompanied by rigorous validation and safeguards.

# 7 Reproducibility statement

We have taken several measures to ensure reproducibility. All implementation details, including network architectures, loss functions, training schedules, optimizer settings, and evaluation metrics, are fully specified in the main paper and appendix. Experiments are conducted on publicly available datasets (KITTI, Make3D, NYUv2, ScanNet) under their respective licenses, with preprocessing steps and data splits explicitly described. Our method is implemented in PyTorch, and training/testing was performed on NVIDIA GPUs. We will release the complete source code, pretrained models, and configuration scripts to enable exact replication of our results and facilitate further research.

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

# A  APPENDIX

## A.1  DEFINITION OF $\mathcal{L}_{\mathrm{ph}}$ AND $\mathcal{L}_s$

The photometric reconstruction loss $\mathcal{L}_{\mathrm{ph}}$ is defined over all source frames $\mathcal{S}_t = \{I_s\}$ relative to the target $I_t$. Given predicted depth $D_t$ for $I_t$, each source is warped into the target frame:

$$\hat{I}_{s \to t}(u,v) = I_s \langle \mathrm{proj}(D_t(u,v), K, T_{t \to s}) \rangle, \tag{12}$$

where $\mathrm{proj}(\cdot)$ maps target-view depths $D_t$ to source-view pixel coordinates, $K$ is the camera intrinsic matrix, $T_{t \to s}$ is the relative camera pose from target to source, and $\langle \cdot \rangle$ denotes the sampling operator. The photometric loss then combines SSIM and $\ell_1$ distance:

$$\mathcal{L}_{\mathrm{ph}}(u,v; D_t) = \sum_{s \in \mathcal{S}_t} \sum_{(u,v) \in \Omega} \alpha \frac{1 - \mathrm{SSIM}\big(I_t(u,v), \hat{I}_{s \to t}(u,v)\big)}{2} + (1-\alpha) \big\| I_t(u,v) - \hat{I}_{s \to t}(u,v) \big\|_1, \tag{13}$$

where $\alpha = 0.85$, following the default setting (Godard et al., 2019).

The smoothness $\mathcal{L}_s$ is defined as an edge-aware term:

$$\mathcal{L}_s(d_t) = \sum_{u,v} \Big( |\partial_x d_t^*| e^{-|\partial_x I_t|} + |\partial_y d_t^*| e^{-|\partial_y I_t|} \Big), \tag{14}$$

where $d_t^* = \frac{d_t}{\overline{d_t}}$, with $\overline{d_t}$ the mean depth value.

## A.2  PROBLEM DESCRIPTION: BREAKDOWN OF GEOMETRIC CONSISTENCY IN DYNAMIC ENVIRONMENTS

Self-supervised monocular depth estimation assumes that adjacent frames are related by rigid geometry. Given the target frame $I_t$ with predicted depth $D_t$, the projection of a pixel $p_t = (u,v,1)^\top$ into a source frame $I_s$ can be modeled as:

$$p_s \sim K\, T_{t \to s}\, D_t(u,v)\, K^{-1} p_t. \tag{15}$$

The corresponding source pixel intensity is used to reconstruct the target view:

$$\hat{I}_{s \to t}(u,v) = I_s \langle p_s \rangle, \tag{16}$$

and under the static-scene assumption, photometric consistency should ideally hold:

$$\mathcal{R}(u,v) = \big\| I_t(u,v) - \hat{I}_{s \to t}(u,v) \big\| \approx 0. \tag{17}$$

**Dynamic violation.** In dynamic regions, the true 3D point $X_t$ at $p_t$ moves independently of the camera and should follow:

$$p_s^{\mathrm{dyn}} \sim K\, (T_{t \to s}\, \Delta T_{t \to s})\, D_t(u,v)\, K^{-1} p_t, \tag{18}$$

where $\Delta T_{t \to s}$ denotes the object-specific motion. As this motion is ignored during training, the warping uses $p_s$ instead of the true correspondence $p_s^{\mathrm{dyn}}$, causing geometric misalignment and large residuals even when the predicted depth is correct:

$$\mathcal{R}(u,v) = \big\| I_t(u,v) - I_s \langle p_s \rangle \big\| \quad \text{is large even if } D_t(u,v) = D_t^\star(u,v), \tag{19}$$

as the true correspondence lies at $p_s^{\mathrm{dyn}}$, which differs from $p_s$.

**Consequences.** These large, misaligned residuals exhibit heavy-tailed distributions and undermine the validity of the photometric objective:

$$\mathcal{L}_{\mathrm{ph}} = \tfrac{1}{N} \sum_{(u,v)} \mathcal{R}(u, v). \tag{20}$$

Dynamic outliers dominate the summation, while reliable residuals from static regions are numerically suppressed. As a result, the loss no longer reflects geometric correctness—gradients are driven by misaligned dynamic pixels rather than by accurate correspondences. This biases the optimization toward stable background areas and causes the network to ignore ambiguous yet informative dynamic cues.

## A.3 ARCHITECTURES OF $\phi(\cdot)$ AND $\varphi(\cdot)$

As shown in Fig. 10, both $\phi(\cdot)$ and $\varphi(\cdot)$ adopt a lightweight residual convolutional architecture consisting of five cascaded blocks, each containing a $3 \times 3$ convolution followed by batch normalization (BN) and ReLU. For stability, the input of each of the last three blocks is element-wise added to its output as a residual connection. The two modules differ only in their channel dimensions:

**Spectral encoder $\phi(\cdot)$.** It operates on the spectral magnitude tensor $\mathbf{\Psi} \in \mathbb{R}^{B \times M \times H \times W}$, where $M$ is the number of frequency bins (typically $M \approx D/2$). All intermediate and output channels are set to $M$.

**Regression head $\varphi(\cdot)$.** It operates on the entropy map $\mathcal{E}_{\mathrm{SDC}} \in \mathbb{R}^{B \times 1 \times H \times W}$, with all intermediate layers using 1 channel and the final output being a single-channel uncertainty map $\mathcal{U} \in \mathbb{R}^{B \times 1 \times H \times W}$.

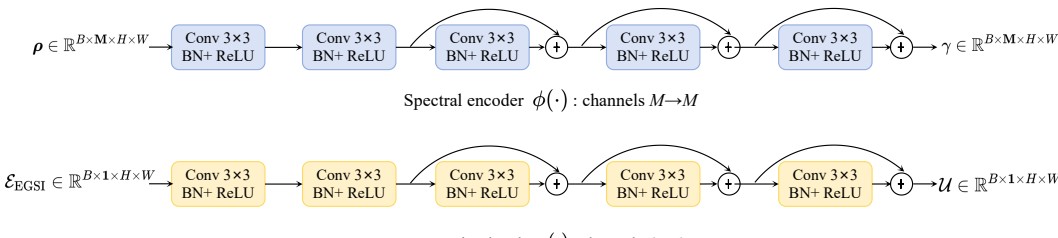

Figure 10: Architectures of $\phi(\cdot)$ and $\varphi(\cdot)$. Both modules consist of five cascaded Conv–BN–ReLU blocks with residual connections on the last three blocks, differing only in channel dimensionality ($M$ vs. 1).

## A.4 KEY COMPONENTS OF VCI

The overall pipeline of VCI is shown in Fig. 11. We detail the noisy-OR rule below.

### A.4.1 NOISY-OR CONSENSUS

Given credibility scores $W_{b,c}(u, v)$ from each of the $C$ source views, VCI aggregates them into a unified consensus field via a noisy-OR rule:

$$r_b(u, v) = 1 - \prod_{c=1}^{C} \big(1 - W_{b,c}(u, v)\big). \tag{21}$$

This rule can be derived from the probability that at least one source view supports a pixel: if each view independently marks it as valid with probability $W_{b,c}$, then the probability of being invalid in all views is $\prod_c (1 - W_{b,c})$. Taking one minus this value yields the probability that the pixel is supported by any view.

Figure 11: **Illustration of VCI.** It trims heavy-tailed residuals, maps them to soft credibility, and fuses per-view evidence via a noisy-OR consensus to weight the photometric loss with calibrated supervision.

Unlike averaging or minimum pooling, which can be dominated by noisy sources or overly suppress valid pixels, noisy-OR preserves partially consistent regions (e.g., one-sided occlusions) while strongly down-weighting pixels rejected by all views. This soft consensus avoids brittle all-or-nothing masking and retains more informative supervision.

**Differentiability and stability.** We clamp confidences to $[\epsilon, 1 - \epsilon]$ (e.g., $\epsilon = 10^{-6}$) to prevent underflow in $\prod_c(1 - W_{b,c})$. The aggregation is smooth and admits analytical gradients:

$$\frac{\partial r_b(u, v)}{\partial W_{b,k}(u, v)} = \prod_{c \neq k} \big(1 - W_{b,c}(u, v)\big). \tag{22}$$

## A.5 DEPTH GENERATION DETAILS

This section describes the generation of the two depth signals used in our framework: the monocular depth $d^{\mathrm{mono}}$ and the multi-view depth $d^{\mathrm{mv}}$.

**Monocular depth $d^{\mathbf{mono}}$.** We obtain $d^{\mathrm{mono}}$ from a pretrained MonoDepth2 (Godard et al., 2019) model. MonoDepth2 is trained in a self-supervised manner on monocular video using cross-frame photometric consistency (with jointly estimated camera poses), which provides strong single-image depth priors but remains subject to scale ambiguity and temporal inconsistency across frames.

**Multi-view depth $d^{\mathbf{mv}}$.** The multi-view estimate $d^{\mathrm{mv}}$ is produced by our network from a MAF-gated cost volume (Sec. 3.2). For each scale $\ell$, we compute the gated cost volume $\mathcal{C}^{(\ell)}(u', v', d)$ between target and source features $\mathbf{F}_t'^{(\ell)}$ and $\mathbf{F}_{t-1}'^{(\ell)}$. A local-max refinement operator extracts coarse depth candidates by selecting the most likely depth index and applying local quadratic fitting for sub-pixel precision:

$$d_\ell^*(u', v') = \arg\max_d \mathcal{C}^{(\ell)}(u', v', d), \qquad \widehat{d}_\ell = \mathrm{QuadFit}\big(d_\ell^*, \mathcal{C}^{(\ell)}\big). \tag{23}$$

We then perform coarse-to-fine guided refinement, progressively upsampling $\widehat{d}_\ell$ and aligning it with higher-resolution image features $\mathbf{F}_t^{(\ell-1)}$ to recover spatial details and boundary structures:

$$d^{\ell-1} = \mathrm{Refine}\big(\mathrm{Upsample}(\widehat{d}_\ell), \mathbf{F}_t^{(\ell-1)}\big), \qquad \ell = L, \ldots, 1. \tag{24}$$

After iterating through all scales, the full-resolution multi-view depth is obtained as: $d^{\mathrm{mv}} = d^0$.

This procedure yields a sharp and geometrically consistent depth estimate, while MAF suppresses dynamic mismatches that would otherwise contaminate the cost volume.

## A.6 Algorithmic Details and Pseudocode for ROSS-Net

This section specifies the exact procedures used to train and evaluate ROSS-Net. **Algorithm 1** presents the end-to-end training loop, while **Algorithm 2** summarizes the inference procedure. Unless noted otherwise, the notation and symbols follow Sec. 3. For completeness, we also provide standalone pseudocode for each major component: MAF is detailed in **Algorithm 3**, SDC is detailed in **Algorithm 4**, and VCI is detailed in **Algorithm 5**. Both SDC (uncertainty-guided fusion) and VCI (consensus supervision) are training-only and introduce no inference-time overhead.

---

**Algorithm 1** Training ROSS-Net with MAF, SDC, and VCI

---

**Require:** Monocular video $\{I_t\}_{t=0}^{T-1}$ with intrinsics $K$; learning rate $\alpha$; thresholds $\varepsilon, \{\zeta_\ell\}$; decay $\eta$; VCI trims $(\kappa, \vartheta)$; sharpness $\lambda$; SDC regs $\delta_s, \delta_e$; weight $\gamma$; depth set $\mathcal{D}$

1: **Init:** initialize network parameters $(E, \text{Head}_{\text{mono}}, \text{Head}_{\text{mv}}, \phi, \varphi)$
2: **Init:** initialize MAF masks lazily — at $t = 1$ set $\mathcal{K}_{\ell,1} \leftarrow \mathcal{K}_\ell^{\text{cons}}$; for $t > 1$ update $\mathcal{K}_{\ell,t} \leftarrow \eta\,\mathcal{K}_{\ell,t-1} + (1 - \eta)\,\mathcal{K}_\ell^{\text{cons}}$
3: **for** each minibatch of time indices $\mathcal{B} \subset \{1, \ldots, T - 1\}$ **do**
4:     **for** each $t \in \mathcal{B}$ **do**
5:         **Feature pyramid:** $\{\mathbf{F}_t^{(\ell)}\} \leftarrow E(I_t)$,    $\{\mathbf{F}_{t-1}^{(\ell)}\} \leftarrow E(I_{t-1})$
6:         **MAF gating:** $\{\mathcal{K}_{\ell,t}\}, \{\mathbf{F}'^{(\ell)}_{\{t,t-1\}}\} \leftarrow \text{MAF}(I_{t-1}, I_t, \{\mathbf{F}_{t-1}^{(\ell)}\}, \{\mathbf{F}_t^{(\ell)}\}, \{\mathcal{K}_{\ell,t-1}\}; \varepsilon, \{\zeta_\ell\}, \eta)$
7:         **Cost volumes:** For each level $\ell$ and depth $d \in \mathcal{D}$,

$$\mathcal{C}^{(\ell)}(u', v', d) = \left\langle \mathbf{F}_t'^{(\ell)}(u', v'),\ \mathbf{F}_{t-1}'^{(\ell)}\big(\pi(u', v', d)\big) \right\rangle$$

8:         **Align & stack** $\{\mathcal{C}^{(\ell)}\}_\ell \to \mathcal{C}$
9:         **Predictions:** $d^{\text{mv}} \leftarrow \text{Head}_{\text{mv}}(\mathcal{C})$;    $d^{\text{mono}} \leftarrow \text{Head}_{\text{mono}}(I_t)$
10:        **SDC uncertainty fusion:** $\mathcal{U}, \widetilde{d} \leftarrow \text{SDC}(\mathcal{C}, d^{\text{mv}}, d^{\text{mono}}; \phi, \varphi, \delta_s, \delta_e)$
11:        **Photometric residual tensors:** build $\mathcal{R}$ for each candidate depth $\{d^{\text{mono}}, d^{\text{mv}}, \widetilde{d}\}$ via warping $I_{t-1} \to I_t$ with $\pi(\cdot)$
12:        **VCI consensus masks:** $R^{\text{mono}}, R^{\text{mv}}, R^{\text{fuse}} \leftarrow \text{VCI}(\mathcal{R}; \kappa, \vartheta, \lambda)$
13:        **Losses:**

$$\mathcal{L}_{\text{VCI}}(d) = \sum_{(u,v)} R(u, v)\,\mathcal{L}_{\text{ph}}(u, v; d)\ +\ \gamma\,\mathcal{L}_s(d)$$

$$\mathcal{L}_{\text{total}} = \mathcal{L}_{\text{VCI}}(d^{\text{mono}}) + \mathcal{L}_{\text{VCI}}(d^{\text{mv}}) + \mathcal{L}_{\text{VCI}}(\widetilde{d})$$

14:        **Update:** $\theta \leftarrow \theta - \alpha\,\nabla_\theta \mathcal{L}_{\text{total}}$ for all trainable params $\theta$ of $(E, \text{Head}_{\text{mono}}, \text{Head}_{\text{mv}}, \phi, \varphi)$
15:     **end for**
16: **end for**

---

**Algorithm 2** Inference (without SDC or VCI)

---

**Require:** Test frames $(I_{t-1}, I_t)$, intrinsics $K$

1: Extract pyramids $\{\mathbf{F}_{t-1}^{(\ell)}\}, \{\mathbf{F}_t^{(\ell)}\}$; run MAF to obtain gated features
2: Build stacked cost volume $\mathcal{C}$ and predict $d^{\text{mv}} = \text{Head}_{\text{mv}}(\mathcal{C})$
3: **Output:** $d^{\text{mv}}$                            $\triangleright$ Optionally also output $d^{\text{mono}}$ for analysis

---

## A.7 Dataset Details

Table 3 summarizes the eight datasets used in our experiments, including the number of training/testing samples, depth range, and key challenges. We further describe their characteristics and usage below to facilitate reproducibility and clarify their evaluation roles.

**KITTI.** KITTI (Geiger et al., 2013) is an outdoor driving dataset captured from a moving vehicle equipped with stereo cameras and a 64-beam Velodyne LiDAR. Scenes cover urban streets, residential areas, and rural roads with vehicles, pedestrians, cyclists, traffic signs, vegetation, and frequent moving occluders. Depth maps are sparse LiDAR projections, which often miss dynamic objects and exhibit discontinuities at far distances. The dataset contains long-range driving sequences with

---

**Algorithm 3** MAF

---

**Require:** input $I_t, I_{t-1}$; multi-scale features $\{F_t^l, F_{t-1}^l\}_{l=1}^L$; appearance threshold $\varepsilon$; per-level feature thresholds $\{\zeta_l\}_{l=1}^L$; EMA factor $\eta$

**Ensure:** stabilized masks $\mathcal{K}_{\ell,t}$ and refined cost volume $\mathcal{C}$

1: $\mathcal{K}^{\text{seed}}(u,v) \leftarrow \mathbb{I}\left[ \left\| I_t(u,v) - I_{t-1}(u,v) \right\|_1 > \varepsilon \right]$      ▷ Eq.(1)

2: **for** $\ell = 1$ to $L$ **do**

3:     $\mathcal{G}_\ell(u',v') \leftarrow \mathbb{I}\left[ \left\| \mathbf{F}_t^{(\ell)}(u',v') - \mathbf{F}_{t-1}^{(\ell)}(u',v') \right\|_2 > \zeta_\ell \right]$      ▷ Eq.(2)

4:     $\mathcal{K}_\ell^{\text{cons}}(u',v') \leftarrow \mathbb{I}\left[ \mathcal{K}_\ell^{\text{seed}}(u',v') = 1 \ \wedge \ \mathcal{G}_\ell(u',v') = 1 \right]$      ▷ Eq.(3)

5:     **if** $\mathcal{K}_{\ell,t-1}$ exists **then**

6:        $\mathcal{K}_{\ell,t} \leftarrow \eta \mathcal{K}_{\ell,t-1} + (1-\eta)\mathcal{K}_\ell^{\text{cons}}$      ▷ Eq.(4)

7:     **else**

8:        $\mathcal{K}_{\ell,t} \leftarrow \mathcal{K}_\ell^{\text{cons}}$

9:     **end if**

10:    $\mathbf{F'}_t^{(\ell)} \leftarrow \mathcal{K}_{\ell,t} \odot \mathbf{F}_t^{(\ell)}; \quad \mathbf{F'}_{t-1}^{(\ell)} \leftarrow \mathcal{K}_{\ell,t} \odot \mathbf{F}_{t-1}^{(\ell)}$      ▷ Eq.(5)

11:    $\mathcal{C}^{(\ell)} \leftarrow \text{CostVol}(\mathbf{F'}_t^{(\ell)}, \mathbf{F'}_{t-1}^{(\ell)})$      ▷ Eq.(6)

12: **end for**

13: $\mathcal{C} \leftarrow \text{StackAlign}(\{\mathcal{C}^{(\ell)}\}_{\ell=1}^L)$

14: **return** $\mathcal{K}_{\ell,t}, \mathcal{C}$

---

**Algorithm 4** SDC (train-only)

---

**Require:** refined cost volume $\mathcal{C}$; spectral encoder $\phi(\cdot)$; linear map $(A, b)$; uncertainty regressor $\varphi(\cdot)$; stabilizers $\delta_s, \delta_e$

**Ensure:** uncertainty map $\mathcal{U}$ and fused depth $\widetilde{d}$ (train-only)

1: $\hat{\mathcal{C}} \leftarrow \mathcal{N}_d\big(\sigma(\mathcal{C})\big)$

2: $\Psi_m \leftarrow |\mathcal{F}_d(\hat{\mathcal{C}})|_m / (|\mathcal{F}_d(\hat{\mathcal{C}})|_1 + \delta_s)$

3: $\mathcal{E}_{\text{SDC}} \leftarrow - \underbrace{\sigma\big(A\,\phi(\Psi_m) + b\big)}_{\text{confidence score } c_m} \odot \underbrace{\Big(\Psi_m \odot \ln(\Psi_m + \delta_e)\Big)}_{\text{raw entropy } \mathcal{E}_r}$      ▷ Eq.(7)

4: $\mathcal{U} \leftarrow \varphi(\mathcal{E}_{\text{SDC}})$

5: $\widetilde{d} \leftarrow d^{\text{mv}} + \mathcal{U}(d^{\text{mono}} - d^{\text{mv}})$      ▷ Eq.(8)

6: **return** $\mathcal{U}, \widetilde{d}$

---

frequent viewpoint changes and fast ego-motion, which amplify the impact of sparse supervision. As a result, KITTI poses challenges including dynamic traffic participants, large depth variation, strong lighting changes, and occlusion-induced correspondence breaks.

**Make3D.** Make3D (Saxena et al., 2009) contains outdoor images captured by a fixed monocular camera with dense depth from a laser scanner. Typical scenes include building facades, roads, lawns, walls, and distant sky regions with large depth variation and limited texture. Depth maps are dense but geometry-smoothed, and tend to saturate on far surfaces, especially in low-texture regions. The dataset consists of static viewpoints and wide open scenes that differ markedly from driving settings, which magnifies the domain gap from KITTI-trained models. As a result, Make3D poses challenges including extremely large depth ranges, weak textures, strong perspective distortion, and severe cross-dataset appearance shifts.

**NYUv2.** NYUv2 (Gupta et al., 2010) is an indoor RGB-D dataset collected using a Microsoft Kinect structured-light sensor. It covers diverse indoor environments such as offices, classrooms, kitchens, and living rooms containing furniture, walls, floors, and cluttered household objects. Depth maps are dense but exhibit noise at object boundaries and missing pixels on reflective or transparent surfaces. The dataset contains close-range handheld captures with strong viewpoint changes and frequent occlusions in cluttered spaces, which exacerbate alignment errors. As a result, NYUv2 poses challenges including cluttered layouts, low-texture planar surfaces, heavy occlusions, and severe depth noise at object boundaries.

---

**Algorithm 5** VCI (train-only)

---

**Require:** residuals $\{\mathcal{R}_{b,c}(u,v)\}$; trim ratio $\kappa$, quantile $\vartheta$; penalty scale $\lambda$; photometric loss $\mathcal{L}_{\text{ph}}$;
    smoothness $\mathcal{L}_s$; depths $\{D_b\} = \{d^{\text{mono}}, d^{\text{mv}}, \widetilde{d}\}$
**Ensure:** VCI-weighted loss $\mathcal{L}_{\text{total}}$ and consensus field $R$ (train-only)

1: **for** each batch $b$ and source view $c$ **do**
2:      $\mathcal{S}_{b,c} \leftarrow \{\mathcal{R}_{b,c}(u,v) \mid (u,v) \in \Omega\}$
3:      $\tau_{b,c} \leftarrow \text{TrimQuantile}_{\vartheta}^{(\kappa)}(\mathcal{S}_{b,c})$
4:      $W_{b,c}(u,v) \leftarrow \sigma(\lambda(\tau_{b,c} - \mathcal{R}_{b,c}(u,v)))$                $\triangleright$ Eq.(9)
5: **end for**
6: **for** each batch $b$ **do**
7:      $r_b(u,v) \leftarrow 1 - \prod_c \left(1 - W_{b,c}(u,v)\right)$
8: **end for**
9: $\mathcal{L}_{\text{VCI}}(D_b) \leftarrow \sum_b \sum_{(u,v) \in \Omega} r_b(u,v)\, \mathcal{L}_{\text{ph}}(u,v; D_b) + \gamma \mathcal{L}_s(D_b)$      $\triangleright$ Eq.(10)
10: $\mathcal{L}_{\text{total}} \leftarrow \mathcal{L}_{\text{VCI}}(d^{\text{mono}}) + \mathcal{L}_{\text{VCI}}(d^{\text{mv}}) + \mathcal{L}_{\text{VCI}}(\widetilde{d})$      $\triangleright$ Eq.(11)
11: **return** $\mathcal{L}_{\text{total}}, R$

---

Table 3: Summary of datasets used in our experiments.

| Dataset | #Train / #Test | Depth Range | Key Challenges |
|---|---|---|---|
| KITTI | 39,180 / 697 | 0–80m | Outdoor driving, dynamic traffic, occlusions |
| Make3D | — / 134 | 0–80m | Outdoor scenes, large depth range, sky regions |
| NYUv2 | 335 / 654 | 0–10m | Indoor RGB-D, clutter, low texture |
| ScanNet | — / 533 | 0–10m | Indoor large-scale, sensor noise, layout diversity |
| Cityscapes | — / 1,525 | — | Dense urban motion, domain shift |
| DrivingStereo | — / 2,000 | — | Weather diversity, occlusions, thin structures |
| BONN RGB-D | — / 1,785 | 0–10m | Non-rigid motion, heavy occlusions |
| TUM-Dynamic | — / 1,375 | — | Dynamic indoor RGB-D, unstable matches |

**ScanNet.** ScanNet (Dai et al., 2017) is a large-scale RGB-D dataset of indoor scenes reconstructed via TSDF fusion from handheld RGB-D camera streams. It spans diverse environments including corridors, meeting rooms, bathrooms, storage rooms, and exhibition halls with widely varying layouts and viewpoints. Depth maps suffer from accumulated fusion noise, sensor drift, and temporal misalignments during reconstruction. The dataset contains long video scans with continuous ego-motion and significant viewpoint drift, which amplify the impact of sensor noise and domain shifts. As a result, ScanNet poses challenges including large scene-scale variation, accumulated depth noise, inconsistent geometry, and strong cross-domain appearance gaps from NYUv2.

**Cityscapes.** Cityscapes (Cordts et al., 2016) is a large-scale outdoor urban driving dataset recorded from a vehicle across multiple cities. It provides high-resolution street-view imagery with rich dynamic content such as vehicles, pedestrians, and cyclists. As a result, Cityscapes poses challenges including crowded dynamic scenes, complex occlusion boundaries, strong domain gaps in appearance and geometry, and long-range depth ambiguity. It provides 69,730 training image triplets and 1,525 test images. In our cross-domain evaluation, we train the model on KITTI and directly test on the official Cityscapes test set without any fine-tuning, following standard zero-shot generalization practice.

**DrivingStereo.** DrivingStereo (Yang et al., 2019) is a large-scale real-world driving stereo dataset covering diverse road types and weather/lighting conditions. The scenes feature dense traffic participants and frequent dynamic motion, with many thin structures and high-frequency boundaries (e.g., poles, signs, bicycles), which introduce severe occlusion/disocclusion and cross-domain appearance shifts. To assess scalability and cross-domain generalization, we train on KITTI and directly test on the official DrivingStereo test set in a zero-shot manner. Following the standard protocol, the test set contains 500 images per weather condition (fog, cloudy, rainy, sunny). This setting evaluates whether the robustness learned on KITTI transfers to a larger and more diverse driving domain.

**BONN RGB-D.** BONN RGB-D Dynamic (Palazzolo et al., 2019) is an indoor RGB-D dataset specifically collected to evaluate robustness under strong non-rigid motion and occlusion. It contains

24 dynamic sequences with diverse moving objects (e.g., people and handheld items) that frequently violate the static-scene assumption. The dataset provides synchronized RGB images, depth maps, and ground-truth trajectories for each sequence. As a result, BONN RGB-D poses challenges including strong object motion, heavy occlusions, rapid appearance changes, and unreliable geometric evidence in dynamic areas. In our evaluation, we follow a challenging dynamic protocol and use 4 representative video sequences for testing, totaling 1,785 images, while all remaining sequences are used for training.

**TUM-Dynamic.** TUM-Dynamic is the dynamic subset of the TUM RGB-D benchmark (Sturm et al., 2012), captured with a Microsoft Kinect sensor. Typical sequences (e.g., desk-with-person, walking sequences) feature people moving near the camera or across the scene, leading to complex occlusion patterns and non-rigid shape changes. As a result, TUM-Dynamic poses challenges including non-rigid motion, persistent foreground dynamics, strong parallax ambiguities, and heavy-tailed photometric residuals. We intentionally select only sequences labeled with dynamic objects to ensure the model is trained in genuinely dynamic indoor scenes. In total, we use 11 dynamic sequences. Following a challenging split, the last two sequences (1,375 images) are reserved for testing, while the remaining nine sequences are used for training.

## A.8 METRIC DEFINITIONS

For completeness, we provide the formal definitions of the evaluation metrics used in our experiments. Let $d_i$ and $\hat{d}_i$ denote the ground-truth and predicted depth for pixel $i$, respectively, and $N$ the number of valid pixels.

**Error metrics.** These metrics measure the deviation between predictions and ground truth:

$$\text{Abs Rel} = \frac{1}{N} \sum_i \frac{|d_i - \hat{d}_i|}{d_i}, \tag{25}$$

$$\text{Sq Rel} = \frac{1}{N} \sum_i \frac{(d_i - \hat{d}_i)^2}{d_i}, \tag{26}$$

$$\text{RMSE} = \sqrt{\frac{1}{N} \sum_i (d_i - \hat{d}_i)^2}, \tag{27}$$

$$\text{RMSE log} = \sqrt{\frac{1}{N} \sum_i (\log d_i - \log \hat{d}_i)^2}. \tag{28}$$

Abs Rel and Sq Rel capture the mean absolute and squared relative errors, emphasizing proportional deviations from the ground truth. RMSE measures the overall scale of absolute errors, while RMSE log focuses on relative consistency by penalizing multiplicative discrepancies.

**Accuracy metrics.** These metrics measure the proportion of accurate predictions within error thresholds:

$$\delta < 1.25^k = \frac{1}{N} \sum_i \mathbb{I}\Big( \max\Big(\frac{d_i}{\hat{d}_i}, \frac{\hat{d}_i}{d_i}\Big) < 1.25^k \Big), \quad k \in \{1, 2, 3\}. \tag{29}$$

$\delta < 1.25^k$ denotes the percentage of pixels whose predicted-to-true depth ratio lies within a threshold of $1.25^k$. Higher values indicate more predictions falling within the acceptable relative error bounds.

These definitions follow the standard protocol introduced by Eigen et al. (Eigen et al., 2014) and widely adopted in subsequent works such as AdaBins (Bhat et al., 2021).

## A.9 ADDITIONAL QUANTITATIVE COMPARISONS

On outdoor scenes, Fig. 12 shows that ROSS-Net achieves favorable extremes on the KITTI benchmark, yielding lower error (Abs Rel) and higher accuracy ($\delta < 1.25$) than all baselines. The box plots further reveal a more favorable median, suggesting improved central tendency under diverse

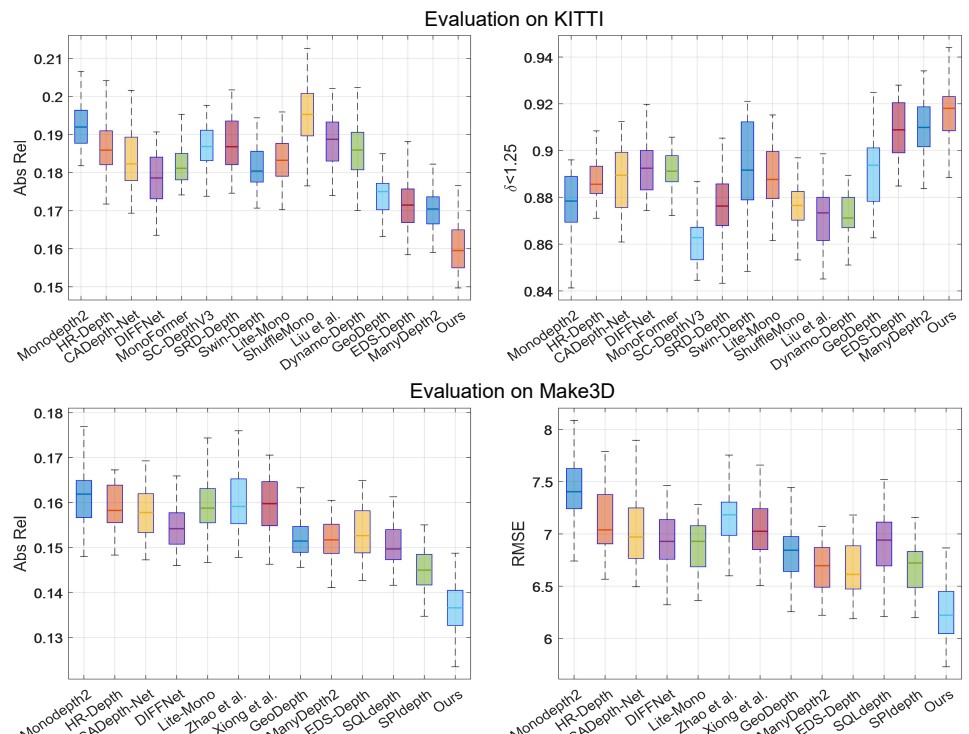

Figure 12: Box-plot comparison on KITTI and Make3D datasets.

driving conditions. These gains are driven by our robustness mechanisms: MAF stabilizes multi-frame correspondences before cost-volume construction, SDC counteracts over-smoothing under ambiguous likelihoods, and VCI downweights heavy-tailed residuals from motion and occlusions, jointly reducing large-error tails. To assess cross-domain transfer, we directly evaluate the KITTI-trained models on the Make3D dataset without fine-tuning. Fig. 12 demonstrates that ROSS-Net maintains competitive performance and consistently positions at advantageous ends of the distributions, indicating robust generalization from structured urban driving scenes to outdoor landscapes with distinct statistics. Further, we report evaluations on Cityscapes and DrivingStereo in Tables 4 and 5. The improvements on these driving benchmarks follow the same trend as KITTI, confirming that the robustness gains transfer across outdoor domains with different camera settings and motion patterns. These results underscore the model's ability to preserve structural fidelity and mitigate outlier influence even under severe domain shifts.

On indoor scenes, we report comprehensive quantitative results in Table 6. On NYUv2, ROSS-Net achieves the lowest RMSE and Abs Rel, reducing them by $4.42\%$ and $4.48\%$, and improves $\delta < 1.25$, $\delta < 1.25^2$, and $\delta < 1.25^3$ by $0.24\%$, $0.21\%$, and $0.61\%$. These consistent gains across both error- and accuracy-based metrics indicate that ROSS-Net not only reduces overall depth estimation errors but also increases the proportion of pixels reconstructed within tight thresholds. The zero-shot evaluation on ScanNet shows that the gains largely mirror those on NYUv2, suggesting that improved reliability is not tied to dataset-specific tuning but to domain-agnostic handling of unstable evidence. Further, we add dedicated dynamic indoor evaluations on BONN RGB-D and TUM-Dynamic in Table7. ROSS-Net remains consistently competitive and reduces dynamic outlier errors, matching our intent that MAF and VCI stabilize supervision under motion while SDC prevents low-frequency collapse in ambiguous regions. Such improvements demonstrate the model's ability to recover sharper structures and maintain reliability in cluttered indoor layouts, complementing distributional evidence provided by the box plots in the main text.

Table 4: Quantitative comparison on Cityscapes dataset. "Mode" indicates supervision type: M = Monocular. Best results are in **bold**.

| Method | Mode | Train | Test | RMSE↓ | RMSE log↓ | Sq Rel↓ | Abs Rel↓ | $\delta < 1.25$ ↑ | $\delta < 1.25^2$ ↑ | $\delta < 1.25^3$ ↑ |
|---|---|---|---|---|---|---|---|---|---|---|
| Monodepth2 (Godard et al., 2019) | M | × | ✓ | 8.590 | 0.234 | 1.785 | 0.153 | 0.774 | 0.926 | 0.976 |
| SD-SSMDE (Petrovai & Nedevschi, 2022) | M | × | ✓ | 8.441 | 0.221 | 1.635 | 0.143 | 0.789 | 0.931 | 0.980 |
| MonoPP (Elazab et al., 2025) | M | × | ✓ | 12.113 | - | 3.156 | 0.216 | 0.580 | - | - |
| RobustDepth (Saunders et al., 2023) | M | × | ✓ | 7.912 | 0.224 | 1.569 | 0.160 | 0.757 | 0.937 | 0.982 |
| Jasmine (Wang et al., 2025) | M | × | ✓ | 6.618 | - | - | 0.123 | 0.852 | - | - |
| **Ours** | M | × | ✓ | **5.437** | **0.155** | **0.882** | **0.103** | **0.881** | **0.971** | **0.992** |

Table 5: Quantitative comparison on DrivingStereo dataset. "Mode" indicates supervision type: M = Monocular. Best results are in **bold**.

| Domain | Method | Mode | Train | Test | RMSE↓ | RMSE log↓ | Sq Rel↓ | Abs Rel↓ | $\delta < 1.25$ ↑ | $\delta < 1.25^2$ ↑ | $\delta < 1.25^3$ ↑ |
|---|---|---|---|---|---|---|---|---|---|---|---|
| Foggy | Monodepth2 (Godard et al., 2019) | M | × | ✓ | 7.927 | 0.195 | 1.514 | 0.125 | 0.849 | 0.950 | 0.980 |
| | MonoViT (Zhao et al., 2022) | M | × | ✓ | 6.313 | 0.150 | 0.934 | **0.096** | 0.893 | 0.974 | **0.993** |
| | RobustDepth (Saunders et al., 2023) | M | × | ✓ | 9.098 | 0.203 | 1.907 | 0.140 | 0.827 | 0.949 | 0.980 |
| | D4RD (Wang et al., 2024a) | M | × | ✓ | 7.102 | 0.154 | 1.061 | 0.105 | 0.883 | 0.975 | 0.994 |
| | Yan *et al.* (Yan et al., 2025) | M | × | ✓ | 6.422 | - | 1.023 | 0.128 | 0.845 | - | - |
| | Jasmine (Wang et al., 2025) | M | × | ✓ | **5.702** | - | - | 0.098 | **0.902** | - | - |
| | **Ours** | M | × | ✓ | 6.427 | **0.148** | **0.930** | 0.102 | 0.891 | **0.976** | 0.991 |
| Cloudy | Monodepth2 (Godard et al., 2019) | M | × | ✓ | 6.976 | 0.209 | 1.900 | 0.155 | 0.813 | 0.943 | 0.979 |
| | MonoViT (Zhao et al., 2022) | M | × | ✓ | 5.970 | 0.177 | 1.300 | 0.125 | 0.861 | 0.958 | 0.986 |
| | RobustDepth (Saunders et al., 2023) | M | × | ✓ | 8.269 | 0.231 | 2.281 | 0.173 | 0.782 | 0.933 | 0.973 |
| | D4RD (Wang et al., 2024a) | M | × | ✓ | 7.271 | 0.198 | 1.560 | 0.141 | 0.830 | 0.948 | 0.983 |
| | Jasmine (Wang et al., 2025) | M | × | ✓ | **5.651** | - | - | 0.133 | 0.849 | - | - |
| | **Ours** | M | × | ✓ | 5.833 | **0.169** | **1.224** | **0.118** | **0.865** | **0.961** | **0.987** |
| Rainy | Monodepth2 (Godard et al., 2019) | M | × | ✓ | 11.040 | 0.301 | 3.339 | 0.240 | 0.591 | 0.857 | 0.952 |
| | MonoViT (Zhao et al., 2022) | M | × | ✓ | 8.604 | 0.219 | 1.925 | 0.169 | 0.733 | 0.934 | 0.985 |
| | RobustDepth (Saunders et al., 2023) | M | × | ✓ | 10.595 | 0.260 | 2.670 | 0.199 | 0.677 | 0.902 | 0.972 |
| | D4RD (Wang et al., 2024a) | M | × | ✓ | 8.584 | 0.208 | **1.722** | 0.158 | 0.773 | 0.946 | 0.985 |
| | Yan *et al.* (Yan et al., 2025) | M | × | ✓ | 8.004 | - | 1.755 | 0.171 | 0.730 | - | - |
| | Jasmine (Wang et al., 2025) | M | × | ✓ | **7.194** | - | - | 0.160 | **0.787** | - | - |
| | **Ours** | M | × | ✓ | 8.293 | **0.197** | 1.816 | **0.157** | 0.752 | **0.950** | **0.986** |
| Sunny | Monodepth2 (Godard et al., 2019) | M | × | ✓ | 6.744 | 0.214 | 1.74 | 0.155 | 0.819 | 0.941 | 0.977 |
| | MonoViT (Zhao et al., 2022) | M | × | ✓ | 6.109 | 0.186 | 1.266 | 0.130 | 0.851 | 0.956 | 0.985 |
| | RobustDepth (Saunders et al., 2023) | M | × | ✓ | 8.084 | 0.246 | 2.174 | 0.185 | 0.765 | 0.919 | 0.965 |
| | D4RD (Wang et al., 2024a) | M | × | ✓ | 7.121 | 0.207 | 1.437 | 0.149 | 0.815 | 0.946 | 0.980 |
| | **Ours** | M | × | ✓ | **5.032** | **0.165** | **1.105** | **0.117** | **0.869** | **0.967** | **0.985** |

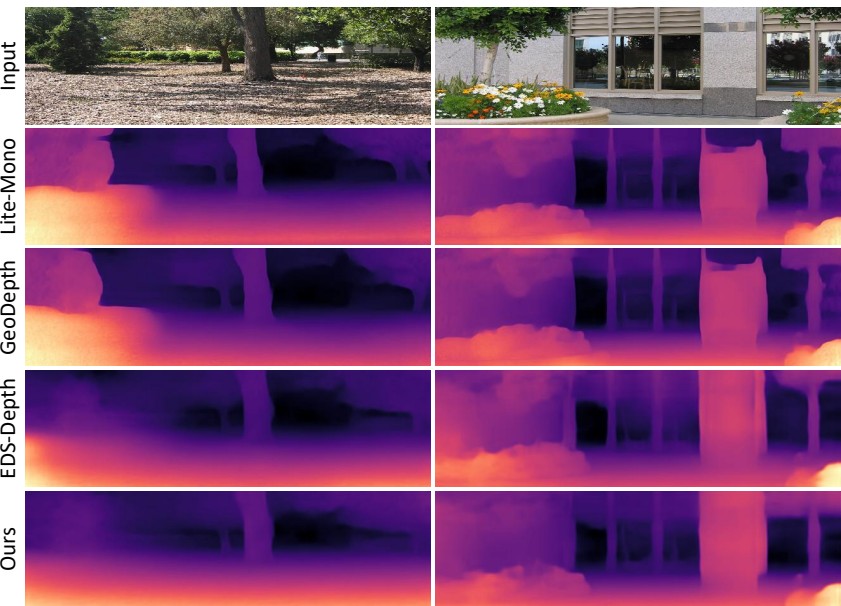

Figure 13: Qualitative results on the Make3D dataset.

Table 6: Quantitative comparison on NYUv2 and ScanNet datasets. "Mode" indicates supervision type: M = Monocular. Best results are in **bold**.

| Dataset | Method | Size | Mode | Train | Test | RMSE↓ | Abs Rel ↓ | $\delta < 1.25$ ↑ | $\delta < 1.25^2$ ↑ | $\delta < 1.25^3$ ↑ |
|---|---|---|---|---|---|---|---|---|---|---|
| NYUv2 | MovingIndoor (Zhou et al., 2019) | 320×256 | M | ✓ | ✓ | 0.712 | 0.208 | 0.674 | 0.900 | 0.968 |
| | Monodepth2 (Godard et al., 2019) | 320×256 | M | ✓ | ✓ | 0.601 | 0.160 | 0.767 | 0.949 | 0.988 |
| | P$^2$Net (Yu et al., 2020) | 320×256 | M | ✓ | ✓ | 0.599 | 0.159 | 0.772 | 0.942 | 0.987 |
| | SC-Depth (Bian et al., 2021b) | 320×256 | M | ✓ | ✓ | 0.639 | 0.159 | 0.734 | 0.937 | 0.989 |
| | PLNet (Jiang et al., 2021) | 320×256 | M | ✓ | ✓ | 0.562 | 0.151 | 0.790 | 0.953 | 0.987 |
| | StructDepth (Li et al., 2021) | 320×256 | M | ✓ | ✓ | 0.540 | 0.142 | 0.813 | 0.954 | 0.988 |
| | ADPDepth (Song et al., 2023) | 320×256 | M | ✓ | ✓ | 0.529 | 0.163 | 0.783 | 0.953 | 0.989 |
| | F$^2$Depth (Guo et al., 2024a) | 320×256 | M | ✓ | ✓ | 0.569 | 0.153 | 0.795 | 0.951 | 0.988 |
| | Guo et al. (Guo et al., 2024b) | 320×256 | M | ✓ | ✓ | 0.567 | 0.152 | 0.792 | 0.952 | 0.988 |
| | GeoDepth (Wu et al., 2025) | 320×256 | M | ✓ | ✓ | 0.520 | 0.134 | 0.833 | 0.963 | 0.991 |
| | SC-DepthV2 (Bian et al., 2021a) | 320×256 | M | ✓ | ✓ | 0.532 | 0.138 | 0.820 | 0.956 | 0.989 |
| | **Ours** | 320×256 | M | ✓ | ✓ | **0.497** | **0.128** | **0.835** | **0.965** | **0.997** |
| ScanNet | MovingIndoor (Zhou et al., 2019) | 320×256 | M | × | ✓ | 0.483 | 0.212 | 0.650 | 0.905 | 0.976 |
| | Monodepth2 (Godard et al., 2019) | 320×256 | M | × | ✓ | 0.458 | 0.170 | 0.672 | 0.922 | 0.981 |
| | TrainFlow (Zhao et al., 2020) | 320×256 | M | × | ✓ | 0.415 | 0.179 | 0.727 | 0.927 | 0.981 |
| | P$^2$Net (Yu et al., 2020) | 320×256 | M | × | ✓ | 0.420 | 0.175 | 0.740 | 0.932 | 0.982 |
| | PLNet (Jiang et al., 2021) | 320×256 | M | × | ✓ | 0.414 | 0.176 | 0.741 | 0.930 | 0.982 |
| | IFMNet (Wei et al., 2021) | 320×256 | M | × | ✓ | 0.412 | 0.176 | 0.738 | 0.929 | 0.982 |
| | SC-Depth (Bian et al., 2021b) | 320×256 | M | × | ✓ | 0.392 | 0.169 | 0.754 | 0.932 | 0.983 |
| | StructDepth (Li et al., 2021) | 320×256 | M | × | ✓ | 0.400 | 0.165 | 0.754 | 0.939 | 0.983 |
| | GeoDepth (Wu et al., 2025) | 320×256 | M | × | ✓ | 0.387 | 0.161 | 0.769 | 0.946 | 0.987 |
| | **Ours** | 320×256 | M | × | ✓ | **0.367** | **0.155** | **0.776** | **0.953** | **0.992** |

Table 7: Quantitative comparison on BONN RGB-D and TUM-Dynamic datasets. "Mode" indicates supervision type: M = Monocular. Best results are in **bold**.

| Dataset | Method | Size | Mode | Train | Test | RMSE↓ | Abs Rel↓ | $\delta < 1.25$ ↑ | $\delta < 1.25^2$ ↑ | $\delta < 1.25^3$ ↑ |
|---|---|---|---|---|---|---|---|---|---|---|
| BONN RGB-D | SC-Depth (Bian et al., 2021b) | 320×256 | M | ✓ | ✓ | 0.733 | 0.272 | 0.623 | 0.858 | 0.948 |
| | SC-DepthV2 (Bian et al., 2021a) | 320×256 | M | ✓ | ✓ | 0.619 | 0.211 | 0.714 | 0.873 | 0.936 |
| | SC-DepthV3 (Sun et al., 2023) | 320×256 | M | ✓ | ✓ | **0.379** | 0.126 | 0.889 | 0.961 | 0.980 |
| | Dyna-MSDepth (Yao et al., 2024) | 320×256 | M | ✓ | ✓ | 0.385 | 0.120 | **0.898** | 0.967 | **0.984** |
| | **Ours** | 320×256 | M | ✓ | ✓ | 0.392 | **0.117** | 0.896 | **0.969** | **0.984** |
| TUM-Dynamic | SC-Depth (Bian et al., 2021b) | 320×256 | M | ✓ | ✓ | 0.283 | 0.257 | 0.616 | 0.814 | 0.909 |
| | SC-DepthV2 (Bian et al., 2021a) | 320×256 | M | ✓ | ✓ | 0.282 | 0.223 | 0.643 | 0.862 | 0.932 |
| | SC-DepthV3 (Sun et al., 2023) | 320×256 | M | ✓ | ✓ | 0.265 | 0.163 | 0.797 | 0.882 | 0.937 |
| | Dyna-MSDepth (Yao et al., 2024) | 320×256 | M | ✓ | ✓ | 0.259 | 0157 | 0.801 | **0.885** | 0.947 |
| | **Ours** | 320×256 | M | ✓ | ✓ | **0.247** | **0.153** | **0.805** | 0.879 | **0.953** |

## A.10 ADDITIONAL QUALITATIVE COMPARISONS

We compare ROSS-Net with Lite-Mono (Zhang et al., 2023), GeoDepth (Wu et al., 2025), and EDS-Depth (Yu et al., 2025b) under zero-shot transfer on Make3D and ScanNet, without any dataset-specific tuning.

On Make3D (see Fig. 13), ROSS-Net better preserves long-range depth ordering and planar consistency on roads and facades, produces cleaner horizon lines, and mitigates sky bleeding and boundary leakage, especially around thin structures.

On ScanNet (see Fig. 14), it generates flatter wall and floor planes, sharper room corners, and more stable estimates in low-texture or repetitive regions such as tiles and shelves, while exhibiting fewer holes, reduced edge fattening, and less ringing near high-contrast boundaries.

On BONN RGB-D and TUM-Dynamic (see Fig. 15), ROSS-Net yields sharper and more coherent depth contours on moving people, with noticeably less boundary bleeding than SC-DepthV3. It also maintains clearer foreground–background separation and more consistent depth ordering in the multi-person scene.

## A.11 SEED VARIANCE, CONFIDENCE INTERVALS, AND SENSITIVITY ANALYSIS

To assess the robustness of our design and exclude gains from fragile tuning, we report multi-seed statistics and sensitivity curves for the key hyperparameters of MAF, SDC, and VCI. Table 8 summarizes the seed variance on KITTI: across five random seeds {42, 43, 44, 45, 46}, our method remains

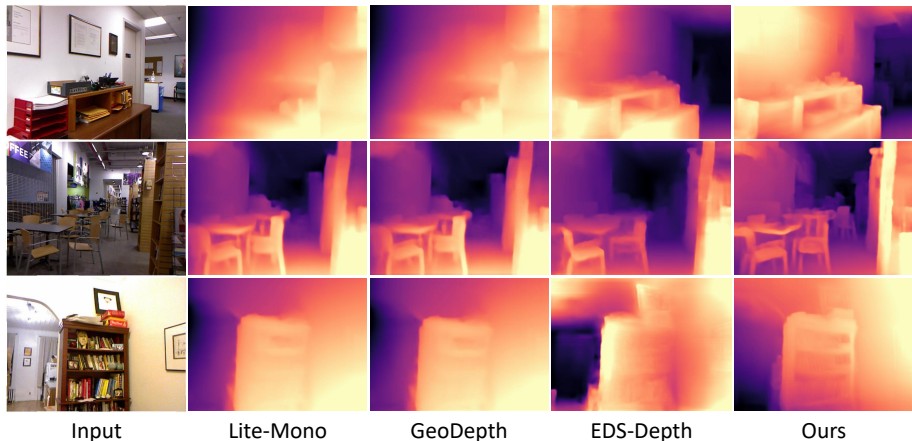

Figure 14: Qualitative results on the ScanNet dataset.

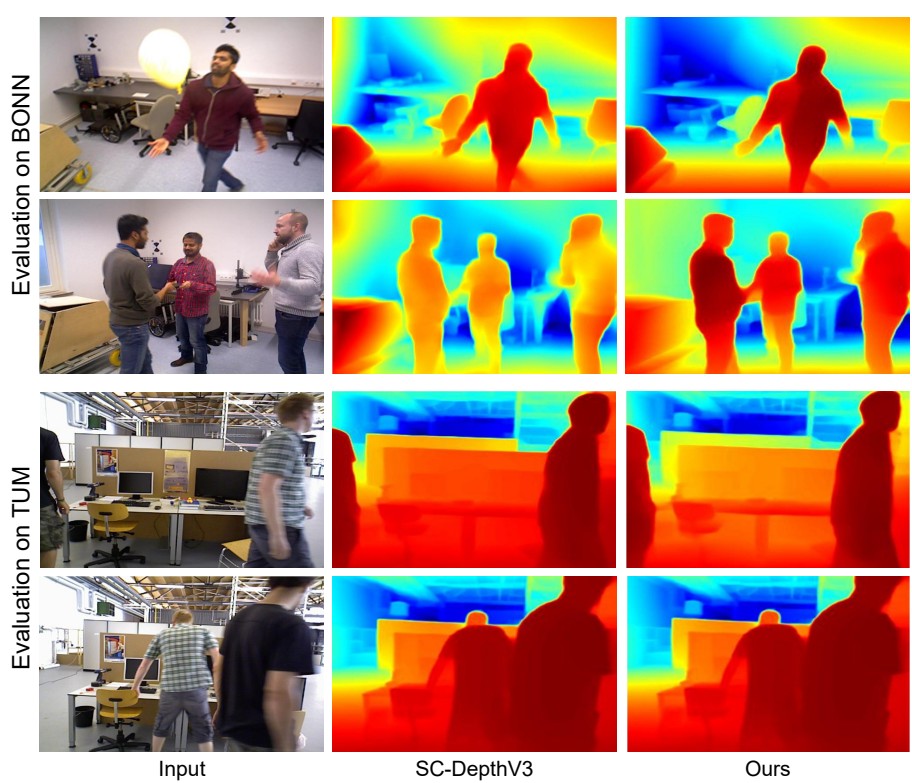

Figure 15: Qualitative results on the BONN RGB-D and TUM-Dynamic datasets.

consistent with $\delta < 1.25 = 0.921 \pm 0.004$ and Abs Rel $= 0.082 \pm 0.003$, indicating reliable improvements beyond run-to-run noise. Figs. 16, 17, 18, 19, 20 further show stable performance under wide parameter sweeps. Specifically, varying the MAF threshold $\varepsilon$ within $[0, 0.2]$ keeps $\delta < 1.25$ in 0.918–0.921 and Abs Rel in 0.082–0.085 (Fig. 16), while sweeping the EMA factor $\eta$ over $[0, 1]$ yields $\delta < 1.25$ in 0.917–0.921 and Abs Rel in 0.082–0.086 (Fig. 17). For SDC, changing the depth discretization from 10 to 100 bins gives $\delta < 1.25$ in 0.910–0.922 and Abs Rel in 0.082–0.090 (Fig. 18), showing low sensitivity to discretization choices. For VCI, adjusting the penalty sharpness $\lambda$ within $[0, 16]$ results in $\delta < 1.25$ in 0.914–0.921 and Abs Rel in 0.082–0.088 (Fig. 19). Finally, Fig. 20 compares different consensus variants. The noisy-OR consensus consistently yields the best

Table 8: Seed variance on KITTI. Abs Rel and $\delta < 1.25$ over different random seeds.

| Seed | 42 | 43 | 44 | 45 | 46 |
|---|---|---|---|---|---|
| **Abs Rel↓** | 0.0818 | 0.0821 | 0.0819 | 0.0822 | 0.0817 |
| **$\delta < 1.25$ ↑** | 0.9207 | 0.9212 | 0.9210 | 0.9214 | 0.9208 |

Abs Rel among avg- and max-consensus, supporting our integration choice. Overall, these results confirm that ROSS-Net achieves robust gains without delicate hyperparameter tuning.

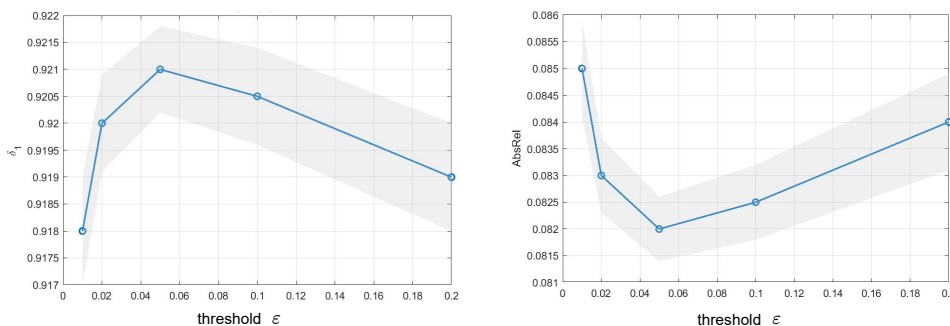

Figure 16: Sensitivity to MAF thresholds $\varepsilon$.

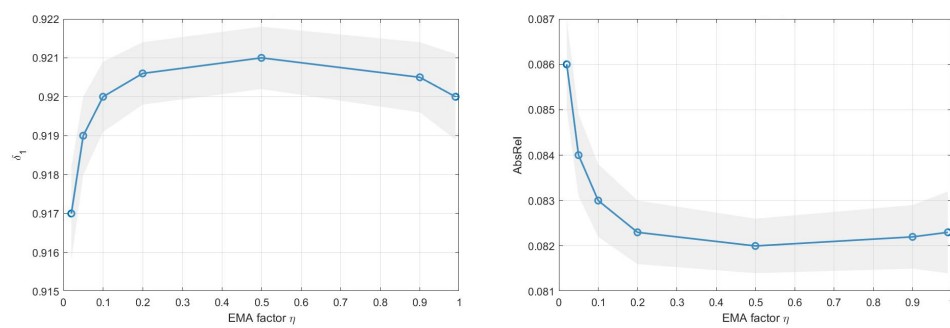

Figure 17: Sensitivity to EMA factor $\eta$ in MAF.

### A.12  I/O FLOW CLARIFICATION

Fig. 21 summarizes the end-to-end data flow of ROSS-Net. Given consecutive frames, DepthNet predicts the monocular depth $d^{\text{mono}}$, while MAF gates multi-scale features to construct the cost volume $\mathcal{C}$, which is then fed into a depth head to obtain the multi-view prediction $d^{\text{mv}}$. During training, SDC performs spectral correction to produce the fused depth $\widetilde{d}$, and VCI supplies robust consensus supervision for all depth branches.

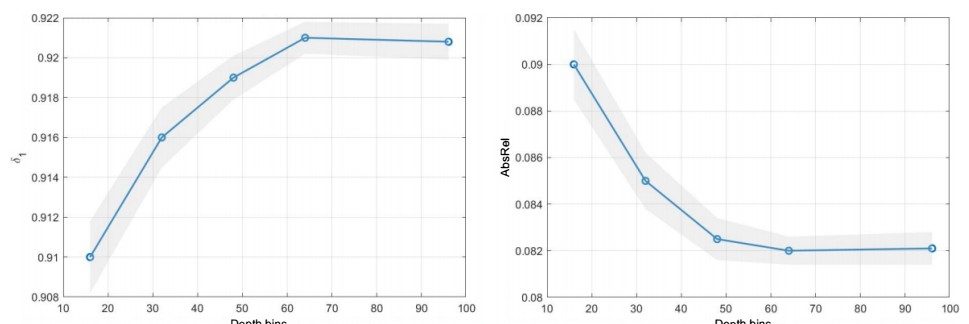

Figure 18: Sensitivity to depth sampling choices (bin number).

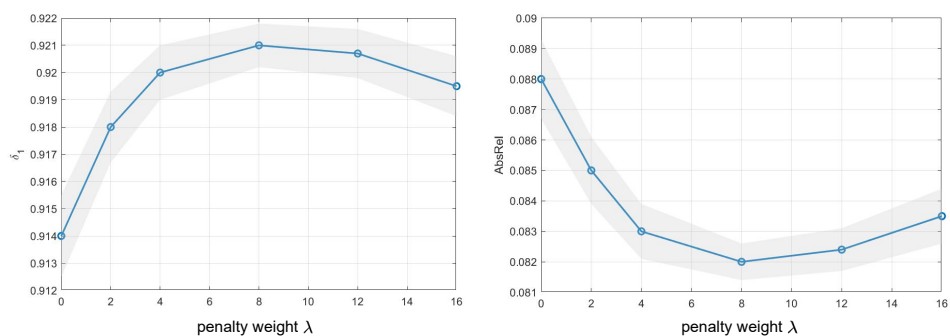

Figure 19: Sensitivity to VCI penalty setting $\lambda$.

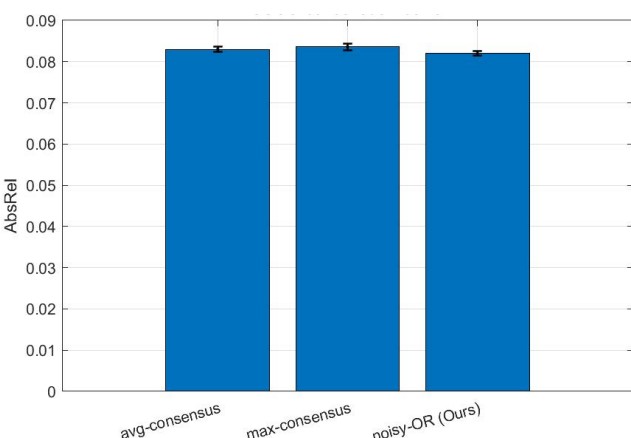

Figure 20: Sensitivity to VCI consensus settings.

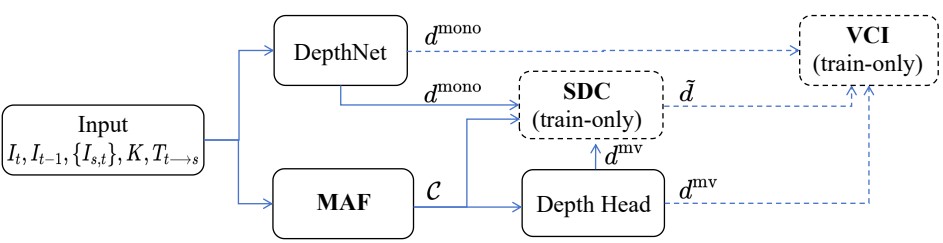

Figure 21: End-to-end input–output flow of ROSS-Net.

