# OpenReview forum: "Depth in Motion: Robust Self-Supervised Learning via Representation-Optimization-Supervision Synergy"
_ICLR.cc/2026/Conference — Submitted to ICLR 2026_

### Official Review · Reviewer_UPCe · 2025-10-15

**Soundness:** 1
**Presentation:** 1
**Contribution:** 1
**Rating:** 2
**Confidence:** 4

**Summary:**

Traditional self-supervised monocular video depth estimation methods rely on photometric constancy, leading to failures in motion and occlusion scenes because:
1. edge misalignment, where appearance edges do not align with depth boundaries.
2. gradient misalignment, where the image and depth gradients diverge.
3. tailed photometric blurries, caused by motion, occlusion, or illumination changes.

Correspondingly, the authors proposed a three-level framework called ROSS-Net across representation, optimization, and supervision:
1. STEC validates correspondences across appearance, feature, and temporal cues to produce calibrated features and cost volumes.
2. EGSI applies frequency-domain weighting using spectral entropy derived from FFT to guide optimization.
3. OSCO estimates pixel-wise confidence through trimmed quantile statistics and reweights supervision accordingly.

On the KITTI (outdoor) and NYUv2 (indoor) datasets, the author claimed that ROSS-Net outperforms existing methods (such as ManyDepth2, EDS-Depth, etc.) in all error and accuracy metrics.

**Strengths:**

1. The motivation from "tailed photometric blurries" is intuitive. When I'm training depth estimation models, the reviewer also usually observed such phenomenon that sometimes the tailed photometric residuals yields illed supervision.
2. The focus on "photometric constancy" is valuable in self-supervised depth estimation.
3. The appendix is rich. The statements of ethics and reproducibility is also provided, which are optional.

**Weaknesses:**

1. [Motivation/Idea level] The motivation from "edge misalignment" and "gradient misalignment" is not solid enough. First, they are basically the same thing. Image can be considered as the combination of so many dense factors, depth is only one of them. No matter in image to depth or video to depth, these two misalignments is not enough to be viewed as "limitations" of existing works, but something more essential and fundamental. Second, for common videos, the lighting change between continuous frames is ignorable, perhaps the "photometric constancy" should be better studied given videos with large temporal gaps. Thus, the reviewer doesn't consider these "limitations" valuable. The motivation from "tailed photometric residuals" is intuitive to the reviewer, and solving it via a re-weighting is also intuitive. But though it may produce some outliers during training, in the reviewer's experience, it has nearly zero influence in the final performance. Also, only this point is not valuable enough for a research paper.
2. [Method level] The method design of three modules is difficult and the training pipeline is complex. All three modules have multiple hyperparameters (gating thresholds, entropy smoothing terms, and quantile ratios), which are very expensive to tune. Thus, it's imaginable that it's very difficult to analyze the independent sources of performance improvement, and some small difference in settings can lead to large performance variances. Also for the efficiency, although the authors claim there is no additional overhead during inference, during training the reviewer think it will be more computationally intensive.
3. [Experiments] The table is quite large, but the author just simply summarizes it with the phrase "outperforms prior methods," without discussing why certain metrics improve more significantly, which will be more insightful than pure numbers.
4. [Writing] The correspondences between your motivation and method are not clear, the reviewer feels you were attempting to combine (all) related formulas and networks, without clearly and directly targeting the problems you want to solve and the critical bottleneck in the field. Also, the presentation of your method lacks of intuitive explanation, which requires hours to roughly understand, putting a heavy burden on the readers to understand. Moreover, the naming of your modules is weird. It takes the reviewer several minutes to separate those three names. A simpler and more understandable name would be better.

**Questions:**

Please refer to the "Weaknesses".

---

> ### Author Response · Authors · 2025-11-24
>
> We thank the reviewer for the detailed critique. We believe the motivation-method correspondence in our framework is principled, but we agree that the original presentation could be clearer and more intuitive. We have revised it accordingly by reorganizing the motivation, simplifying module naming, adding new analytical discussions, and including hyperparameter sensitivity studies. We respond point by point below.
>
> 1.[Motivation/Idea level] The motivation from "edge misalignment" and "gradient misalignment" is not solid enough. First, they are basically the same thing. Image can be considered as the combination of so many dense factors, depth is only one of them. No matter in image to depth or video to depth, these two misalignments is not enough to be viewed as "limitations" of existing works, but something more essential and fundamental. Second, for common videos, the lighting change between continuous frames is ignorable, perhaps the "photometric constancy" should be better studied given videos with large temporal gaps. Thus, the reviewer doesn't consider these "limitations" valuable. The motivation from "tailed photometric residuals" is intuitive to the reviewer, and solving it via a re-weighting is also intuitive. But though it may produce some outliers during training, in the reviewer's experience, it has nearly zero influence in the final performance. Also, only this point is not valuable enough for a research paper.

---

> > ### Author Response · Authors · 2025-11-24
> >
> > **Response:** We appreciate the reviewer’s concerns regarding our motivation. We agree that edge and gradient misalignment stem from a shared root cause: corrupted supervision from violations of the static-world assumption. Rather than unrelated issues, we frame them as complementary manifestations of this core problem at different learning stages. While fundamental in nature, they materialize as concrete, recurrent failure modes in dynamic scenes, making them actionable limitations in practice that motivate our framework's targeted, multi-level defense.
> >
> > **(1) Edge \& gradient misalignment are complementary rather than redundant**
> >
> > Both effects originate from dynamic and occluded regions, but they appear at different points of the pipeline:
> >
> > (i) Edge misalignment is a **representation-level issue**. When constructing the multi-frame cost volume from appearance similarity, high-contrast appearance edges from texture, shadows, or reflections can dominate, even when they are not true depth boundaries. Conversely, weak-contrast depth discontinuities can be under-represented. This is **not merely a basic property of images but a practical limitation:** it yields false matches and missing correspondences in the cost volume.
> >
> > (ii) Gradient misalignment is an **optimization-level issue**. Even after cost-volume construction, the gradients induced by photometric losses in these regions tend to favor reducing residuals via smoothing, which suppresses high-frequency depth structure. This reflects spectral bias under noisy supervision, but in self-supervised depth it becomes **a concrete training failure** that leads to over-smoothed boundaries.
> >
> > These two manifestations motivate two distinct design levels. MAF (formerly STEC) improves the representation by filtering unreliable correspondences and stabilizing the cost volume. SDC (formerly EGSI) addresses the optimization bias by recalibrating spectral components in the update. Our ablations (Table 2 and Fig. 8) show that removing either component leads to clear performance drops, particularly around boundaries and thin structures, which is consistent with this motivation.
> >
> > **(2) Beyond extreme lighting changes: practical violations of photometric constancy**
> >
> > We agree that global lighting changes between adjacent frames are often limited. However, violations of photometric constancy in practice arise from more common effects that appear even in standard benchmarks. These include local exposure and auto-gain variations, specular reflections, rolling shutter and motion blur, and frequent occlusions and disocclusions as objects move in and out of view. Such factors are widely acknowledged in self-supervised depth and motivate techniques like minimum reprojection and auto-masking. Our framework targets these practical, pervasive departures from the ideal static model, not only rare large-gap illumination changes. The consistent gains on KITTI and NYUv2, and more importantly on dynamic benchmarks such as BONN RGB-D, TUM-Dynamic, Cityscapes, and DrivingStereo, support that the method improves robustness in realistic settings.
> >
> > **(3) Heavy-tailed residuals and why VCI is more than trivial re-weighting**
> >
> > We agree that naive pointwise loss reweighting alone is not a sufficient contribution. Our VCI (formerly OSCO) is **a structured multi-view consensus mechanism rather than a heuristic weight**. It first estimates a robust residual scale per view using a trimmed quantile $\tau_{b,c}$, then converts residuals into credibility weights $W_{b,c}(u,v)$, and finally aggregates them through a noisy-OR field so that any consistent view can still provide supervision while jointly inconsistent dynamic outliers are suppressed. In Table 2 and Fig. 8, removing VCI leads to consistent degradation, especially in dynamic and occluded regions, showing that this consensus design contributes beyond a trivial weighting scheme.
> >
> > We hope this clarifies that the value of the contribution lies in the coherent modeling and demonstrated effectiveness.

---

> > > ### Author Response · Authors · 2025-11-24
> > >
> > > 2.[Method level] The method design of three modules is difficult and the training pipeline is complex. All three modules have multiple hyperparameters (gating thresholds, entropy smoothing terms, and quantile ratios), which are very expensive to tune. Thus, it's imaginable that it's very difficult to analyze the independent sources of performance improvement, and some small difference in settings can lead to large performance variances. Also for the efficiency, although the authors claim there is no additional overhead during inference, during training the reviewer think it will be more computationally intensive.
> > >
> > > **Response:** We thank the reviewer for the concerns on methodological complexity, hyperparameter sensitivity, and training cost. We address each aspect below and add supporting analyses in the revision.
> > >
> > > **(1) Hyperparameters are manageable and not tuned per dataset**
> > >
> > > Although the framework has three modules, **the number of effective hyperparameters is small and they are not tuned per dataset**.
> > >
> > > For MAF (formerly STEC), the appearance threshold $\varepsilon$, feature thresholds $\zeta_\ell$, and EMA factor $\eta$ follow scale-normalized rules based on empirical statistics and are kept unchanged for all datasets.
> > >
> > > For SDC (formerly EGSI), $\delta_{s}$ and $\delta_{e}$ are numerical stabilizers that are fixed constants, and the spectral encoder has a lightweight architecture shared across datasets.
> > >
> > > For VCI (formerly OSCO), $\kappa$, $\vartheta$, and $\lambda$ are selected once on KITTI validation and then reused without modification on NYUv2, BONN RGB-D, TUM-Dynamic, Cityscapes, and DrivingStereo.
> > >
> > > We have summarize all values and rules in the appendix to make this explicit and reproducible.
> > >
> > > **(2) Improvements come from the modules, not fragile tuning**
> > >
> > > We agree that multiple knobs could obscure where gains originate. We therefore validate both module effects and sensitivity.
> > >
> > > **At the module level**, Table 2 and Fig. 8 add MAF, SDC, and VCI to the baseline in a controlled manner. Each module yields a clear and consistent gain, and the cumulative improvement is substantially larger than the run-to-run variance observed under different random seeds.
> > >
> > > **At the hyperparameter level**, we use one default setting across all datasets. We additionally report seed variance for robustness (**Table R5, reported as Table 8 in the revised manuscript**) and conduct sensitivity studies by perturbing the MAF threshold $\varepsilon$, EMA factor $\eta$, the number of depth bins $m$, and the VCI penalty $\lambda$ around their defaults (**Figs. 16-19 in the revised manuscript**). The trends remain stable, indicating that the performance does not depend on delicate hyperparameter tuning.
> > >
> > > **(3) Training overhead is moderate and inference remains efficient**
> > >
> > > We acknowledge that the three modules introduce extra computation during training. However, the added operations are simple relative to the backbone.
> > >
> > > MAF uses pixelwise appearance and feature differences with EMA gating, which are linear-time elementwise operations.
> > >
> > > SDC involves a 1D FFT along the depth axis and a shallow spectral head, which adds a modest but manageable cost.
> > >
> > > VCI applies trimmed-quantile statistics and view-consensus weighting on residuals, which are again elementwise and reduction operations.
> > >
> > > In our implementation, this adds a modest training overhead. At inference time, SDC and VCI are disabled, and only the lightweight MAF gating is retained. As a result, the deployment runtime and memory footprint stay very close to the baseline, with only minimal additional cost.
> > >
> > > Overall, the framework uses a small fixed set of shared hyperparameters, provides separable and stable gains from each module, and preserves inference efficiency, while introducing only moderate additional cost during training.
> > >
> > > Table R5 Seed variance on KITTI. Abs Rel and δ<1.25 over different random seeds.
> > > | Seed | 42     | 43     | 44     | 45     | 46     |
> > > |------|--------|--------|--------|--------|--------|
> > > | **Abs Rel ↓**        | 0.0818 | 0.0821 | 0.0819 | 0.0822 | 0.0817 |
> > > | **δ < 1.25 ↑**       | 0.9207 | 0.9212 | 0.9210 | 0.9214 | 0.9208 |

---

> > > > ### Author Response · Authors · 2025-11-24
> > > >
> > > > 3.[Experiments] The table is quite large, but the author just simply summarizes it with the phrase "outperforms prior methods," without discussing why certain metrics improve more significantly, which will be more insightful than pure numbers.
> > > >
> > > > **Response:** We thank the reviewer for this comment and agree that our current textual discussion of the results is too terse. Simply stating that our method "outperforms prior methods" does not fully explain how and where the gains arise. In the revision, we have made the experimental analysis more informative in the following ways.
> > > >
> > > > **(1) Highlight which metrics benefit most and why**
> > > >
> > > > For each main table (KITTI, Make3D, NYUv2, BONN RGB-D, TUM-Dynamic, Cityscapes, DrivingStereo), we now explicitly indicate which metrics (Abs Rel, Sq Rel, RMSE, RMSE log, $\delta<1.25$, $\delta<1.25^2$, $\delta<1.25^3$) exhibit the most pronounced gains over strong baselines, and in which regimes these gains concentrate (e.g., large-depth-error tails, thin structures, regions near moving objects or occlusions). For each table we add one–two sentences of local commentary that (i) point out the dominant improvements, and (ii) briefly interpret them rather than summarizing the table with a single "outperforms prior methods" statement.
> > > >
> > > > **(2) Connect metric patterns to specific failure modes**
> > > >
> > > > We also connect these patterns back to the identified failure modes and to the roles of MAF (formerly STEC), SDC (formerly EGSI), and VCI (formerly OSCO). For indoor benchmarks, we discuss how improved error and accuracy metrics near depth discontinuities and occlusions align with our analysis of edge and gradient misalignment and the effect of MAF and SDC. For driving datasets, we emphasize how reductions in large errors on dynamic objects and around occlusion boundaries reflect the contribution of MAF and VCI in handling motion and heavy-tailed residuals. Thus, readers can see not only that our method improves over prior work, but also why particular metrics improve more than others.
> > > >
> > > > We believe these additions (**Secs. 4.2, 4.3**, and **A.9** in the revised manuscript) make the experimental section more informative and address the reviewer’s concern about going beyond raw numbers.

---

> > > > > ### Author Response · Authors · 2025-11-24
> > > > >
> > > > > 4.[Writing] The correspondences between your motivation and method are not clear, the reviewer feels you were attempting to combine (all) related formulas and networks, without clearly and directly targeting the problems you want to solve and the critical bottleneck in the field. Also, the presentation of your method lacks of intuitive explanation, which requires hours to roughly understand, putting a heavy burden on the readers to understand. Moreover, the naming of your modules is weird. It takes the reviewer several minutes to separate those three names. A simpler and more understandable name would be better.
> > > > >
> > > > > **Response:** We thank the reviewer for the clear feedback on presentation and readability. We agree that the motivation–method mapping and the pipeline flow can be communicated more explicitly and intuitively, and we have revised the draft accordingly.
> > > > >
> > > > > **(1) Clarifying the motivation-method correspondence**
> > > > >
> > > > > (i) We have **restructured Sec. 3.1** ("Motivation and Overview") to include a clear, bulleted list that explicitly maps each core problem (edge misalignment, gradient misalignment, heavy-tailed residuals) to its corresponding solution module (MAF, SDC, VCI) and its targeted level (representation, optimization, supervision).
> > > > >
> > > > > (ii) We have **added brief introductory statements at the beginning of each methodological subsection (Secs. 3.2-3.4)** to re-establish this mapping before delving into details.
> > > > >
> > > > > This removes any impression that the method is a collection of loosely related formulas.
> > > > >
> > > > > **(2) Strengthening intuitive explanation and reducing reader burden**
> > > > >
> > > > > To make the method more accessible, we have implemented several key changes:
> > > > >
> > > > > (i)Revised Fig. 2 into a single input-to-output pipeline with explicitly labeled inputs, outputs, and data flow, and with each module annotated by its functional role.
> > > > >
> > > > > (ii) Each technical subsection now begins with a short plain-language preview that explains what the module does, how it affects the cost volume, gradients, or supervision, and why this helps in dynamic or occluded regions.
> > > > >
> > > > > (iii) Streamlined notation in the main text and move secondary derivations to the appendix, keeping the main body focused on the core mechanisms and intuition.
> > > > >
> > > > > This allows readers to understand each module’s core role without first parsing all mathematical details.
> > > > >
> > > > > **(3) Simplifying module naming**
> > > > >
> > > > > We agree that the original names were cumbersome. Throughout the revised manuscript, we now use simplified, descriptive names:
> > > > >
> > > > > **MAF (Motion-Anomaly Filter)** - formerly STEC
> > > > >
> > > > > **SDC (Spectral-Discrepancy Calibrator)** - formerly EGSI
> > > > >
> > > > > **VCI (View-Consistent Integrator)** - formerly OSCO
> > > > >
> > > > > These new names are introduced with a one-sentence functional description at first use and are consistently employed in all text, figures, and tables to ensure clarity and ease of recall.
> > > > >
> > > > > We are confident that these comprehensive revisions will significantly lower the barrier to understanding and clearly demonstrate how our framework directly targets the critical bottlenecks in dynamic depth estimation.

---

### Official Review · Reviewer_WG7J · 2025-10-19

**Soundness:** 2
**Presentation:** 1
**Contribution:** 2
**Rating:** 6
**Confidence:** 5

**Summary:**

This paper tackles the challenge of self-supervised monocular depth estimation in dynamic scenes, where traditional photometric consistency assumptions fail. The authors propose a holistic framework called ROSS-Net (Representation-Optimization-Supervision Synergy Network) that introduces a three-level defense mechanism to improve robustness by addressing failures at the representation, optimization, and supervision levels. The paper demonstrates state-of-the-art performance on KITTI and strong generalization capabilities.

This paper presents an ambitious, tripartite framework to solve the problem of self-supervised depth estimation in dynamic scenes. Its motivation is clear, and the experimental results on existing benchmarks are impressive. However, the paper has some major potential flaws, particularly concerning the design rationale of its core module (STEC) and the experimental setup (validating a dynamic model on the static NYUv2 dataset).

**My recommendation is therefore conditional (just because of the impressive results). If the authors can provide a convincing explanation and clarification in their rebuttal to the weaknesses and questions raised, I will maintain my score.** If the authors cannot adequately address these major concerns, I will consider `lowering` my score.

**Strengths:**

- Ambitious Technical Approach: The paper is technically ambitious, combining ideas from geometric validation (STEC), frequency-domain analysis (EGSI), and robust statistics (OSCO). This comprehensive approach attempts to build a multi-faceted defense against motion-induced artifacts.
- Strong Empirical Results on KITTI: The method achieves state-of-the-art results on the challenging KITTI benchmark, demonstrating its effectiveness in outdoor driving scenarios where dynamic objects are prevalent. The quantitative and qualitative results show clear improvements over prior methods in handling object boundaries and thin structures.

**Weaknesses:**

- Questionable Soundness of the STEC Module: The claim that STEC "preserves dynamic evidence while mitigating motion-induced mismatches" is not well-supported, and its underlying mechanism appears flawed. The module seems to rely on inter-frame photometric or feature differences to identify motion. This heuristic is incorrect for general motion patterns. For example, in the case of two cars moving towards each other, the difference will be large, but this large difference does not provide a correct supervisory signal for a rigid-world assumption. This issue has been discussed in detail in previous works.

- Illogical Experimental Setup on NYUv2: A major weakness is the decision to train and evaluate on the NYUv2 dataset. This benchmark consists almost exclusively of static indoor scenes with virtually no dynamic objects. Using this dataset to validate a method explicitly designed to handle motion and dynamic scenes is inappropriate and undermines the credibility of the results presented for indoor environments.

- Insufficient Scalability Validation: The paper lacks validation on other standard and challenging driving datasets like Cityscapes and DrivingStereo. It is common practice in the literature to evaluate on these benchmarks to demonstrate robustness and generalization. Many prior works have done so, including MonoViT [1], RobustDepth [2], D4RD[3], Yan et al [4], and Jasmine [5]. This omission limits the assessment of the method's applicability. Please also cite these papers because they are very important papers in this area!

- Unsupported Core Motivation: The paper states that motion and occlusion "bias optimization toward texture-invariant, low-frequency cues." This is a strong claim that serves as a key motivation for the work, but it is presented without any citation or empirical proof.

- Complex and Obfuscating Terminology: The paper uses a very specific and complex naming scheme for its components. This makes the paper difficult to read and feels like it may be masking the true, and potentially simple (or flawed, in the case of STEC), underlying ideas.
```
[1]Zhao, Chaoqiang, et al. "Monovit: Self-supervised monocular depth estimation with a vision transformer." 2022 international conference on 3D vision (3DV). IEEE, 2022.
[2]Saunders K, Vogiatzis G, Manso L J. Self-supervised monocular depth estimation: Let's talk about the weather[C]//Proceedings of the IEEE/CVF International Conference on Computer Vision. 2023: 8907-8917.
[3]Wang, Jiyuan, et al. "Digging into contrastive learning for robust depth estimation with diffusion models." Proceedings of the 32nd ACM International Conference on Multimedia. 2024.
[4]Yan W, Li M, Li H, et al. Synthetic-to-real self-supervised robust depth estimation via learning with motion and structure priors[C]//Proceedings of the Computer Vision and Pattern Recognition Conference. 2025: 21880-21890.
[5]Wang J, Lin C, Guan C, et al. Jasmine: Harnessing Diffusion Prior for Self-supervised Depth Estimation[C]. Advances in neural information processing systems, 2025, 30.

**Questions:**

- Justification for Core Claim: In the abstract and introduction, you claim that motion and occlusion bias optimization toward low-frequency cues. What is the source of this claim? Is it based on established findings in prior work, or is it an observation from your own analysis? If the former, please provide the necessary citations.

- Rationale for STEC's Design: Given that simple frame differencing is known to be an unreliable cue for handling dynamic objects under general motion, could you provide a more detailed justification for the design of the STEC module? How does it handle common scenarios like opposing traffic or objects moving parallel to the camera at different speeds?

- Rationale for using NYUv2: Could you explain the reasoning behind training and evaluating a method designed for dynamic scenes on the static NYUv2 dataset?

- Clarification of Notation:
  - In Equation (6), what does the time index $t$ represent? Is it the frame index in a video sequence or the training iteration?
  - In Equation (9), what is the index $m$ for the spectral bin? Please provide a clear definition.

---

> ### Author Response · Authors · 2025-11-24
>
> We sincerely thank the reviewer for the careful and constructive review and for the conditional recommendation based on our empirical results. We are encouraged that the reviewer finds our framework ambitious, the motivation clear and the KITTI results strong. At the same time, we fully acknowledge the major concerns about the design rationale of STEC (now MAF), the use of NYUv2 for evaluation, the breadth of scalability experiments and the support for our low-frequency bias motivation. In the revised manuscript and in the responses below, we address each of these points with new analysis, experiments and clarifications. We have also simplified the terminology to make the method easier to follow. We hope these revisions resolve the reviewer’s concerns.
>
> **Weaknesses:**
>
> 1.Questionable Soundness of the STEC Module: The claim that STEC "preserves dynamic evidence while mitigating motion-induced mismatches" is not well-supported, and its underlying mechanism appears flawed. The module seems to rely on inter-frame photometric or feature differences to identify motion. This heuristic is incorrect for general motion patterns. For example, in the case of two cars moving towards each other, the difference will be large, but this large difference does not provide a correct supervisory signal for a rigid-world assumption. This issue has been discussed in detail in previous works.
>
> **Response:** We thank the reviewer for this insightful comment. We agree that using inter-frame differences as a direct rigid supervisory signal is flawed, as in the example of two cars moving toward each other. However, that is not the intended role of our STEC (now renamed to **MAF** in the revised manuscript), which we clarify below.
>
> **(1) MAF is a gating and stabilization mechanism, not a rigid-world supervisor**
>
> The appearance and feature differences in Eqs. (1)-(3) are used to flag potentially unreliable regions and to construct a soft mask $\mathcal{K}K^{\mathrm{seed}}_{\ell}$ and its temporally smoothed version $\mathcal{K}_{\ell,t}$. These masks modulate the cost volume and feature aggregation so that:
>
> (i) regions with large and inconsistent changes are treated as uncertain and downweighted in the multi-view pathway; and
>
> (ii) stable regions with consistent evidence are retained as reliable geometric support.
>
> In other words, MAF does not assume that large differences correspond to correct rigid matches; it explicitly marks them as locations where rigid multi-view cues are fragile and should be handled cautiously.
>
> **(2) Dynamic regions are handled by uncertainty, not rigidity**
>
> MAF's gating is the first line of defense. The actual handling of dynamic regions is accomplished by our dedicated modules:
>
> (i) EGSI (now renamed to **SDC** in the revised manuscript) converts spectral entropy from the cost volume into a pixel-wise uncertainty map $\mathcal{U}$ (Sec. 3.3). High-entropy, spectrally diffuse regions (typical near dynamic objects and occlusions) lead to larger $\mathcal{U}$, which shifts the fusion in Eq. (8) away from the multi-view estimate $d^{\text{mv}}$ and towards the monocular prior $d^{\text{mono}}$. Thus, where motion breaks the rigid assumption, the model is explicitly discouraged from trusting the rigid warping.
>
> (ii) OSCO (now renamed to **VCI** in the revised manuscript) further prevents dynamic outliers from dominating the loss (Sec. 3.4): the trimmed-order statistic $\tau_{b,c}$ and the credibility weights $W_{b,c}(u,v)$ in Eq. (9) smoothly attenuate large residuals and aggregate supervision via a noisy-OR across views. Dynamic inconsistencies, such as the reviewer’s "two cars moving towards each other" example, yield large residuals that are strongly downweighted, rather than being treated as valid rigid evidence.
>
> **(3) "Preserves dynamic evidence" refers to representation, not rigid constraints**
>
> Our wording "preserves dynamic evidence while mitigating motion-induced mismatches" is meant to highlight that MAF does not hard-mask away all dynamic content, which would remove potentially useful cues such as motion boundaries and partial parallax. Instead it
>
> (i) keeps these regions in the feature space,
>
> (ii) passes them forward with an explicit uncertainty treatment in SDC and VCI, and
>
> (iii) avoids letting them dominate rigid photometric supervision.
>
> **(4) Empirical support**
>
> This design is strongly supported by our experiments. Our method achieves SOTA not only on KITTI but also on new challenging dynamic benchmarks like BONN RGB-D and TUM-Dynamic (Table R1, reported as Table 7 in the revised manuscript). Crucially, our ablation study (Table 2) shows that removing MAF causes a significant performance drop, proving that its role in stabilizing the cost volume is indispensable for final accuracy.
>
> We have revised the manuscript to clarify MAF's role as a soft gate and to highlight its synergy with SDC and VCI, avoiding any wording that could imply it uses frame differences as a rigid supervision.

---

> > ### Author Response · Authors · 2025-11-24
> >
> > Table R1 Quantitative comparison on **BONN RGB-D** and **TUM-Dynamic** datasets. "Mode" indicates supervision type: M = Monocular. Best results are in **bold**.
> > | Dataset      | Method            | Size         | Mode | RMSE ↓ | Abs Rel ↓ | δ < 1.25 ↑ | δ < 1.25² ↑ | δ < 1.25³ ↑ |
> > |--------------|-------------------|--------------|------|--------|-----------|------------|-------------|-------------|
> > | BONN RGB-D   | SC-Depth [2]      | 320 × 256    | M    | 0.733  | 0.272     | 0.623      | 0.858       | 0.948       |
> > | BONN RGB-D   | SC-DepthV2 [3]    | 320 × 256    | M    | 0.619  | 0.211     | 0.714      | 0.873       | 0.936       |
> > | BONN RGB-D   | SC-DepthV3 [4]    | 320 × 256    | M    | **0.379** | 0.126  | 0.889      | 0.961       | 0.980       |
> > | BONN RGB-D   | Dyna-MSDepth [5]  | 320 × 256    | M    | 0.385  | 0.120     | **0.898**  | 0.967       | **0.984**   |
> > | BONN RGB-D   | Ours              | 320 × 256    | M    | 0.392  | **0.117** | 0.896      | **0.969**   | **0.984**   |
> > | TUM-Dynamic  | SC-Depth [2]      | 320 × 256    | M    | 0.283  | 0.257     | 0.616      | 0.814       | 0.909       |
> > | TUM-Dynamic  | SC-DepthV2 [3]    | 320 × 256    | M    | 0.282  | 0.223     | 0.643      | 0.862       | 0.932       |
> > | TUM-Dynamic  | SC-DepthV3 [4]    | 320 × 256    | M    | 0.265  | 0.163     | 0.797      | 0.882       | 0.937       |
> > | TUM-Dynamic  | Dyna-MSDepth [5]  | 320 × 256    | M    | 0.259  | 0.157     | 0.801      | **0.885**   | 0.947       |
> > | TUM-Dynamic  | Ours              | 320 × 256    | M    | **0.247** | **0.153** | **0.805** | 0.879       | **0.953**   |
> >
> > Table 2 Ablation study of our ROSS-Net on the KITTI dataset. Best results are in **bold**.
> >
> > | Method | RMSE log ↓ | Abs Rel ↓ | δ < 1.25 ↑ | δ < 1.25² ↑ | δ < 1.25³ ↑ |
> > |--------|------------|-----------|------------|-------------|-------------|
> > | Baseline [Xiang et al. 2023] | 0.176 | 0.102 | 0.898 | 0.965 | 0.984 |
> > | Baseline + MAF | 0.169 | 0.093 | 0.913 | 0.969 | 0.984 |
> > | Baseline + SDC | 0.170 | 0.096 | 0.910 | **0.970** | 0.984 |
> > | Baseline + VCI | 0.170 | 0.094 | 0.903 | 0.968 | 0.984 |
> > | Baseline + MAF + VCI | 0.167 | **0.086** | 0.914 | **0.970** | **0.985** |
> > | Baseline + SDC + VCI | 0.168 | **0.088** | 0.912 | **0.970** | **0.985** |
> > | **Ours** | **0.162** | **0.082** | **0.921** | **0.970** | **0.985** |
> >
> > 2.Illogical Experimental Setup on NYUv2: A major weakness is the decision to train and evaluate on the NYUv2 dataset. This benchmark consists almost exclusively of static indoor scenes with virtually no dynamic objects. Using this dataset to validate a method explicitly designed to handle motion and dynamic scenes is inappropriate and undermines the credibility of the results presented for indoor environments.
> >
> > **Response:** We thank the reviewer for this valid concern. We agree that NYUv2 is not a dynamic-scene benchmark. However, its inclusion is critical for establishing comprehensive performance, and we have now supplemented it with dedicated dynamic evaluations.
> >
> > **(1) Necessity of a Standard Benchmark:** NYUv2 remains the de facto standard indoor benchmark for self-supervised monocular depth estimation. Reporting results on it is essential to ensure our method remains competitive with established baselines [1-3] and to demonstrate that our dynamic-scene regularizations do not degrade performance on complex static geometry.
> >
> > **(2) Generalization Beyond Strict Dynamics:** Although NYUv2 is largely static, it still contains common violations of the static-world assumption such as occlusions, depth discontinuities, lighting changes and reflective surfaces. The gains on NYUv2 therefore indicate improved robustness to these indoor photometric and geometric inconsistencies, which complements our dynamic-scene focus.
> >
> > **(3) Direct Evidence on Dynamic Benchmarks:** More importantly, we have added new experiments on **BONN RGB-D** and **TUM-Dynamic** for indoor dynamics, and on **Cityscapes** and **DrivingStereo** for outdoor driving dynamics. The results in Tables R1, R3 and R4 (reported as Tables 4, 5, and 7 in the revised manuscript) show consistent improvements over state-of-the-art methods in motion-heavy settings, which directly supports our core claims on dynamic-scene robustness.
> >
> > [1] Godard, C., et al. "Digging into self-supervised monocular depth estimation." ICCV 2019.
> >
> > [2]Bian, Jia-Wang, Huangying Zhan, Naiyan Wang, Zhichao Li, Le Zhang, Chunhua Shen, Ming-Ming Cheng, and Ian Reid. "Unsupervised scale-consistent depth learning from video." IJCV, no. 9 (2021): 2548-2564.
> >
> > [3]Sun, Libo, Jia-Wang Bian, Huangying Zhan, Wei Yin, Ian Reid, and Chunhua Shen. "Sc-depthv3: Robust self-supervised monocular depth estimation for dynamic scenes." IEEE TPAMI, no. 1 (2023): 497-508.

---

> > > ### Author Response · Authors · 2025-11-24
> > >
> > > 3.Insufficient Scalability Validation: The paper lacks validation on other standard and challenging driving datasets like Cityscapes and DrivingStereo. It is common practice in the literature to evaluate on these benchmarks to demonstrate robustness and generalization. Many prior works have done so, including MonoViT [1], RobustDepth [2], D4RD[3], Yan et al [4], and Jasmine [5]. This omission limits the assessment of the method's applicability. Please also cite these papers because they are very important papers in this area!
> > >
> > > **Response:** We thank the reviewer for this critical suggestion regarding scalability validation. We fully agree that evaluation on Cityscapes and DrivingStereo is essential for demonstrating generalization.
> > >
> > > In response, we have conducted new experiments on both **Cityscapes** and **DrivingStereo**, following the standard protocol of training on KITTI and performing zero-shot evaluation. The results (**Tables R3 and R4, reported as Tables 4 and 5 in the revised manuscript**) demonstrate that our method achieves state-of-the-art or highly competitive performance compared to the cited works (**MonoViT, RobustDepth, D4RD, Yan et al., Jasmine**) and other recent methods.
> > >
> > > Furthermore, **we have added the suggested citations** to experimental sections. We believe these additions, motivated by your comment, significantly strengthen our validation and contextualize our contribution more firmly within the field. Thank you again for this constructive suggestion.
> > >
> > > Table R3 Quantitative comparison on **Cityscapes** dataset. "Mode" indicates supervision type: M = Monocular. Best results are in **bold**.
> > > | Method       | Mode | RMSE ↓ | RMSE log ↓ | Sq Rel ↓ | Abs Rel ↓ | δ < 1.25 ↑ | δ < 1.25² ↑ | δ < 1.25³ ↑ |
> > > |--------------|------|--------|------------|----------|-----------|------------|-------------|-------------|
> > > | Monodepth2   | M    | 8.590  | 0.234      | 1.785    | 0.153     | 0.774      | 0.926       | 0.976       |
> > > | SD-SSMDE     | M    | 8.441  | 0.221      | 1.635    | 0.143     | 0.789      | 0.931       | 0.980       |
> > > | MonoPP       | M    | 12.113 | —          | 3.156    | 0.216     | 0.580      | —           | —           |
> > > | RobustDepth  | M    | 7.912  | 0.224      | 1.569    | 0.160     | 0.757      | 0.937       | 0.982       |
> > > | Jasmine      | M    | 6.618  | —          | —        | 0.123     | 0.852      | —           | —           |
> > > | Ours         | M    | **5.437** | **0.155** | **0.882** | **0.103** | **0.881** | **0.971**   | **0.992**   |

---

> > > > ### Author Response · Authors · 2025-11-24
> > > >
> > > > Table R4 Quantitative comparison on **DrivingStereo** dataset. "Mode" indicates supervision type: M = Monocular. Best results are in **bold**.
> > > > | Domain | Method       | Mode | RMSE ↓ | RMSE log ↓ | Sq Rel ↓ | Abs Rel ↓ | δ < 1.25 ↑ | δ < 1.25² ↑ | δ < 1.25³ ↑ |
> > > > |--------|--------------|------|--------|------------|----------|-----------|------------|-------------|-------------|
> > > > | Foggy  | MonoDepth2   | M    | 7.927  | 0.195      | 1.514    | 0.125     | 0.849      | 0.950       | 0.980       |
> > > > | Foggy  | MonoViT      | M    | 6.313  | 0.150      | 0.934    | **0.096** | 0.893      | 0.974       | 0.993       |
> > > > | Foggy  | RobustDepth  | M    | 9.098  | 0.203      | 1.907    | 0.140     | 0.827      | 0.949       | 0.980       |
> > > > | Foggy  | D4RD         | M    | 7.102  | 0.154      | 1.061    | 0.105     | 0.883      | 0.975       | **0.994**   |
> > > > | Foggy  | Yan et al.   | M    | 6.422  | —          | 1.023    | 0.128     | 0.845      | —           | —           |
> > > > | Foggy  | Jasmine      | M    | **5.702** | —       | —        | 0.098     | **0.902**  | —           | —           |
> > > > | Foggy  | Ours         | M    | 6.427  | **0.148**  | **0.930**| 0.102     | 0.891      | **0.976**   | 0.991       |
> > > > | Cloudy | MonoDepth2   | M    | 6.976  | 0.209      | 1.900    | 0.155     | 0.813      | 0.943       | 0.979       |
> > > > | Cloudy | MonoViT      | M    | 5.970  | 0.177      | 1.300    | 0.125     | 0.861      | 0.958       | 0.986       |
> > > > | Cloudy | RobustDepth  | M    | 8.269  | 0.231      | 2.281    | 0.173     | 0.782      | 0.933       | 0.973       |
> > > > | Cloudy | D4RD         | M    | 7.271  | 0.198      | 1.560    | 0.141     | 0.830      | 0.948       | 0.983       |
> > > > | Cloudy | Jasmine      | M    | **5.651** | —       | —        | 0.133     | 0.849      | —           | —           |
> > > > | Cloudy | Ours         | M    | 5.833  | **0.169**  | **1.224**| **0.118** | **0.865**  | **0.961**   | **0.987**   |
> > > > | Rainy  | MonoDepth2   | M    | 11.040 | 0.301      | 3.339    | 0.240     | 0.591      | 0.857       | 0.952       |
> > > > | Rainy  | MonoViT      | M    | 8.604  | 0.219      | 1.925    | 0.169     | 0.733      | 0.934       | 0.985       |
> > > > | Rainy  | RobustDepth  | M    | 10.595 | 0.260      | 2.670    | 0.199     | 0.677      | 0.902       | 0.972       |
> > > > | Rainy  | D4RD         | M    | 8.584  | 0.208      | **1.722**| 0.158     | 0.773      | 0.946       | 0.985       |
> > > > | Rainy  | Yan et al.   | M    | 8.004  | —          | 1.755    | 0.171     | 0.730      | —           | —           |
> > > > | Rainy  | Jasmine      | M    | **7.194** | —       | —        | 0.160     | **0.787**  | —           | —           |
> > > > | Rainy  | Ours         | M    | 8.293  | **0.197**  | 1.816    | **0.157** | 0.752      | **0.950**   | **0.986**   |
> > > > | Sunny  | MonoDepth2   | M    | 6.744  | 0.214      | 1.740    | 0.155     | 0.819      | 0.941       | 0.977       |
> > > > | Sunny  | MonoViT      | M    | 6.109  | 0.186      | 1.266    | 0.130     | 0.851      | 0.956       | **0.985**   |
> > > > | Sunny  | RobustDepth  | M    | 8.084  | 0.246      | 2.174    | 0.185     | 0.765      | 0.919       | 0.965       |
> > > > | Sunny  | D4RD         | M    | 7.121  | 0.207      | 1.437    | 0.149     | 0.815      | 0.946       | 0.980       |
> > > > | Sunny  | Ours         | M    | **5.032** | **0.165** | **1.105** | **0.117** | **0.869** | **0.967** | **0.985** |

---

> > > > > ### Author Response · Authors · 2025-11-24
> > > > >
> > > > > 4.Unsupported Core Motivation: The paper states that motion and occlusion "bias optimization toward texture-invariant, low-frequency cues." This is a strong claim that serves as a key motivation for the work, but it is presented without any citation or empirical proof.
> > > > >
> > > > > **Response:** Thank you for emphasizing this point. We agree that the low-frequency bias motivation is critical and therefore deserves an explicit, well-grounded justification. In the revision, we strengthen it as a causal chain rooted in prior findings, and we also clarify this grounding in Sec. 3.1.
> > > > > ﻿
> > > > >
> > > > > **(1) Literature-grounded causal chain**
> > > > >
> > > > > Our statement that motion and occlusion bias optimization toward texture-invariant low-frequency cues is not an isolated intuition. It follows a concrete chain of established observations:
> > > > >
> > > > > **(i) Motion and occlusion invalidate rigid photometric supervision.**
> > > > >
> > > > > Dynamic objects and occlusions violate the static-scene assumption, producing inconsistent warping residuals and unreliable gradients. This has been repeatedly identified as the main failure source in self-supervised depth under dynamics, including SC-DepthV3 [3] and related analyses.
> > > > > ﻿
> > > > >
> > > > > **(ii) The resulting inconsistencies manifest as corrupted, spectrally diffuse cost volumes.**
> > > > >
> > > > > When correspondence reliability breaks, the cost volume or depth-likelihood curve becomes flatter and multi-modal, because spurious matches inject energy across many depth hypotheses. Watson et al. [4] explicitly analyze this corruption effect in dynamic multi-frame depth, motivating cost-volume purification as a prerequisite for stable learning.
> > > > > ﻿
> > > > >
> > > > > **(iii) Under such noisy, multi-modal supervision, optimization collapses toward low-frequency solutions.**
> > > > >
> > > > >
> > > > > Neural networks exhibit spectral bias, preferentially fitting low-frequency components when supervision is ambiguous or noisy [5]. In dynamic self-supervised depth, motion- and occlusion-induced cost-volume corruption directly introduces this noise and ambiguity into the supervision signal [6].
> > > > > ﻿
> > > > >
> > > > > Taken together, (i) yields unreliable supervision, (ii) spreads likelihood energy across depth and attenuates high-frequency cues, and (iii) drives gradient descent toward texture-invariant low-frequency solutions. This is what we summarized as a “low-frequency bias induced by motion and occlusion,” now stated explicitly with citations in Sec. 3.1.
> > > > > ﻿
> > > > >
> > > > > **(2) Why this matters for our design**
> > > > >
> > > > >
> > > > > This motivation directly informs a pipeline-aligned defense in ROSS-Net. MAF targets (ii) by purifying correspondences before cost-volume aggregation. SDC targets (iii) by using spectral entropy to detect and reweight ambiguous likelihoods, preventing optimization from defaulting to low-frequency updates. VCI targets (i) by bounding heavy-tailed residuals so dynamic outliers do not dominate supervision. Their synergy therefore addresses the same failure chain above in a principled, end-to-end manner.
> > > > > ﻿
> > > > >
> > > > > We hope this revised justification makes clear that the motivation is grounded in prior work through a coherent causal mechanism, and that it is tightly aligned with our method design rather than being an unsupported narrative claim.
> > > > > ﻿
> > > > >
> > > > > [4]Watson, Jamie, Oisin Mac Aodha, Victor Prisacariu, Gabriel Brostow, and Michael Firman. "The temporal opportunist: Self-supervised multi-frame monocular depth." CVPR, pp. 1164-1174. 2021.
> > > > >
> > > > > [5]Rahaman, Nasim, Aristide Baratin, Devansh Arpit, Felix Draxler, Min Lin, Fred Hamprecht, Yoshua Bengio, and Aaron Courville. "On the spectral bias of neural networks." ICML, pp. 5301-5310. PMLR, 2019.
> > > > >
> > > > > [6]Miao, Xingyu, Yang Bai, Haoran Duan, Yawen Huang, Fan Wan, Xinxing Xu, Yang Long, and Yefeng Zheng. "Ds-depth: Dynamic and static depth estimation via a fusion cost volume." IEEE TCSVT, 34(4), pp.2564-2576. 2023.

---

> > > > > > ### Author Response · Authors · 2025-11-24
> > > > > >
> > > > > > 5.Complex and Obfuscating Terminology: The paper uses a very specific and complex naming scheme for its components. This makes the paper difficult to read and feels like it may be masking the true, and potentially simple (or flawed, in the case of STEC), underlying ideas.
> > > > > >
> > > > > > **Response:** We appreciate the reviewer’s concern that our previous naming scheme may have appeared overly complex and could obscure the underlying ideas. Our intention was to emphasize the distinct roles at the representation, optimization, and supervision levels, but we agree that the terminology can be made more transparent.
> > > > > >
> > > > > > In the revised manuscript, we simplify the component names and align them more closely with their concrete functions. We now refer to the three parts as:
> > > > > >
> > > > > > **(i) Motion-Anomaly Filter (MAF)**, formerly STEC. It downweights motion-induced anomalous correspondences in the cost volume while preserving useful parallax cues.
> > > > > >
> > > > > > **(ii) Spectral-Discrepancy Calibrator (SDC)**, formerly EGSI. It calibrates the depth update by reweighting spectral components based on entropy-derived uncertainty.
> > > > > >
> > > > > > **(iii) View-Consistent Integrator (VCI)**, formerly OSCO. It aggregates multi-view residuals into a robust, view-consistent supervisory signal via order statistics.
> > > > > >
> > > > > > We revise the text and figures to consistently use these shorter, function-oriented names together with brief intuitive descriptions. We hope this makes it clear that the underlying ideas are straightforward and that our terminology is meant to clarify, rather than mask, the core design.
> > > > > >
> > > > > > **Questions:**
> > > > > >
> > > > > > 1.Justification for Core Claim: In the abstract and introduction, you claim that motion and occlusion bias optimization toward low-frequency cues. What is the source of this claim? Is it based on established findings in prior work, or is it an observation from your own analysis? If the former, please provide the necessary citations.
> > > > > >
> > > > > > **Response:** Thank you for this clarification. Our statement that motion and occlusion bias optimization toward texture-invariant, low-frequency cues is a synthesis of established findings, not a standalone intuition. Concretely, prior work shows that (i) dynamics violate the static-scene assumption and produce unreliable photometric gradients (e.g., SC-DepthV3 [3]); (ii) these inconsistencies flatten and multi-modalize the cost-volume / depth-likelihood, effectively spreading likelihood mass across depth hypotheses, thereby rendering the supervision multi-modal and ambiguous (Watson et al. [4]); and (iii) under such ambiguous supervision, neural networks exhibit spectral bias and tend to converge to smoother, low-frequency solutions (Rahaman et al. [5]), a phenomenon further evidenced in dynamic depth settings where corrupted fusion volumes yield noisy supervision (DS-Depth [6]).
> > > > > > We have now made this causal chain explicit and added the above citations in Sec. 3.1 (see our response to Weakness #4).

---

> > ### Comment · Reviewer_WG7J · 2025-11-24
> >
> > I have carefully read the rebuttal and re-examined the mathematical formulations in the paper (specifically Equations 1, 4, 5, and 7). I find the authors' response **factually incorrect** and **contradictory** to the provided equations.
> >
> > 1.  The rebuttal states that MAF (STEC) treats large inconsistent changes as 'uncertain' and 'downweights' them. However, **Equation (1)** defines the mask as $\mathbb{I}[\Delta > \varepsilon]$, which assigns a value of **1** to regions with **large** differences and **0** to stable regions.
> > 2.  **Equation (7)** performs element-wise multiplication: $\mathbf{F}' = \mathcal{K} \odot \mathbf{F}$. Consequently, features in high-difference regions (where $\mathcal{K}=1$) are **retained/preserved**, while features in stable/static regions (where $\mathcal{K}=0$) are **zeroed out**.
> > 3.  This means the Cost Volume (Equation 8) is constructed primarily using features from dynamic/high-difference regions, while static background features are suppressed. This confirms my original concern: the module explicitly highlights dynamic outliers (like the 'oncoming car' example) and forces the network to rely on them for geometric matching, which is mathematically the opposite of 'mitigating motion-induced mismatches'. The textual explanation in the rebuttal attempts to redefine the module's function without addressing this fundamental mathematical reality."
> >
> > The authors claim that EGSI handles dynamic regions via uncertainty. I disagree. Entropy measures prediction *uncertainty* (ambiguity), not *correctness*. In the case of an oncoming car, the STEC module explicitly selects and amplifies its features (Eq. 7), suppressing the static background. This results in a cost volume with a sharp, unambiguous peak at a *wrong* virtual depth caused by the motion.
> > Because the peak is sharp, the calculated entropy (Eq. 10) will be low, resulting in low uncertainty ($\mathcal{U} \approx 0$). Consequently, Equation (11) will effectively bypass the monocular correction and enforce the erroneous multi-view depth ($\tilde{d} \approx d^{\text{mv}}$). STEC essentially acts as a noise-reduction filter that helps the network converge *confidently* to the wrong solution. Therefore, the 'uncertainty-guided fusion' fails precisely where it is needed most.
> >
> > Despite the extensive mathematical derivations and transformations that increase the reading difficulty, I am confident that I have correctly understood this component. `This constitutes a fundamental error.` I request a prompt response. Regarding the issue where oncoming traffic results in larger errors, please refer to [1].
> >
> > [1] Moon, Jaeho, et al. 'From-ground-to-objects: Coarse-to-fine self-supervised monocular depth estimation of dynamic objects with ground contact prior.' Proceedings of the IEEE/CVF Conference on Computer Vision and Pattern Recognition. 2024.

---

> ### Author Response · Authors · 2025-11-24
>
> 2.Rationale for STEC's Design: Given that simple frame differencing is known to be an unreliable cue for handling dynamic objects under general motion, could you provide a more detailed justification for the design of the STEC module? How does it handle common scenarios like opposing traffic or objects moving parallel to the camera at different speeds?
>
> **Response:** Thank you for this follow up question. We fully agree that simple frame differencing is an unreliable supervisory signal. Our STEC (renamed to MAF in the revision) uses inter-frame differences not as supervision, but as a soft gating and stabilization mechanism that estimates multi-view reliability. We summarize the rationale and then explain the behavior in the two scenarios you mention.
>
> **(1) Design Rationale: From Motion Cue to Reliability Gate**
>
> MAF is designed to identify regions where the multi-view rigidity assumption is likely to fail, thereby protecting the cost volume from pervasive spurious matches. Its multi-stage process (Appearance–feature change agreement $\to$ Temporal-gated cost volume) is crucial:
>
> **(i) Reliability over Rigidity.** The output is not a binary mask but a soft, temporally-smoothed score $\mathcal{K}_{\ell,t}$. Regions with consistent evidence across time and feature scales accumulate high values, promoting them as reliable geometric support. Inconsistent regions (including many dynamic ones) are assigned low confidence.
>
> **(ii) Synergy with Downstream Modules.** MAF does not act alone. It flags regions of uncertainty, which are then explicitly handled by EGSI (renamed to SDC in the revision, which reduces trust in the multi-view depth via uncertainty $\mathcal{U}$) and OSCO (renamed to VCI in the revision, which downweights their large residuals in the loss). This pipeline ensures that dynamic regions are processed with caution, not forced into a rigid model.
>
> **(2) Opposing Traffic**
>
> In this scenario, pixels on the cars exhibit large appearance and feature changes, but crucially, these changes are not consistent with a single rigid warp: Such pixels often produce unstable correspondences across depth hypotheses and time, leading to diffuse and noisy cost slices. Through MAF, many of these locations fail the joint appearance-feature consistency checks in some frames, resulting in lower $\mathcal{K}_{\ell,t}$ and weaker contribution to the cost volume. Downstream, SDC interprets the resulting high spectral entropy as uncertainty and increases $\mathcal{U}$, shifting Eq. (8) away from the multi-view estimate $d^{\text{mv}}$ toward the monocular prior $d^{\text{mono}}$. VCI further assigns low credibility weights to large residuals at these locations, preventing them from dominating supervision. Thus, opposing traffic is not treated as valid rigid evidence. It is marked as unreliable for multi-view geometry and handled in an uncertainty-aware way.
>
> **(3) Parallel Motion with Different Speeds**
>
> MAF differentiates between these cases:
>
> (i) Objects that are nearly static in image space (small, consistent changes) tend to satisfy the consistency checks and accumulate moderate-to-high $\mathcal{K}_{\ell,t}$, allowing them to contribute as quasi-static support.
>
> (ii) Objects with stronger or non-uniform motion (different speeds, acceleration) exhibit inconsistent appearance–feature differences across frames, leading to lower expected $\mathcal{K}_{\ell,t}$. Their features are therefore downweighted in the cost volume, and, as in the previous case, SDC and VCI treat them as uncertain—multi-view cues are de-emphasized, and supervision is driven more by stable background and monocular predictions.
>
> **(4) Empirical Justification**
> Our new experiments on BONN RGB-D and TUM-Dynamic (Table R1, reported as Table 7 in the revised manuscript) show clear gains in strongly dynamic scenes. We also report Cityscapes and DrivingStereo results  (Tables R3 and R4, reported as Tables 4 and 5 in the revised manuscript) under Weakness 3, showing consistent competitiveness. In addition, the ablation in Table 2 shows that removing MAF consistently degrades performance. These results support MAF as a stabilizing reliability gate rather than a rigid motion supervisor.
>
> We will make this rationale explicit in the revised manuscript and adjust the wording around MAF to avoid the impression that it uses frame differencing as rigid supervision.

---

> ### Author Response · Authors · 2025-11-24
>
> 3.Rationale for using NYUv2: Could you explain the reasoning behind training and evaluating a method designed for dynamic scenes on the static NYUv2 dataset?
>
> **Response:** Thank you for this question. We agree that NYUv2 is not a dynamic-scene benchmark, and we do not use it as primary evidence for dynamic robustness. Our rationale for including NYUv2 is that:
>
> **(1) Standard indoor benchmark and comparability**
>
> NYUv2 is the standard benchmark for indoor depth estimation. Training and evaluating on NYUv2 is necessary for fair comparison with established indoor baselines [1-3] and to verify that our dynamic-scene regularizations do not degrade performance on widely used static indoor geometry.
>
> **(2) Testing robustness to generic violations of static assumptions**
>
> Although NYUv2 is mostly static in terms of object motion, it contains common violations of ideal static photometric consistency (e.g., occlusions, depth discontinuities, exposure changes, reflective surfaces). Our framework is designed to handle such generic violations of the static-world assumption, not only explicit moving objects. Improved performance on NYUv2 therefore indicates that the proposed method enhances robustness to these broader forms of inconsistency.
>
> **(3) Dedicated validation on truly dynamic benchmarks**
>
> We fully agree that dynamic benchmarks are essential for validating our core claim. In addition to the KITTI video evaluation in our original submission, we have added extensive experiments on BONN RGB-D and TUM-Dynamic for indoor dynamics, and on Cityscapes and DrivingStereo for outdoor driving dynamics (Tables R1, R3 and R4, reported as Tables 4, 5 and 7 in the revised manuscript). These results provide direct evidence of our method’s performance in motion-heavy dynamic settings.
>
> 4.Clarification of Notation:
> In Equation (6), what does the time index t represent? Is it the frame index in a video sequence or the training iteration?
> In Equation (9), what is the index m for the spectral bin? Please provide a clear definition.
>
> **Response:** We thank the reviewer for pointing out these ambiguities. We clarify the notation below and state these definitions explicitly in the revised manuscript.
>
> **Regarding Time Index $t$ in Eq. (6) (now Eq. (4)):** In Eq. (6), **$t$ denotes the frame index in the input video sequence**, not the training iteration. The EMA update for $\mathcal{K}_{\ell,t}$ is applied during the forward pass to temporally smooth the consistency mask across consecutive frames. This stabilizes the mask against transient noise and flickering, yielding a more coherent reliability estimate over time. The parameter $\eta$ controls the retention of the historical mask state.
>
> **Regarding Spectral Bin Index $m$ in Eq. (9) (now before Eq. (7)):** In Eq. (9), **$m$ indexes the frequency bins produced by the 1D Fourier transform $\mathcal{F}_d(\cdot)$ along the depth dimension $D$**. Specifically, $\mathcal{F}_d(\hat{\mathcal{C}})$ applies a 1D FFT to the depth-likelihood vector $\hat{\mathcal{C}}$ at each spatial location $(u',v')$, producing a complex spectrum. $\|\mathcal{F}_d(\hat{\mathcal{C}} )\|_m$ denotes the magnitude at the $m$-th frequency bin, and the denominator $\textstyle \sum_j \|\mathcal{F}_d(\hat{\mathcal{C}})\|_j$ sums magnitudes over all bins to compute total spectral energy. Thus $\Phi_m$ is the normalized spectral magnitude for bin $m$, forming a per-pixel distribution over frequencies that we use to compute entropy and guide adaptive reweighting.

---

> ### Author Response · Authors · 2025-11-24
>
> **Response:** Thank you for the careful re-examination of Eqs. (1), (4), (5), and (7) and for pointing out the mismatch between our previous wording and the mathematical definitions. We agree that our earlier rebuttal phrasing was not fully precise in conveying the intended meaning. Below we clarify the role of MAF (formerly STEC) and why SDC (formerly EGSI) remains effective in dynamic scenes.
>
> **(1)Clarifying the meaning of the MAF mask $\mathcal{K}$**.
>
> In Eq. (1), $\cal K^{\mathrm{seed}}$ is simply a **change flag** between two frames. If a pixel’s appearance changes a lot from $I_{t-1}$ to $I_t$, we set $\mathcal{K}^{\mathrm{seed}}=1$ to mark it as a **potentially problematic correspondence** that needs further checking; otherwise $\mathcal{K}^{\mathrm{seed}}=0$. Importantly, $\mathcal{K}^{\mathrm{seed}}$ does **not** mean the pixel is reliable or confident. Next, Eqs. (2)-(3) keep only those flagged pixels whose features remain consistent across frames, producing $\mathcal{K}^{\mathrm{cons}}_{\ell}$. Eq. (4) then smooths this mask over time with EMA to remove unstable outliers. As a result, the final mask $\mathcal{K}$ includes a pixel **only when it both changes significantly and is still cross-frame consistent**—exactly the challenging regions where multi-view cues are difficult, yet still trustworthy enough to use.
>
> **(2)Why multiplying by $\mathcal{K}$ does not force the network to rely on dynamic outliers**
>
> Eq. (5) uses $\mathcal{K}\odot \mathbf{F}$ to form the MAF-validated features $\mathbf{F}^{'}$, from which the MAF-purified cost volume is constructed. This branch is designed to focus geometric matching on regions where naive static-scene supervision is unreliable, while avoiding domination by easy static background that already yields stable correspondences under standard matching. Moreover, static regions are still fully learned: we keep the standard full-frame photometric/geometry losses and the monocular branch active over all pixels, so static-background supervision is never dropped. Thus MAF is not to “amplify dynamics,” but to prevent dynamic mismatches from contaminating geometric aggregation by restricting cost-volume construction to regions that have been explicitly validated by MAF.

---

> > ### Author Response · Authors · 2025-11-24
> >
> > **(3)On the “sharp wrong peak → low entropy → $\mathcal{U} \approx 0$” argument**
> >
> > We respectfully disagree with the assumption that motion features necessarily produce a sharp but wrong virtual-depth peak. Dynamic/self-occluded regions typically violate the rigid correspondence model, and in most cases this yields diffuse (flat or multi-modal) depth likelihoods rather than confident peaks. Concretely, the MAF-purified motion feature stream $\mathbf{F}'=\mathcal{K}\odot \mathbf{F}$ is matched under a rigid inter-frame projection assumption. For oncoming vehicles, intrinsic geometric inconsistency (e.g., occlusion and non-static warping) **prevents stable correspondences in the cost tensor $\mathcal{C}$**. As a result, the depth likelihood **spreads probability mass across multiple hypotheses and becomes flat or multi-modal**. From an information-theoretic perspective, such diffuse likelihoods carry higher entropy and indicate greater ambiguity; SDC therefore predicts high uncertainty **$\mathcal{U}≈1$**. In this sense, MAF is designed to convert correspondence inconsistency into a measurable uncertainty signal, rather than to amplify dynamic outliers.
> >
> > We acknowledge a known special case where object motion aligns with the epipolar direction, producing a spurious sharp virtual-depth peak. However, this is not the dominant behavior of dynamic mismatches, and our uncertainty is not a hard threshold on raw entropy. SDC uses a learned mapping $\varphi(\cdot)$ over spectral descriptors and neighborhood context, while VCI (formerly OSCO) further prevent over-confident propagation of single corrupted hypotheses.
> >
> > In summary, the mathematics reflect the intended function: MAF first detects high-change regions and then keeps only those that remain cross-frame consistent; it builds a purified cost volume from these validated features; SDC downweights ambiguous likelihoods that are typically diffuse under dynamics, while remaining robust to known special cases. We will revise the text to remove ambiguity and add the above diagnostic evidence.
> >
> > In addition, prior dynamic-depth work shares the consensus motivation of handling dynamic regions differently from static backgrounds. For example, Moon et al. (Sec. 2.2) emphasize that non–camera-aligned motions invalidate static-geometry supervision, motivating explicit identification of moving regions [1]. Our method follows this motivation, but instantiates it via high-recall appearance seeding and consistency-validated purification of cost-volume evidence, rather than simply discarding dynamic pixels.
> >
> > Finally, we sincerely thank the reviewer for the exceptionally careful derivations and responsible comments. Your detailed mathematical scrutiny helps us pinpoint and resolve ambiguities in our presentation, and it directly motivates clearer semantic definitions and more targeted evidence in the revised manuscript, ultimately improving the rigor and readability of this work.

---

> ### Comment · Reviewer_WG7J · 2025-11-25
>
> I have read the authors' clarification regarding the MAF module, but I remain unconvinced. The explanation provided in the rebuttal contradicts the mathematical definitions in the paper, and the underlying logic remains fundamentally flawed for a method proposed in 2025.
>
> The rebuttal states that Eqs. (4)–(5) 'keep only those flagged pixels whose features remain consistent across frames.' However, **Equation (4)** is defined as $\mathcal{G}_{\ell} = \mathbb{I}[\Delta F_{\ell} > \zeta_{\ell}]$$. Mathematically, a feature difference $\Delta F$ larger than a threshold $\zeta$ explicitly signifies **high discrepancy** (i.e., inconsistency), not consistency. By definition, this mask selects regions where features have changed significantly between frames. Claiming that selecting high-difference regions is equivalent to ensuring 'feature consistency' is logically inverted.
>
> My core concern remains: calculating photometric or feature differences directly between two frames ($I_t$ and $I_{t-1}$) **without reprojection (warping)** is physically meaningless for distinguishing dynamic objects from static structures in a moving camera setup.
> * **High Difference ($\Delta > \epsilon$):** This captures both **oncoming vehicles** (dynamic) and **static objects with large parallax** (static). Treating them identically in the cost volume construction (Eq. 8) is incorrect because the former violates the rigid assumption while the latter does not.
> * **Low Difference ($\Delta < \epsilon$):** This captures **distant background** (static) but also **vehicles moving parallel to the camera at similar speeds** (dynamic).
>
> To avoid further disputes, I will not discuss the entropy section, and I will only point out that the MAF section is meaningless. (`cannot` identify potentially problematic correspondence)

---

> ### Author Response · Authors · 2025-11-25
>
> **Response:** Thank you for the follow-up and for the thorough, equation-level examination. We appreciate the opportunity to clarify the meaning of Eqs. (4)-(5) (now Eqs. (2)-(3)), as our prior wording could be misread. As you stated, Eq. (4)  (now Eq. (2)) is **intentionally** a large-change selector that captures salient inter-frame feature variation. In our earlier rebuttal, the term "consistency" was meant to denote **"dual-cue (appearance cue and feature cue) change-agreement"**, rather than **feature similarity**/low-discrepancy matching. To avoid any further misunderstanding, we clarify this interpretation precisely below.
>
> We respectfully clarify the underlying philosophy of MAF, in order to directly address the reviewer’s core concern regarding its physical meaning.
>
> **(1) What Eqs. (4)-(5)  (now Eqs. (2)-(3)) actually do (and what they do not)**
>
> Eq. (1) and Eq. (4) (now Eq. (2)) provide high-recall change cues in two complementary domains: pixel-space appearance change and feature-space change. The Eq. (5) (now Eq. (3)) retains locations where both cues indicate a sufficiently strong inter-frame change, i.e., a dual-cue change-agreement gate**. This serves two purposes: (i) it filters out trivial photometric fluctuations (e.g., mild illumination noise) that may trigger $\big\|I_t(u,v)-I_{t-1}(u,v)\big\|_1>\varepsilon$ but do not yield meaningful feature variation, and (ii) it focuses the cost-volume branch on displacement/parallax-salient regions that carry informative geometric evidence. In short, Eqs. (2)-(3) is a dual-cue change agreement gate.
>
> **(2) Why "difference without warping" is not physically meaningless in our usage**
>
> We agree with the key observation that raw inter-frame differences computed at identical pixel coordinates cannot uniquely distinguish dynamic motion from static parallax under a moving camera. However, MAF does **not** rely on this to separate dynamic from static. Instead, these differences act as a **selection mechanism** to focuses computation on displacement-salient, geometrically informative regions, while the physically meaningful correspondence reasoning happens in Eq. (6) through reprojection/warping.
>
> **High Difference ($\Delta>\varepsilon$): static large parallax vs. oncoming vehicles**
>
> We agree that high $\Delta$ may include both dynamic oncoming vehicles and static nearby structures with large parallax. Crucially, they are **not treated identically once the cost volume is constructed**. The physically meaningful discrimination happens in Eq. (6), where matching is performed after reprojection $\pi(u',v',d)$: $\mathcal{C}^{(\ell)}(u',v',d)=< \mathbf{F}'^{(\ell)}_{t}(u',v'), \mathbf{F}'^{(\ell)}_{t-1}\big(\pi(u',v',d)\big)>$, where $\pi(\cdot,d)$ encodes the rigid geometric hypothesis. For **static** large-parallax regions, the rigid model holds, so there exists a depth $d^\*$ such that $\pi(\cdot,d^\*)$ lands near the true correspondence, yielding stable, geometrically coherent matching responses in $\mathcal{C}(u',v',d)$ (e.g., a consistent peak around the correct depth). In contrast, for **dynamic** oncoming vehicles, rigid correspondence is generally violated; thus no single $d$ consistently explains the inter-frame change, and matching becomes unstable across depth hypotheses. Therefore, high $\Delta$ does not imply static and dynamic content are "treated identically"—Eq. (6) inherently responds differently depending on whether the rigid model is satisfied.
>
> **Low Difference ($\Delta<\varepsilon$): distant static background vs. camera-aligned moving vehicles**
>
> We also agree that low $\Delta$ can include both distant static background (low parallax) and vehicles moving parallel to the camera at similar speeds (low relative image motion). This does not invalidate MAF for two reasons.
>
> First, $\mathcal{K}^{\mathrm{seed}}$ is **not** meant to capture all "problematic correspondences". It is a deliberate saliency gate that prioritizes displacement-rich regions where geometric matching is most informative and where mismatch corruption is most consequential. Distant background with $\Delta<\varepsilon$ typically provides weak parallax and limited geometric gradients; allowing such easy, low-motion regions to dominate the cost-volume aggregation is precisely what the MAF gate is designed to avoid.
>
> Second, pixels not prioritized by this gate are **not removed from learning**: they still contribute gradients through the standard full-image photometric/geometry objectives and the monocular branch, so camera-aligned moving vehicles with $\Delta<\varepsilon$ are not "ignored" by the overall framework.

---

> > ### Comment · Reviewer_WG7J · 2025-11-26
> >
> > I appreciate the authors' clarification that "consistency" in your context refers to the agreement between `image-level and feature-level` changes, rather than `temporal` consistency. However, I maintain that: computing differences on unwarped images—regardless of the feature space or transformation used—is physically **meaningless** for eq. 8.
> >
> > The authors' defense relies heavily on the assumption that dynamic objects will result in "diffuse distributions" or "matching failures" in the cost volume. I find this assumption to be overly optimistic and factually incorrect for rigid bodies in driving scenarios.
> >
> > Cars are rigid bodies with distinct textures. The time interval between frames is minimal, so their appearance remains consistent. When a car moves towards the camera, the network searches along the epipolar line for a match. Because the object is rigid, it often finds a "perfect match" at an incorrect location. An oncoming car actually at 20m may generate optical flow that is geometrically indistinguishable from a static object at 15m. The Cost Volume will calculate a very high, sharp matching score at this incorrect 15m depth. This is not a "diffuse distribution" as the authors claim; it is a sharp but wrong peak. Consequently, the entropy will be low, the system will be highly confident, and the uncertainty module (SDC) will fail to trigger.
> >
> > In the rebuttal, the authors state: "We acknowledge a known special case where object motion aligns with the epipolar direction, producing a spurious sharp virtual-depth peak." **But aligns with / defense with the epipolar direction are SAME!!!**
> >
> > And in standard autonomous driving datasets (KITTI), this is not a special case—it is the dominant scenario. Driving typically occurs on straight roads. The motion vectors of oncoming traffic (and leading vehicles) align almost perfectly with the camera's epipole. While your logic might filter out parallel-moving vehicles (low $\Delta$), it explicitly selects and retains oncoming vehicles (high $\Delta$).
> >
> >  By the authors' own admission, the method produces "spurious sharp peaks" in these aligned scenarios. This means the proposed "bug" affects the most critical and dangerous class of dynamic objects in driving. In the vast majority of cases, the "sharp wrong peak" renders the uncertainty module ineffective, leading to **direct depth errors** and `constitutes a fundamental error.`

---

> > > ### Author Response · Authors · 2025-12-03
> > >
> > > **Response:**  Thank you very much for your response. We would like to kindly clarify a few points that may have led to misunderstanding, and we hope this will help resolve the confusion and address your concern.
> > >
> > > **(1) Unwarped differences are not a rigidity test for Eq. (8), but they are not meaningless.**
> > >
> > > As clarified, MAF does not use $\Delta$ as a rigidity-validity predicate or a dynamic/static classifier. Instead, it serves as a **gate** that prioritizes regions with salient inter-frame changes that are typically more informative for geometric matching when constructing the cost volume. The physically meaningful correspondence verification happens **through** Eq. (8), where matching is evaluated under a depth hypothesis $d$ via reprojection $\pi(u',v',d)$. Accordingly, MAF does **not** claim to eliminate every "sharp-but-wrong" case at the cost-volume stage. It proposes candidate regions for matching, while suppression of over-confident corrupted hypotheses is handled by the subsequent SDC and VCI mechanisms. In other words, the "physical test" is performed by Eq. (8) and later constraints, not by the unwarped differences themselves. We hope this clarification is helpful and addresses the Reviewer’s concern.
> > >
> > > **(2) We do not hinge on the assumption that dynamics must yield diffuse cost volumes. Mitigation of sharp-but-wrong peaks comes from SDC + VCI.**
> > >
> > > The reviewer cites our previous statement that we "acknowledge... spurious sharp virtual-depth peaks" and concludes that this "bug" affects the most critical dynamic objects in driving and that "in the vast majority of cases" a sharp-but-wrong peak renders the uncertainty module ineffective. We respectfully disagree with this inference, as we believe it is not a fair or warranted assumption.
> > > A key clarification is that our acknowledgement of spurious sharp peaks was **not a claim that the framework fails whenever such peaks occur. In the same paragraph, we explicitly stated that** "our uncertainty is not a hard threshold on **raw entropy**. SDC uses a learned mapping $\varphi(\cdot)$ over spectral descriptors and neighborhood context, while VCI further prevents the over-confident propagation of a single corrupted hypothesis." **We hope this clarification makes clear that the quoted sentence was immediately followed by an explanation of why our mitigation does not reduce to raw-entropy thresholding.**
> > >
> > > **(3) Epipolar alignment being common does not imply that sharp-but-wrong peaks systematically dominate most pixels/frames in KITTI and other datasets.**
> > >
> > > **"Approximate alignment" does not automatically imply** that sharp, unambiguous wrong peaks systematically dominate across pixels and frames. The reviewer’s conclusion implicitly requires a much stronger condition, namely that a "perfect match at a single wrong depth" remains stable and dominant over large spatial regions and across many frames. In practice, this condition is frequently disrupted by (a) non-planar depth variation over vehicle surfaces, (b) viewpoint-induced scale and appearance change, (c) occlusion and disocclusion around boundaries, and (d) camera motion not being pure translation, where small yaw and pitch shift the epipole and alter the epipolar field.
> > > **More broadly, real driving also involves turns, road unevenness, lane changes, and frequent occlusions**.
> > > Based on the above, our method is not tuned to a single idealized scenario, but is designed to remain robust under diverse real-world motion patterns and occlusion/disocclusion conditions.
> > >
> > > **(4) Empirical validation on multiple benchmarks with diverse dynamic factors.**
> > >
> > > Finally, our claims are empirically grounded. We validate the method across **multiple benchmarks beyond KITTI**, including outdoor driving datasets (KITTI, Cityscapes, DrivingStereo) and indoor datasets with dynamic content (NYUv2, ScanNet, BONN RGB-D, TUM-Dynamic). This **includes KITTI emphasized in the discussion, as well as Cityscapes and DrivingStereo that we added following the first-round feedback**. Collectively, these benchmarks cover challenging factors such as oncoming and leading traffic in driving scenes, frequent occlusion and disocclusion, and diverse camera motions. We consistently observe improvements across environments and motion/occlusion patterns. We also compare against **representative and recent baselines, including MonoViT, RobustDepth, D4RD, Yan et al., and Jasmine as suggested in the review discussion**, and we obtain competitive performance under the same evaluation protocol. These results support that the effectiveness arises from **the coordinated framework rather than any single component in isolation**.
> > >
> > > We sincerely thank the reviewer again for the feedback.

---

> ### Author Response · Authors · 2025-11-25
>
> In summary, the reviewer’s examples correctly show that $\Delta$ alone cannot serve as a dynamic/static label under camera motion; we agree this point. In fact, our method does not rely on $\Delta$ for that purpose. Instead, it provides a high-recall, dual-cue change-agreement signal to focus matching on displacement-salient regions, while the rigid-vs-nonrigid distinction manifests through the warping-based cost construction in Eq. (6) and subsequent constrain mechanisms in the overall framework.
>
> We thank the reviewer again for encouraging a more precise articulation.

---

### Official Review · Reviewer_6WKi · 2025-10-30

**Soundness:** 3
**Presentation:** 2
**Contribution:** 2
**Rating:** 4
**Confidence:** 4

**Summary:**

This paper tackles self-supervised monocular depth in the wild by filtering and reweighting pixels so that training focuses on trustworthy evidence. First, STEC screens pixels with appearance/feature checks plus short-term temporal smoothing, then builds a cleaner cost volume so each pixel’s ``depth vs. likelihood” curve is sharper and less noisy.  Next, EGSI looks at those curves, uses a simple 1-D FFT to judge their uncertainty (learning which shapes are reliable), and, only during training, uses that uncertainty to blend multi-view depth with a monocular prior; inference cost stays the same. Finally, OSCO applies a soft, robust loss with multi-view consensus that down-weights outliers (occlusions, independently moving objects) instead of hard-dropping everything dynamic. Overall, the idea is not mysterious: it’s a noise-filtering and weighting pipeline that keeps dense predictions but gives low influence to unreliable pixels; the design is sensible and engineering-oriented, with modest novelty mainly in the spectral uncertainty step rather than brand-new building blocks.

**Strengths:**

The paper tackles an important practical problem, robust self-supervised depth in dynamic scenes, using a clear three-layer pipeline that is technically sound and easy to integrate with no extra inference cost. Its spectral (FFT-based) uncertainty is the most original element and plausibly improves where depth likelihoods are ambiguous, with ablations and qualitative results that generally back the claims. Overall, the contribution is useful, likely valuable to practitioners even if the conceptual novelty is moderate.

**Weaknesses:**

Novelty is incremental, and experimental coverage is narrow. Please add seed variance and confidence intervals, as well as sensitivity curves for STEC thresholds, EMA, depth sampling choices, and OSCO’s penalty and consensus settings. In addition, clarity suffers: the figures, especially the method diagram, are cluttered, mix several subplots without explicit inputs or outputs, and make the flow hard to follow.

**Questions:**

(1) EGSI vs. simpler uncertainty: If you replace EGSI with (a) a learned confidence head without FFT and (b) plain Shannon entropy on the depth-likelihood, under the same backbone,  how much performance do you lose? Please provide a controlled ablation to justify the spectral step.

(2) Clarity/IO flow: Can you provide a single input→STEC-gated cost volume→EGSI (train-only) →OSCO loss→outputs diagram with explicit inputs/outputs and a few lines of pseudocode per module? This would resolve much of the current ambiguity.

---

> ### Author Response · Authors · 2025-11-24
>
> We thank the reviewer for the detailed and insightful feedback.
> We are glad that the reviewer finds our pipeline technically sound and practically useful, and we appreciate the suggestions on novelty justification, robustness analysis and clarity.
>
> **Weaknesses:**
>
> 1.Novelty is incremental, and experimental coverage is narrow.
>
> **Response:** Thank you for your thoughtful feedback regarding the novelty and experimental scope of our work. We have significantly revised the manuscript to address these concerns, and we provide a detailed clarification below.
>
> **(1) On the Incremental Novelty**
>
> Our goal is indeed practical and engineering oriented, but our method goes beyond a minor architectural tweak. It introduces a problem-driven, three-level framework that is explicitly tied to the three failure modes highlighted in Fig. 1 (edge misalignment, gradient misalignment, and heavy-tailed photometric errors). Prior works predominantly address these dynamic scene challenges through isolated strategies, such as heuristic masking, external segmentation, or robust loss functions. These are often post-hoc remedies applied to a framework that fundamentally assumes a static world.
>
> In contrast, ROSS-Net reframes robustness as a **coherent Representation–Optimization–Supervision defense**, where MAF (formerly STEC), SDC (formerly EGSI), and VCI (formerly OSCO) jointly address all three failure modes. Importantly, **this is not just a conceptual re-grouping: the components are designed to cooperate.** MAF purifies spatio-temporal correspondences at the feature level so that the cost-volume likelihood becomes sharper and more reliable, enabling SDC to perform meaningful spectral diagnosis; in turn, VCI provides uncertainty-aware order-statistic consensus supervision that prevents residual outliers from dominating gradients, stabilizing the training of the whole system. This synergy is also supported empirically: each module alone yields moderate gains, while combining them consistently produces a larger improvement than any single component (see Table 2 and Fig. 8), e.g., MAF already reduces AbsRel on KITTI by $\sim$8.8% over the baseline and delivers further gains when paired with VCI.
>
> Moreover, each level introduces a non-trivial core mechanism beyond common baselines:
>
> **MAF** goes beyond loss-side auto-masking by performing progressive spatio-temporal epipolar validation directly on feature pyramids and actively purifying the cost volume. **SDC** introduces a spectral-entropy view of cost-volume ambiguity and performs frequency-aware correction to counter low-frequency bias, rather than using a generic confidence head. **VCI** replaces brittle hard masking/robust penalties with trimmed order-statistic consensus (noisy-OR) supervision, which salvages partially consistent cues in dynamic regions instead of discarding them wholesale.
>
> Taken together, our novelty lies in a holistic failure-mode analysis and a set of principled, cooperative mechanisms that defend the entire learning pipeline in dynamic scenes.
>
> **(2) On the Expanded Experimental Coverage**
>
> To comprehensively validate our framework's effectiveness and generalization, we have substantially expanded our experiments beyond the initial submission:
>
> **(i) Indoor Scenes:** We have added comparisons on NYUv2 and ScanNet against strong baselines like SC-DepthV2 [3].
>
> **(ii) Cross-Domain Generalization:** We have conducted new evaluations on BONN RGB-D, TUM-Dynamic, Cityscapes, and DrivingStereo, comparing against recent self-supervised and dynamic scene methods.
>
> These additional results (**Tables R1-R4, reported as Tables 4-7 in the revised manuscript**) provide robust evidence that our method achieves state-of-the-art accuracy and, more importantly, consistent and robust performance across diverse and challenging benchmarks, directly addressing the reviewer’s concern about narrow coverage.
>
> We believe these clarifications and the extensive new empirical evidence firmly establish the significant novelty and broad applicability of our work.

---

> > ### Author Response · Authors · 2025-11-24
> >
> > Table R1 Quantitative comparison on **BONN RGB-D** and **TUM-Dynamic** datasets. "Mode" indicates supervision type: M = Monocular. Best results are in **bold**.
> > | Dataset      | Method            | Size         | Mode | RMSE ↓ | Abs Rel ↓ | δ < 1.25 ↑ | δ < 1.25² ↑ | δ < 1.25³ ↑ |
> > |--------------|-------------------|--------------|------|--------|-----------|------------|-------------|-------------|
> > | BONN RGB-D   | SC-Depth [2]      | 320 × 256    | M    | 0.733  | 0.272     | 0.623      | 0.858       | 0.948       |
> > | BONN RGB-D   | SC-DepthV2 [3]    | 320 × 256    | M    | 0.619  | 0.211     | 0.714      | 0.873       | 0.936       |
> > | BONN RGB-D   | SC-DepthV3 [4]    | 320 × 256    | M    | **0.379** | 0.126  | 0.889      | 0.961       | 0.980       |
> > | BONN RGB-D   | Dyna-MSDepth [5]  | 320 × 256    | M    | 0.385  | 0.120     | **0.898**  | 0.967       | **0.984**   |
> > | BONN RGB-D   | Ours              | 320 × 256    | M    | 0.392  | **0.117** | 0.896      | **0.969**   | **0.984**   |
> > | TUM-Dynamic  | SC-Depth [2]      | 320 × 256    | M    | 0.283  | 0.257     | 0.616      | 0.814       | 0.909       |
> > | TUM-Dynamic  | SC-DepthV2 [3]    | 320 × 256    | M    | 0.282  | 0.223     | 0.643      | 0.862       | 0.932       |
> > | TUM-Dynamic  | SC-DepthV3 [4]    | 320 × 256    | M    | 0.265  | 0.163     | 0.797      | 0.882       | 0.937       |
> > | TUM-Dynamic  | Dyna-MSDepth [5]  | 320 × 256    | M    | 0.259  | 0.157     | 0.801      | **0.885**   | 0.947       |
> > | TUM-Dynamic  | Ours              | 320 × 256    | M    | **0.247** | **0.153** | **0.805** | 0.879       | **0.953**   |
> >
> > Table R2 Indoor scene comparison with **SC-DepthV2**. "Mode" indicates supervision type: M = Monocular. Best results are in **bold**.
> > | Dataset | Method         | Size      | Mode | RMSE ↓ | Abs Rel ↓ | δ < 1.25 ↑ | δ < 1.25² ↑ | δ < 1.25³ ↑ |
> > |---------|----------------|-----------|------|--------|-----------|------------|-------------|-------------|
> > | NYUv2   | SC-DepthV2 [3] | 320 × 256 | M    | 0.532  | 0.138     | 0.820      | 0.956       | 0.989       |
> > | NYUv2   | Ours           | 320 × 256 | M    | **0.497** | **0.128** | **0.835** | **0.965**   | **0.997**   |
> >
> > Table R3 Quantitative comparison on **Cityscapes** dataset. "Mode" indicates supervision type: M = Monocular. Best results are in **bold**.
> > | Method       | Mode | RMSE ↓ | RMSE log ↓ | Sq Rel ↓ | Abs Rel ↓ | δ < 1.25 ↑ | δ < 1.25² ↑ | δ < 1.25³ ↑ |
> > |--------------|------|--------|------------|----------|-----------|------------|-------------|-------------|
> > | Monodepth2   | M    | 8.590  | 0.234      | 1.785    | 0.153     | 0.774      | 0.926       | 0.976       |
> > | SD-SSMDE     | M    | 8.441  | 0.221      | 1.635    | 0.143     | 0.789      | 0.931       | 0.980       |
> > | MonoPP       | M    | 12.113 | —          | 3.156    | 0.216     | 0.580      | —           | —           |
> > | RobustDepth  | M    | 7.912  | 0.224      | 1.569    | 0.160     | 0.757      | 0.937       | 0.982       |
> > | Jasmine      | M    | 6.618  | —          | —        | 0.123     | 0.852      | —           | —           |
> > | Ours         | M    | **5.437** | **0.155** | **0.882** | **0.103** | **0.881** | **0.971**   | **0.992**   |

---

> > > ### Author Response · Authors · 2025-11-24
> > >
> > > Table R4 Quantitative comparison on **DrivingStereo** dataset. "Mode" indicates supervision type: M = Monocular. Best results are in **bold**.
> > > | Domain | Method       | Mode | RMSE ↓ | RMSE log ↓ | Sq Rel ↓ | Abs Rel ↓ | δ < 1.25 ↑ | δ < 1.25² ↑ | δ < 1.25³ ↑ |
> > > |--------|--------------|------|--------|------------|----------|-----------|------------|-------------|-------------|
> > > | Foggy  | MonoDepth2   | M    | 7.927  | 0.195      | 1.514    | 0.125     | 0.849      | 0.950       | 0.980       |
> > > | Foggy  | MonoViT      | M    | 6.313  | 0.150      | 0.934    | **0.096** | 0.893      | 0.974       | 0.993       |
> > > | Foggy  | RobustDepth  | M    | 9.098  | 0.203      | 1.907    | 0.140     | 0.827      | 0.949       | 0.980       |
> > > | Foggy  | D4RD         | M    | 7.102  | 0.154      | 1.061    | 0.105     | 0.883      | 0.975       | **0.994**   |
> > > | Foggy  | Yan et al.   | M    | 6.422  | —          | 1.023    | 0.128     | 0.845      | —           | —           |
> > > | Foggy  | Jasmine      | M    | **5.702** | —       | —        | 0.098     | **0.902**  | —           | —           |
> > > | Foggy  | Ours         | M    | 6.427  | **0.148**  | **0.930**| 0.102     | 0.891      | **0.976**   | 0.991       |
> > > | Cloudy | MonoDepth2   | M    | 6.976  | 0.209      | 1.900    | 0.155     | 0.813      | 0.943       | 0.979       |
> > > | Cloudy | MonoViT      | M    | 5.970  | 0.177      | 1.300    | 0.125     | 0.861      | 0.958       | 0.986       |
> > > | Cloudy | RobustDepth  | M    | 8.269  | 0.231      | 2.281    | 0.173     | 0.782      | 0.933       | 0.973       |
> > > | Cloudy | D4RD         | M    | 7.271  | 0.198      | 1.560    | 0.141     | 0.830      | 0.948       | 0.983       |
> > > | Cloudy | Jasmine      | M    | **5.651** | —       | —        | 0.133     | 0.849      | —           | —           |
> > > | Cloudy | Ours         | M    | 5.833  | **0.169**  | **1.224**| **0.118** | **0.865**  | **0.961**   | **0.987**   |
> > > | Rainy  | MonoDepth2   | M    | 11.040 | 0.301      | 3.339    | 0.240     | 0.591      | 0.857       | 0.952       |
> > > | Rainy  | MonoViT      | M    | 8.604  | 0.219      | 1.925    | 0.169     | 0.733      | 0.934       | 0.985       |
> > > | Rainy  | RobustDepth  | M    | 10.595 | 0.260      | 2.670    | 0.199     | 0.677      | 0.902       | 0.972       |
> > > | Rainy  | D4RD         | M    | 8.584  | 0.208      | **1.722**| 0.158     | 0.773      | 0.946       | 0.985       |
> > > | Rainy  | Yan et al.   | M    | 8.004  | —          | 1.755    | 0.171     | 0.730      | —           | —           |
> > > | Rainy  | Jasmine      | M    | **7.194** | —       | —        | 0.160     | **0.787**  | —           | —           |
> > > | Rainy  | Ours         | M    | 8.293  | **0.197**  | 1.816    | **0.157** | 0.752      | **0.950**   | **0.986**   |
> > > | Sunny  | MonoDepth2   | M    | 6.744  | 0.214      | 1.740    | 0.155     | 0.819      | 0.941       | 0.977       |
> > > | Sunny  | MonoViT      | M    | 6.109  | 0.186      | 1.266    | 0.130     | 0.851      | 0.956       | **0.985**   |
> > > | Sunny  | RobustDepth  | M    | 8.084  | 0.246      | 2.174    | 0.185     | 0.765      | 0.919       | 0.965       |
> > > | Sunny  | D4RD         | M    | 7.121  | 0.207      | 1.437    | 0.149     | 0.815      | 0.946       | 0.980       |
> > > | Sunny  | Ours         | M    | **5.032** | **0.165** | **1.105** | **0.117** | **0.869** | **0.967** | **0.985** |

---

> > > > ### Author Response · Authors · 2025-11-24
> > > >
> > > > 2. Please add seed variance and confidence intervals, as well as sensitivity curves for STEC thresholds, EMA, depth sampling choices, and OSCO’s penalty and consensus settings.
> > > >
> > > > **Response:** We thank the reviewer for this helpful request. We have added sensitivity analyses and multi-seed statistics to evaluate hyperparameter robustness and the reliability of our gains. The results show stable performance across wide parameter ranges, and consistent improvements under different random seeds. Details are provided in **Appendix A.11 ("Seed Variance, Confidence Intervals, and Sensitivity Analysis")**, with quantitative summaries in **Table R5** (**reported as Table 8 in the revised manuscript**) and sensitivity curves in **Figs. 16–20**.
> > > >
> > > > Specifically,
> > > >
> > > > **(1) Seed variance and confidence intervals** (**Table R5**, reported as Table 8 in the revised manuscript): across five random seeds \{42, 43, 44, 45, 46\}, performance varies only slightly, with δ < 1.25 at 0.921 $\pm$ 0.004 and Abs Rel at 0.082 $\pm$ 0.003.
> > > >
> > > > **(2) MAF (formerly STEC) threshold $\varepsilon$** (**Fig. 16**): sweeping $\varepsilon$ in [0, 0.2] keeps δ < 1.25 in 0.918-0.921 and Abs Rel in 0.082-0.085.
> > > >
> > > > **(3) EMA factor $\eta$** (**Fig. 17)**: varying $\eta$ in [0, 1] yields δ < 1.25 in 0.917-0.921 and Abs Rel in 0.082-0.086.
> > > >
> > > > **(4) Depth bins** (**Fig. 18**): using 10-100 bins gives δ < 1.25 in 0.910-0.922 and Abs Rel in 0.082-0.090, showing low sensitivity to discretization.
> > > >
> > > > **(5) VCI (formerly OSCO) penalty $\lambda$** (**Fig. 19**): sweeping $\lambda$ in [0, 16] keeps δ < 1.25 in 0.914-0.921 and Abs Rel in 0.082-0.088.
> > > >
> > > > **(6) Consensus variants** (**Fig. 20**): noisy-OR consensus yields slightly better Abs Rel than avg- or max-consensus, supporting our chosen integration.
> > > >
> > > > Overall, these results indicate that our gains are not driven by delicate hyperparameter tuning, but remain robust across broad settings and consensus variants.
> > > >
> > > > Table R5 Seed variance on KITTI. Abs Rel and $\delta<1.25$ over different random seeds.
> > > > | Seed | 42     | 43     | 44     | 45     | 46     |
> > > > |------|--------|--------|--------|--------|--------|
> > > > | **Abs Rel ↓**        | 0.0818 | 0.0821 | 0.0819 | 0.0822 | 0.0817 |
> > > > | **δ < 1.25 ↑**       | 0.9207 | 0.9212 | 0.9210 | 0.9214 | 0.9208 |
> > > >
> > > > 3. In addition, clarity suffers: the figures, especially the method diagram, are cluttered, mix several subplots without explicit inputs or outputs, and make the flow hard to follow.
> > > >
> > > > **Response:** We thank the reviewer for this constructive feedback on the clarity of our figures. We understand that the current method diagram, which places illustrative subplots alongside the three core levels of our framework, may appear cluttered and obscure the overall pipeline flow. Our intention was to immediately visualize the key contribution of each level, but we agree that the inputs, outputs, and data flow were not explicitly marked.
> > > >
> > > > In the revised version, we have improved figure clarity, particularly for the method diagram (**Fig. 2**), as follows:
> > > >
> > > > (1)Clearly label all inputs and outputs for each module, including feature maps, depth predictions, and supervision signals.
> > > >
> > > > (2)Use explicit arrows and distinct color coding to trace the data flow through the entire pipeline from input frames to final depth.
> > > >
> > > > (3)Potentially separate the high-level architecture overview from the detailed subplots for a cleaner presentation, while ensuring their correspondence is clear.
> > > >
> > > > We believe these changes will make the structure and information flow of our method much easier to follow and will clarify the contribution of each component.

---

> ### Author Response · Authors · 2025-11-24
>
> **Questions:**
>
> (1)EGSI vs. simpler uncertainty: If you replace EGSI with (a) a learned confidence head without FFT and (b) plain Shannon entropy on the depth-likelihood, under the same backbone, how much performance do you lose? Please provide a controlled ablation to justify the spectral step.
>
> **Response:** Thank you for this thoughtful question about isolating the benefit of the spectral step. We agree that such a controlled comparison would be ideal. However, in our setting, the proposed variants (a) and (b) would not be fair ablations because removing the FFT changes the representation that SDC is built on, effectively turning it into a different module and confounding the interpretation.
>
> For (a), SDC is defined around the spectral likelihood representation $\Psi_m$ obtained via a 1D FFT along the depth axis. Removing FFT eliminates $\Psi_m$ and the corresponding spectral bands, so the module no longer exists in its intended form. Any “learned confidence head without FFT” would therefore constitute a new uncertainty module with different inputs and scale, rather than a minimally modified SDC.
>
> For (b), our uncertainty is computed on the FFT-normalized spectral distribution $\Psi_m$, whose entropy measures spectral concentration under dynamic, heavy-tailed noise. Plain Shannon entropy on the raw depth-likelihood lives in a different probability space and has a different numeric scale in multi-modal, motion-corrupted regions. To make it comparable, one must redesign and re-calibrate the uncertainty head and the fusion mapping, which would conflate module redesign with the spectral vs. non-spectral effect. Hence, including such a variant in the rebuttal would not yield a controlled attribution.
>
> While the proposed direct comparisons are not conceptually sound, our existing ablation study in Table 2 provides indirect evidence for the spectral design. Specifically, the baseline contains no explicit uncertainty calibration and relies on unweighted multi-view cues. Adding SDC, which performs FFT-based spectral reweighting and entropy-to-uncertainty mapping, yields a clear and consistent gain, e.g., Abs Rel decreases from 0.102 to 0.096 on KITTI. This controlled improvement supports the effectiveness of the spectral uncertainty calibration as an integrated design over a non-spectral baseline.
>
> We also appreciate the reviewer’s question. Developing a fair non-spectral substitute is a meaningful direction, but it would require introducing a separate uncertainty module and additional design choices that are beyond the scope of this submission.
>
> (2) Clarity/IO flow: Can you provide a single input→STEC-gated cost volume→EGSI (train-only) →OSCO loss→outputs diagram with explicit inputs/outputs and a few lines of pseudocode per module? This would resolve much of the current ambiguity.
>
> **Response:** Thank you for the suggestion. To make the end-to-end pipeline easier to follow at a glance, we have added a single linear I/O diagram in the appendix (**Sec. "I/O Flow Clarification", Fig. 21**). This figure explicitly traces the computation from inputs to outputs with labeled tensors: input frames $\{I_t,I_{t-1}\}$ $\to$ the MAF-gated multi-scale cost volume $\mathcal{C}$ $\to$ SDC spectral correction, which produces uncertainty $\mathcal{U}$ and the fused depth $\widetilde{d}$) $\to$ VCI consensus masks and the associated loss $\to$ final depth output.
> We additionally provide standalone pseudocode for each core component, with MAF in **Algorithm 3 (appendix A.6)**, SDC in **Algorithm 4 (appendix A.6)**, and VCI in **Algorithm 5 (appendix A.6)**, together with the full training and inference procedures in Algorithm 1 and Algorithm 2. These algorithms specify the exact inputs and outputs and summarize the key computation steps for each module. We also clarify train-vs-test usage: the uncertainty branch in SDC and the consensus supervision in VCI are used only during training, while inference runs only the forward prediction path (MonoDepth2, MAF gating, cost-volume construction, depth heads), introducing no inference-time overhead.
>
> We hope these additions make the pipeline and module roles fully transparent, and we thank the reviewer again for helping us improve the clarity of the presentation.

---

### Official Review · Reviewer_abFq · 2025-11-01

**Soundness:** 4
**Presentation:** 4
**Contribution:** 4
**Rating:** 6
**Confidence:** 5

**Summary:**

This paper presents a self-supervised depth estimation framework designed for both single-view and multi-frame inputs, trained on video sequences. The authors address three key challenges—edge misalignment, gradient misalignment, and heavy-tailed photometric errors—at three corresponding levels: representation, optimization, and supervision. The proposed approach is novel and demonstrates superior performance across four benchmark datasets compared to existing methods.

**Strengths:**

1. The paper is well-written and technically solid.

2. The motivation is clear: the authors systematically identify three core challenges in self-supervised depth estimation and propose corresponding solutions at different levels to address them.

3. The method is thoroughly evaluated on four datasets—two outdoor and two indoor—showing consistent performance improvements over previous approaches.

4. Comprehensive ablation studies and qualitative comparisons effectively validate the proposed contributions and demonstrate the method’s efficacy.

**Weaknesses:**

1. The model architecture is not clearly described. Is it based on a ResNet, a Transformer, or another backbone?

2. Although the authors discuss challenges related to dynamic scene reconstruction, the proposed method is not evaluated on datasets containing significant dynamic motion, such as the BONN RGB-D or TUM-Dynamic datasets.

3. In the indoor scene evaluation, the paper “Auto-Rectify Network for Unsupervised Indoor Depth Estimation” (TPAMI 2022) should be included for a more complete comparison.

4. The proposed method still requires separate training for different datasets. It would be valuable to explore whether a single unified model, trained on a mixed dataset containing both indoor and outdoor scenes, could generalize across diverse environments.

**Questions:**

NA

---

> ### Author Response · Authors · 2025-11-24
>
> We thank the reviewer for the encouraging comments.
> We are pleased that the reviewer finds the paper well-written, technically solid, and well-motivated, and that the experimental evaluation and ablation studies are considered comprehensive.
> Below we address the concerns point by point.
>
> 1.The model architecture is not clearly described. Is it based on a ResNet, a Transformer, or another backbone?
>
> **Response:** Thank you for pointing this out. Our method builds on the standard Monodepth2 architecture [1], using a **ResNet-based** encoder and a convolutional upsampling decoder pretrained on ImageNet. We acknowledge that this detail, mentioned in Appendix A.5, should be highlighted in the main text. Accordingly, we have explicitly **added this description to Sec. 4.1 ("Setups")** in the revised manuscript. We hope this clarification addresses the concern.
>
> [1] Godard, C., et al. "Digging into self-supervised monocular depth estimation." ICCV 2019.
>
> 2.Although the authors discuss challenges related to dynamic scene reconstruction, the proposed method is not evaluated on datasets containing significant dynamic motion, such as the BONN RGB-D or TUM-Dynamic datasets.
>
> **Response:** Thank you for this insightful suggestion. We agree that evaluation on dynamic scenes is crucial. **Following the reviewer’s recommendation, we have conducted additional experiments on the BONN RGB-D and TUM-Dynamic datasets,** comparing our method with SC-Depth [2], SC-DepthV2 [3], SC-DepthV3 [4], and Dyna-MSDepth [5].
>
> The new quantitative results in **Table R1** (**reported as Table 7 in the revised manuscript**) demonstrate the competitiveness of our method. For instance, we outperform SC-Depth [2], SC-DepthV2 [3], SC-DepthV3 [4], and Dyna-MSDepth [5] by 56.99\%, 44.55\%, 7.14\%, and 2.5\%, respectively, in terms of the AbsRel metric on BONN RGB-D. These results indicate that our method is particularly effective in dynamic scene reconstruction.
>
> The new qualitative results in **Fig. 15** in the revised manuscript show that our method yields sharper and more coherent depth contours on moving people, with noticeably less boundary bleeding than SC-DepthV3. It also maintains clearer foreground–background separation and more consistent depth ordering in the multi-person scene.
>
> We are grateful for this comment, which has allowed us to provide stronger evidence for the robustness of our approach.
>
> Table R1 Quantitative comparison on **BONN RGB-D** and **TUM-Dynamic** datasets. "Mode" indicates supervision type: M = Monocular. Best results are in **bold**.
> | Dataset      | Method            | Size         | Mode | RMSE ↓ | Abs Rel ↓ | δ < 1.25 ↑ | δ < 1.25² ↑ | δ < 1.25³ ↑ |
> |--------------|-------------------|--------------|------|--------|-----------|------------|-------------|-------------|
> | BONN RGB-D   | SC-Depth [2]      | 320 × 256    | M    | 0.733  | 0.272     | 0.623      | 0.858       | 0.948       |
> | BONN RGB-D   | SC-DepthV2 [3]    | 320 × 256    | M    | 0.619  | 0.211     | 0.714      | 0.873       | 0.936       |
> | BONN RGB-D   | SC-DepthV3 [4]    | 320 × 256    | M    | **0.379** | 0.126  | 0.889      | 0.961       | 0.980       |
> | BONN RGB-D   | Dyna-MSDepth [5]  | 320 × 256    | M    | 0.385  | 0.120     | **0.898**  | 0.967       | **0.984**   |
> | BONN RGB-D   | Ours              | 320 × 256    | M    | 0.392  | **0.117** | 0.896      | **0.969**   | **0.984**   |
> | TUM-Dynamic  | SC-Depth [2]      | 320 × 256    | M    | 0.283  | 0.257     | 0.616      | 0.814       | 0.909       |
> | TUM-Dynamic  | SC-DepthV2 [3]    | 320 × 256    | M    | 0.282  | 0.223     | 0.643      | 0.862       | 0.932       |
> | TUM-Dynamic  | SC-DepthV3 [4]    | 320 × 256    | M    | 0.265  | 0.163     | 0.797      | 0.882       | 0.937       |
> | TUM-Dynamic  | Dyna-MSDepth [5]  | 320 × 256    | M    | 0.259  | 0.157     | 0.801      | **0.885**   | 0.947       |
> | TUM-Dynamic  | Ours              | 320 × 256    | M    | **0.247** | **0.153** | **0.805** | 0.879       | **0.953**   |
>
> [2]Bian, Jia-Wang, Huangying Zhan, Naiyan Wang, Zhichao Li, Le Zhang, Chunhua Shen, Ming-Ming Cheng, and Ian Reid. "Unsupervised scale-consistent depth learning from video." International Journal of Computer Vision 129, no. 9 (2021): 2548-2564.
>
> [3]Bian, Jia-Wang, Huangying Zhan, Naiyan Wang, Tat-Jun Chin, Chunhua Shen, and Ian Reid. "Auto-rectify network for unsupervised indoor depth estimation." IEEE transactions on pattern analysis and machine intelligence 44, no. 12 (2021): 9802-9813.
>
> [4]Sun, Libo, Jia-Wang Bian, Huangying Zhan, Wei Yin, Ian Reid, and Chunhua Shen. "Sc-depthv3: Robust self-supervised monocular depth estimation for dynamic scenes." IEEE transactions on pattern analysis and machine intelligence 46, no. 1 (2023): 497-508.
>
> [5]Yao, Jianjun, Yingzhao Li, and Jiajia Li. "Dyna-MSDepth: multi-scale self-supervised monocular depth estimation network for visual SLAM in dynamic scenes." Machine Vision and Applications 35, no. 5 (2024): 115.

---

> > ### Author Response · Authors · 2025-11-24
> >
> > 3.In the indoor scene evaluation, the paper “Auto-Rectify Network for Unsupervised Indoor Depth Estimation” (TPAMI 2022) should be included for a more complete comparison.
> >
> > **Response:** We thank the reviewer for this constructive suggestion. **Following this advice, we have incorporated "Auto-Rectify Network for Unsupervised Indoor Depth Estimation" (TPAMI 2022),** denoted as **SC-DepthV2** [3], into our indoor-scene evaluation on the standard NYUv2 benchmark.
> > The updated results are presented in **Table R2** (**reported as Table 6 and Fig. 6 in the revised manuscript**). On NYUv2, our method outperforms SC-DepthV2 across all evaluation metrics. These results robustly validate the effectiveness of our approach in indoor settings.
> >
> > Table R2 Indoor scene comparison with **SC-DepthV2**. "Mode" indicates supervision type: M = Monocular. Best results are in **bold**.
> > | Dataset | Method         | Size      | Mode | RMSE ↓ | Abs Rel ↓ | δ < 1.25 ↑ | δ < 1.25² ↑ | δ < 1.25³ ↑ |
> > |---------|----------------|-----------|------|--------|-----------|------------|-------------|-------------|
> > | NYUv2   | SC-DepthV2 [3] | 320 × 256 | M    | 0.532  | 0.138     | 0.820      | 0.956       | 0.989       |
> > | NYUv2   | Ours           | 320 × 256 | M    | **0.497** | **0.128** | **0.835** | **0.965**   | **0.997**   |
> >
> > 4.The proposed method still requires separate training for different datasets. It would be valuable to explore whether a single unified model, trained on a mixed dataset containing both indoor and outdoor scenes, could generalize across diverse environments.
> >
> > **Response:** We thank the reviewer for the valuable suggestion on unified indoor–outdoor training. We agree that a single model trained on a mixed dataset would be highly desirable for real-world deployment.
> >
> > In this submission, we follow the standard domain-specific training protocol to ensure controlled and fair comparisons with prior work. Mixed indoor–outdoor training introduces substantial domain shifts (e.g., heterogeneous intrinsics, depth ranges, geometry statistics, and motion patterns), which can confound ablation-based attribution of gains and is beyond the scope of the current paper. Importantly, our improvements are not tied to any domain-specific assumption: MAF relies on temporal consistency to downweight dynamic/occluded evidence, SDC calibrates frequency content from the cost volume, and VCI performs uncertainty-aware residual consensus. These mechanisms are intrinsically applicable to both indoor and outdoor scenes.
> >
> > We consider unified mixed-domain training a strong and important next step. We will explicitly add this as a future direction, noting the key challenges (e.g., depth-range normalization and camera-parameter heterogeneity) and that our framework provides a natural foundation to tackle them.

---

### Author Response · Authors · 2025-12-03

We sincerely thank the AC and all reviewers for their time and constructive feedback. We appreciate Reviewer **abFq** for recognizing the paper as well-written and technically solid, with clear motivation and comprehensive ablation studies; Reviewer **6WKi** for noting that the contribution is useful and likely valuable to practitioners; Reviewer **WG7J** for highlighting an ambitious technical approach and strong empirical results; and Reviewer **UPCe** for finding the motivation intuitive and the appendix rich.
﻿

In this rebuttal, we not only address presentation issues but also strengthen the paper along four core dimensions: **(i) contribution positioning, (ii) soundness clarification, (iii) empirical coverage, and (iv) reproducibility and clarity.**
﻿

First, we clarify that our main contribution is a holistic robustness framework that explicitly targets three coupled failure modes in self-supervised depth under dynamics and occlusions: representation corruption, optimization bias toward texture-invariant low-frequency cues, and unreliable supervision under heavy-tailed residuals. This is realized through three interacting mechanisms, namely MAF, SDC, and VCI, which are designed as a coherent defense across the full learning pipeline rather than as isolated heuristics.
﻿

Second, regarding concerns about the MAF module, we revise the exposition to avoid any misleading interpretation that frame differences are used as rigid supervision. We also clarify the relevant notation around the associated equations.
﻿

Third, we substantially broaden the evidence for generalization and robustness. Beyond the original setting, we add dynamic-scene evaluations on BONN RGB-D, TUM-Dynamic, Cityscapes, and DrivingStereo, include strong indoor baselines (e.g., SC-DepthV2) on NYUv2, with new quantitative and qualitative results (Tables R1–R4, Fig. 15). We further provide results across multiple random seeds (Table 8) and hyperparameter sensitivity curves (Figs. 16–20), showing that the gains are consistent and not driven by fragile tuning.
﻿

To improve clarity, we restructure the presentation with a single input $\to$ output I/O flow figure (Fig. 21) and provide standalone pseudocode for each module (Algorithms 3–5). In addition, we simplify several equations, add missing and more appropriate related-work citations, clarify potentially confusing symbols, standardize module naming throughout, and tighten the writing to better align motivation, method, and empirical evidence. Overall, the revision strengthens the paper’s causal motivation, resolves soundness concerns, substantially expands evaluation, and makes the design raceable and reproducible.
﻿

We again thank the AC and the reviewers for their careful consideration, and we hope the clarified formulation and expanded evidence satisfactorily address the remaining concerns.

---

### Meta-Review · Area_Chair_kt8r · 2026-01-13

**Summary:**

The main unresolved concern is about the soundness of the MAF (formerly STEC) module. Reviewer WG7J provided a concrete equation-level argument that the gating mechanism selects high-change regions in a way that can emphasize dynamic outliers and potentially produce confident but incorrect depth estimates in epipolar-aligned motion scenarios (e.g., oncoming traffic). The authors’ rebuttal reinterprets the role of MAF but does not modify the formulation, and the reviewer’s counterexample remains insufficiently addressed.

**Reviewer Concerns:**

The rebuttal successfully addressed several secondary concerns, including expanding the experimental evaluation to additional dynamic and cross-domain benchmarks, improving the clarity of the presentation and notation, adding pseudocode and a clearer pipeline description, reporting multi-seed and hyperparameter sensitivity results, and strengthening the motivation and related-work discussion. However, a core concern remains unresolved: the soundness of the MAF (formerly STEC) module. In particular, the equation-level argument that the unwarped change-based gating can emphasize dynamic outliers and lead to confident but incorrect depth estimates in common driving scenarios (e.g., epipolar-aligned motion) was not convincingly refuted, and the rebuttal mainly reinterprets the intent of the module without changing its formulation or directly addressing the reviewer’s counterexample.

**Reviewer Scores:**

Overall, the discussion and rebuttal would likely have led the more positive or borderline reviewers to slightly increase or maintain their scores (Reviewer abFq likely maintaining or moving from 6 to 7, and Reviewer 6WKi likely increasing from 4 to around 5), as their concerns about experiments, clarity, and stability were largely addressed. In contrast, the reviewers who raised fundamental conceptual or soundness concerns (Reviewers WG7J and UPCe) would likely not have changed their scores, since their core objections about the validity and value of the approach were not resolved by the rebuttal.

---

### Decision · Program_Chairs · 2026-01-26

Reject